# Improved Regret for Decentralized Online Convex Optimization with Compressed Communication

## Abstract

We investigate decentralized online convex optimization with compressed communication, where $n$ learners collaboratively minimize a sequence of global loss functions using only local information and compressed data from their neighbors. Prior work has established regret bounds of $O(\max\{\omega^{-2}\rho^{-4}n^{1/2}, \omega^{-4}\rho^{-8}\}n\sqrt{T})$ and $O(\max\{\omega^{-2}\rho^{-4}n^{1/2}, \omega^{-4}\rho^{-8}\}n\ln T)$ for convex and strongly convex functions, respectively, where $\omega \in (0,1]$ is the compression quality factor ($\omega = 1$ means no compression) and $\rho < 1$ is the spectral gap of the communication matrix. However, these regret bounds suffer from a *quadratic* or even *quartic* dependence on $\omega^{-1}$. Moreover, the *super-linear* dependence on $n$ is also undesirable. To overcome these limitations, we propose a novel algorithm that achieves improved regret bounds of $\tilde{O}(\omega^{-1/2}\rho^{-1}n\sqrt{T})$ and $\tilde{O}(\omega^{-1}\rho^{-2}n\ln T)$ for convex and strongly convex functions, respectively. The primary idea is to design a *two-level blocking update framework* incorporating two novel ingredients: an online gossip strategy and an error compensation scheme, which collaborate to *achieve a better consensus* among local learners. Furthermore, we establish the first lower bounds for this problem, justifying the optimality of our results with respect to both $\omega$ and $T$. Additionally, we consider the bandit feedback scenario, and extend our method with the classic gradient estimators to enhance existing regret bounds.

## 1 Introduction

Decentralized online convex optimization (D-OCO) (Yan et al., 2012; Hosseini et al., 2013) has emerged as a fundamental framework for modeling distributed real-world problems, such as tracking in sensor networks (Li et al., 2002; Lesser et al., 2003) and dynamic packet routing (Awerbuch & Kleinberg, 2004). Specifically, it is formulated as an iterative game between an adversary and a set of local learners, indexed by $1, \ldots, n$, which are connected through a network defined by an undirected graph $\mathcal{G} = ([n], E)$ with $E \subseteq [n] \times [n]$. In each round $t \in [T]$, learner $i \in [n]$ selects a decision $\mathbf{x}_i(t)$ from a convex set $\mathcal{X} \subseteq \mathbb{R}^d$. Subsequently, the adversary chooses a group of convex loss functions $f_{t,i}(\cdot) \colon \mathbb{R}^d \to \mathbb{R}$ and learner $i$ suffers a loss $f_{t,i}(\mathbf{x}_i(t))$. The goal of learner $i$ is to minimize the cumulative loss in terms of the *global* function $f_t(\mathbf{x}) = \sum_{j=1}^n f_{t,j}(\mathbf{x})$ over $T$ rounds, which is equivalent to minimizing the regret

$$R(T, i) = \sum_{t=1}^T \sum_{j=1}^n f_{t,j}(\mathbf{x}_i(t)) - \min_{\mathbf{x} \in \mathcal{X}} \sum_{t=1}^T \sum_{j=1}^n f_{t,j}(\mathbf{x}). \tag{1}$$

The key difficulty in D-OCO lies in the fact that each learner only has access to its local function $f_{t,i}(\mathbf{x})$. To approximate the global loss $f_t(\mathbf{x})$, prior studies (Yan et al., 2012; Hosseini et al., 2013; Zhang et al., 2017b; Wan et al., 2020; 2024; 2025) adopt the gossip protocols to aggregate information about the global loss function, where each learner communicates with its neighbors based on a weight matrix $P$. Nevertheless, the information (e.g., gradients and decisions) transmitted among learners incurs significant *communication overhead* when the number of learners $n$ is large, which limits the practical applicability of these methods in distributed problems.

To tackle the communication bottleneck in D-OCO, Tu et al. (2022) propose a communication-efficient method by leveraging data compression techniques (Tang et al., 2018; Koloskova et al.,

Table 1: A comparison of our work with existing results (Tu et al., 2022) for D-OCO with compressed communication. Here, $n$ is the number of learners, $\rho < 1$ is the spectral gap of the communication matrix and $\omega \in (0, 1]$ is the compression ratio.

| Source | Loss functions | Regret bounds |
|---|---|---|
| Tu et al. (2022) | Convex | $O(\max\{\omega^{-2}\rho^{-4}n^{1/2}, \omega^{-4}\rho^{-8}\}n\sqrt{T})$ |
| | Strongly convex | $O(\max\{\omega^{-2}\rho^{-4}n^{1/2}, \omega^{-4}\rho^{-8}\}n\ln T)$ |
| **This work** | Convex | $O(\omega^{-1/2}\rho^{-1}n\sqrt{\ln n}\sqrt{T})$ |
| | Strongly convex | $O(\omega^{-1}\rho^{-2}n\ln n\ln T)$ |
| **Lower bound** | Convex | $\Omega(\omega^{-1/2}\rho^{-1/4}n\sqrt{T})$ |
| | Strongly convex | $\Omega(\omega^{-1}\rho^{-1/2}n\ln T)$ |

2019) to reduce the volume of transmitted information. Their method, termed DC-DOGD, achieves $O(\max\{\omega^{-2}\rho^{-4}n^{1/2}, \omega^{-4}\rho^{-8}\}n\sqrt{T})$ and $O(\max\{\omega^{-2}\rho^{-4}n^{1/2}, \omega^{-4}\rho^{-8}\}n\ln T)$ regret bounds for convex and strongly convex loss functions, respectively, where $\omega \in (0, 1]$ is the compression ratio that characterizes the quality of compression ($\omega = 1$ means no compression) and $\rho < 1$ is the spectral gap of the communication matrix. However, their regret bounds suffer from a *quadratic* or even *quartic* dependence on $\omega^{-1}$. To enhance the communication efficiency, it is common to employ a compressor with $\omega \ll 1$. In this case, the theoretical guarantees of their method degrade significantly. Moreover, the dependence on $n$ is far from that in $\Omega(\rho^{-1/4}n\sqrt{T})$ and $\Omega(\rho^{-1/2}n\ln T)$ lower bounds for convex and strongly convex functions in D-OCO (Wan et al., 2025). Thus, it is natural to ask whether *the regret bounds in D-OCO with compressed communication could be further improved*.

**Results.** In this paper, we give an affirmative answer to this question. Specifically, we develop a novel algorithm termed T̲wo-level C̲ompressed D̲ecentralized O̲nline G̲radient D̲escent (Top-DOGD), which enjoys better regret bounds of $\tilde{O}(\omega^{-1/2}\rho^{-1}n\sqrt{T})$ and $\tilde{O}(\omega^{-1}\rho^{-2}n\ln T)$ for convex and strongly convex functions, respectively.[1] Furthermore, we establish nearly matching lower bounds of $\Omega(\omega^{-1/2}\rho^{-1/4}n\sqrt{T})$ and $\Omega(\omega^{-1}\rho^{-1/2}n\ln T)$ for convex and strongly convex functions, which are the first lower bounds for this problem. To demonstrate the significance of our work, we present a comparison of our results with Tu et al. (2022) in Table 1. Additionally, we consider the bandit feedback setting, and extend our method with classic gradient estimators (Flaxman et al., 2005; Agarwal et al., 2010). Let $d$ denote the dimensionality. In the one-point bandit feedback setting, we enhance the existing regret bounds of $O(\max\{\omega^{-1}\rho^{-2}n^{1/4}, \omega^{-2}\rho^{-4}\}d^{1/2}nT^{3/4})$ and $O(\max\{\omega^{-2/3}\rho^{-4/3}n^{1/6}, \omega^{-4/3}\rho^{-8/3}\}d^{2/3}nT^{2/3}(\ln T)^{1/3})$ for convex and strongly convex functions to $\tilde{O}(\omega^{-1/4}\rho^{-1/2}d^{1/2}nT^{3/4})$ and $\tilde{O}(\omega^{-1/3}\rho^{-2/3}d^{2/3}nT^{2/3}(\ln T)^{1/3})$. In the two-point bandit feedback setting, we improve the existing regret bounds of $O(\max\{\omega^{-2}\rho^{-4}n^{1/2}, \omega^{-4}\rho^{-8}\}dn\sqrt{T})$ and $O(\max\{\omega^{-2}\rho^{-4}n^{1/2}, \omega^{-4}\rho^{-8}\}d^2n\ln T)$ to $\tilde{O}(\omega^{-1/2}\rho^{-1}dn\sqrt{T})$ and $\tilde{O}(\omega^{-1}\rho^{-2}d^2n\ln T)$. We compare our results with previous work in the bandit feedback setting in Table 2. Experimental results on online classification demonstrate the effectiveness of our methods, which can be found in Appendix D.

**Techniques.** The technical contribution of this paper lies in the development of two novel online strategies to weaken the impacts of decentralization, compression and projection on the regret in D-OCO, together with a *unified framework* that integrates them. Specifically, the effects consist of three components: consensus error, compression error, and projection error. To control the first two, we devise an online compressed gossip strategy, which is achieved through *multiple steps of gossip*. To handle the projection error, we propose a projection error compensation scheme, which *recursively compresses the residual* of the projection error and transmits the data to neighbors at every recursion step. However, both of these techniques inherently require multiple communication rounds per update, which is not allowed in D-OCO. To overcome this dilemma, we design a *two-level blocking update framework*. We divide the total $T$ rounds into blocks of size $L = L_1 + L_2$ and only update the decision *once* at the end of each block. Within each block, we first apply the online compressed gossip strategy over $L_1$ rounds, and then perform the projection error compensation scheme over $L_2$ rounds. Since we only update the decision once per block, we can evenly distribute

---

[1]We use the $\tilde{O}(\cdot)$ notation to hide constant factors and polylogarithmic factors in $n$.

Table 2: A comparison of our work with existing results (Tu et al., 2022) for D-OCO with compressed communication under bandit feedback setting. Here, $d$ is the dimensionality, (1) and (2) denote one-point and two-point bandit feedback settings, respectively.

| Source | Settings | Regret bounds |
|---|---|---|
| Tu et al. (2022) | Convex (1) | $O\left(\max\{\omega^{-1}\rho^{-2}n^{1/4}, \omega^{-2}\rho^{-4}\}d^{1/2}nT^{3/4}\right)$ |
| | Strongly convex (1) | $O(\max\{\omega^{-2/3}\rho^{-4/3}n^{1/6}, \omega^{-4/3}\rho^{-8/3}\}d^{2/3}nT^{2/3}(\ln T)^{1/3})$ |
| | Convex (2) | $O(\max\{\omega^{-2}\rho^{-4}n^{1/2}, \omega^{-4}\rho^{-8}\}dn\sqrt{T})$ |
| | Strongly convex (2) | $O(\max\{\omega^{-2}\rho^{-4}n^{1/2}, \omega^{-4}\rho^{-8}\}d^2 n\ln T)$ |
| **This work** | Convex (1) | $O(\omega^{-1/4}\rho^{-1/2}d^{1/2}n(\ln n)^{1/4}T^{3/4})$ |
| | Strongly convex (1) | $O(\omega^{-1/3}\rho^{-2/3}d^{2/3}n(\ln n)^{1/3}T^{2/3}(\ln T)^{1/3})$ |
| | Convex (2) | $O(\omega^{-1/2}\rho^{-1}dn\sqrt{\ln n}\sqrt{T})$ |
| | Strongly convex (2) | $O(\omega^{-1}\rho^{-2}d^2 n\ln n\ln T)$ |

the communications across rounds. By selecting appropriate block sizes $L_1$ and $L_2$, we can improve the regret bound while ensuring a single communication per round.

## 2 DECENTRALIZED ONLINE CONVEX OPTIMIZATION (D-OCO)

D-OCO is a generalization of online convex optimization (Hazan et al., 2016) with $n \geq 2$ local learners connected through a network defined by an undirected graph $\mathcal{G} = ([n], E)$ with $E \subseteq [n] \times [n]$. Different from centralized OCO, each learner $i$ in D-OCO aims to minimize the regret with respect to the global function $f_t(\mathbf{x}) = \sum_{j=1}^n f_{t,j}(\mathbf{x})$, while only having access to its local function $f_{t,i}(\mathbf{x})$ and the information from its neighbors. The pioneering work of Yan et al. (2012) proposes a decentralized variant of OGD (Zinkevich, 2003), named D-OGD, by directly applying the standard gossip step (Xiao & Boyd, 2004) to the local decisions, and performing a gradient descent update using the gradient of the local function. D-OGD achieves $O(\rho^{-1/2}n^{5/4}\sqrt{T})$ and $O(\rho^{-1}n^{3/2}\ln T)$ regret bounds for convex and strongly convex loss functions, respectively. Later, Hosseini et al. (2013) develop a decentralized variant of FTRL (Hazan et al., 2007), termed D-FTRL, which enjoys the same regret bounds as D-OGD. Notably, there exist large gaps between these bounds and the lower bounds established by Wan et al. (2022), i.e., $\Omega(n\sqrt{T})$ and $\Omega(n)$ lower bounds for convex and strongly convex functions. To fill these gaps, Wan et al. (2025) design an online accelerated gossip strategy and enhance the regret bounds to $O(\rho^{-1/4}n\sqrt{\ln n}\sqrt{T})$ and $O(\rho^{-1/2}n\ln n\ln T)$. They further demonstrate the optimality of these upper bounds by deriving tighter $\Omega(\rho^{-1/4}n\sqrt{T})$ and $\Omega(\rho^{-1/2}n\ln T)$ lower bounds for convex and strongly convex functions.

In practice, the efficacy of D-OCO algorithms may be limited by the communication overhead associated with exchanging information. To alleviate communication costs, several works (Tu et al., 2022; Yuan et al., 2022; Cao & Bacsar, 2023; Zhang et al., 2023) seek to transmit the compressed data $\mathcal{C}(\mathbf{x})$ with fewer bits instead of broadcasting the full vector $\mathbf{x}$, where $\mathbf{x} \in \mathbb{R}^d$ and $\mathcal{C}(\cdot) : \mathbb{R}^d \to \mathbb{R}^d$ is a compression operator such that $\mathcal{C}(\mathbf{x})$ can be more efficiently transmitted. In particular, Tu et al. (2022) propose a communication-efficient method by leveraging compressors, and establishing $O(\max\{\omega^{-2}\rho^{-4}n^{1/2}, \omega^{-4}\rho^{-8}\}n\sqrt{T})$ and $O(\max\{\omega^{-2}\rho^{-4}n^{1/2}, \omega^{-4}\rho^{-8}\}n\ln T)$ regret bounds for convex and strongly convex loss functions. Moreover, they consider the bandit setting, where the learner only has access to the loss value. In the one-point bandit feedback setting (Flaxman et al., 2005), their method achieves $O(\max\{\omega^{-1}\rho^{-2}n^{1/4}, \omega^{-2}\rho^{-4}\}d^{1/2}nT^{3/4})$ and $O(\max\{\omega^{-2/3}\rho^{-4/3}n^{1/6}, \omega^{-4/3}\rho^{-8/3}\}d^{2/3}nT^{2/3}(\ln T)^{1/3})$ regret bounds for convex and strongly convex loss functions, where $d$ is the dimensionality. In the two-point bandit feedback setting (Agarwal et al., 2010), Tu et al. (2022) establish $O(\max\{\omega^{-2}\rho^{-4}n^{1/2}, \omega^{-4}\rho^{-8}\}dn\sqrt{T})$ and $O(\max\{\omega^{-2}\rho^{-4}n^{1/2}, \omega^{-4}\rho^{-8}\}d^2 n\ln T)$ regret bounds for convex and strongly convex loss functions. A contemporaneous work (Cao & Bacsar, 2023) provides the same regret bound for convex loss functions in D-OCO with compressed communication.

We also provide additional discussions of compressed communication in Appendix A.

## 3 MAIN RESULTS

In this section, we first present preliminaries for D-OCO, including the assumptions and techniques employed in our algorithmic design. We then introduce our method that achieves improved regret bounds, and establish nearly matching lower bounds for D-OCO with compressed communication.

### 3.1 PRELIMINARIES

Similar to the previous work on D-OCO (Yan et al., 2012; Hosseini et al., 2013; Wan et al., 2025), we introduce the following assumptions.

**Assumption 1** *(**Communication matrix**) The communication matrix $P \in \mathbb{R}^{n \times n}$ is supported on the graph $\mathcal{G} = ([n], E)$, symmetric, and doubly stochastic, which satisfies*

- $0 < P_{ij} < 1$ *only if* $(i, j) \in E$ *or* $i = j$;

- $\sum_{j=1}^{n} P_{ij} = \sum_{j \in \mathcal{N}_i} P_{ij} = 1, \forall i \in [n]$;

- $\sum_{i=1}^{n} P_{ij} = \sum_{i \in \mathcal{N}_j} P_{ij} = 1, \forall j \in [n]$,

*where $\mathcal{N}_i$ denotes the set including the immediate neighbors of the learner $i$ and itself. Moreover, $P$ is positive semi-definite, and its second largest singular value denoted by $\sigma_2(P)$ is strictly smaller than 1. We define $\rho = 1 - \sigma_2(P) \in (0, 1]$ and $\beta = \|I_n - P\| \in [0, 2]$.*

**Assumption 2** *(**Convexity**) The loss function $f_{t,i}(\cdot)$ of each learner $i \in [n]$ in every round $t \in [T]$ is convex over the feasible domain $\mathcal{X}$.*

**Assumption 3** *(**Strong convexity**) The loss function $f_{t,i}(\cdot)$ of each learner $i \in [n]$ in every round $t \in [T]$ is $\mu$-strongly convex over the domain $\mathcal{X}$, i.e., it holds that $f_{t,i}(\mathbf{y}) \geq f_{t,i}(\mathbf{x}) + \langle \nabla f_{t,i}(\mathbf{x}), \mathbf{y} - \mathbf{x} \rangle + \frac{\mu}{2} \|\mathbf{y} - \mathbf{x}\|^2$, for $\forall \mathbf{x}, \mathbf{y} \in \mathcal{X}$.*

**Assumption 4** *(**Bounded gradient**) The gradient of function $f_{t,i}(\cdot)$ of each learner $i \in [n]$ in every round $t \in [T]$ is bounded by $G$ over the domain $\mathcal{X}$, i.e., it holds that $\|\nabla f_{t,i}(\mathbf{x})\| \leq G$, for $\forall \mathbf{x} \in \mathcal{X}$.*

**Assumption 5** *(**Bounded domain**) The set $\mathcal{X}$ contains the origin $\mathbf{0}$, i.e., $\mathbf{0} \in \mathcal{X}$, and it is bounded by $D$, i.e., it holds that $\|\mathbf{x} - \mathbf{y}\| \leq D$, for $\forall \mathbf{x}, \mathbf{y} \in \mathcal{X}$.*

A *compressor* $\mathcal{C}(\cdot) : \mathbb{R}^d \to \mathbb{R}^d$ is a mapping whose output can be encoded with fewer bits than the original input. In this paper, we consider a broad class of compressors with the following general property (Koloskova et al., 2019).

**Definition 1** *(**Compressor**) A compression operator $\mathcal{C}(\cdot) : \mathbb{R}^d \to \mathbb{R}^d$ is termed an $\omega$-contractive compressor, if it satisfies*

$$\mathbb{E}_{\mathcal{C}} \left[ \|\mathcal{C}(\mathbf{x}) - \mathbf{x}\|^2 \right] \leq (1 - \omega) \|\mathbf{x}\|^2, \forall \mathbf{x} \in \mathbb{R}^d,$$

*for a parameter $\omega > 0$. Here, $\mathbb{E}_{\mathcal{C}}[\cdot]$ denotes the expectation over the internal randomness of $\mathcal{C}(\cdot)$.*

The compression error of the above compressor is $1 - \omega$. To mitigate this error, Huang et al. (2022) design the *repeated compressor*, as summarized in Algorithm 1. The core idea is to repeatedly apply the compressor for $L$ rounds and transmit the compressed data at each round, which involves $L$ rounds of communication. When $L = 1$, the repeated compressor degenerates to the standard compressor. We state the following lemma to provide the compression error of the repeated compressor.

**Lemma 1** *(**Repeated compressor**) (Lemma 2 in Huang et al. (2022)) Given a $\omega$-contractive compressor $\mathcal{C}(\cdot)$ and for any compression rounds $L \geq 1$, Algorithm 1 ensures*

$$\mathbb{E}_{\mathcal{C}} \left[ \|\mathcal{C}_L(\mathbf{x}) - \mathbf{x}\|^2 \right] \leq (1 - \omega)^L \|\mathbf{x}\|^2, \forall \mathbf{x} \in \mathbb{R}^d,$$

*where $\mathcal{C}_L(\mathbf{x}) = \mathbf{c}_L = \sum_{i=1}^{L} \Delta_i$ is the total output produced by Algorithm 1.*

---

**Algorithm 1** Repeated compressor $\mathcal{C}_L(\cdot)$

---

1: **Input:** compression round $L$, compressor $\mathcal{C}$, data $\mathbf{x}$
2: Initialize $\mathbf{c}_0 = \mathbf{0}$
3: **for** $i = 1$ to $L$ **do**
4:     Compute $\Delta_i = \mathcal{C}(\mathbf{x} - \mathbf{c}_{i-1})$ and send to neighbors
5:     Calculate $\mathbf{c}_i = \mathbf{c}_{i-1} + \Delta_i$
6: **end for**

---

**Remark 1** *Lemma 1 shows that the compression error of the repeated compressor decays exponentially with the compression rounds $L$, albeit at the cost of requiring $L$ communication rounds.*

A straightforward approach for decentralized optimization with compressed communication is to integrate a compressor into the standard gossip, where each learner transmits the compressed decision to its neighbors. However, this approach fails to converge to the average decision. To address this, Koloskova et al. (2019) adopt the difference compression technique (Tang et al., 2018) to develop Choco-gossip. Each learner $i$ maintains auxiliary variables $\hat{\mathbf{x}}_j(t) \in \mathbb{R}^d$ to record the data received from neighbors $j$, and $\hat{\mathbf{x}}_i(t) \in \mathbb{R}^d$ to track the data it has transmitted to neighbors over the past rounds. In each round $t$, learner $i$ updates its decision and auxiliary variables $\hat{\mathbf{x}}_j(t)$ as follows:

$$\mathbf{x}_i(t+1) = \mathbf{x}_i(t) + \gamma \sum_{j \in \mathcal{N}_i} P_{ij}(\hat{\mathbf{x}}_j(t) - \hat{\mathbf{x}}_i(t))$$
$$\hat{\mathbf{x}}_j(t+1) = \hat{\mathbf{x}}_j(t) + \mathcal{C}(\mathbf{x}_j(t+1) - \hat{\mathbf{x}}_j(t)), \forall j \in \mathcal{N}_i, \tag{2}$$

where $\gamma \leq 1$ is the consensus step size and $\mathcal{C}(\mathbf{x}_j(t+1) - \hat{\mathbf{x}}_j(t))$ is the received data from neighbor $j$. At each round $t$, each learner $i$ transmits $\mathcal{C}(\mathbf{x}_i(t+1) - \hat{\mathbf{x}}_i(t))$ to its neighbors $\mathcal{N}_i$. One might notice that Choco-gossip requires each learner to store $\deg(i) + 2$ variables, where $\deg(i)$ is the degree of learner $i$. This is not necessary and Koloskova et al. (2019) present an efficient version that only involves three additional variables. We provide more details in Appendix F.4.

In D-OCO with compressed communication, Tu et al. (2022) integrate D-OGD with Choco-gossip to develop a communication-efficient method, referred to as DC-DOGD. At each round $t$, each player $i$ plays a decision $\mathbf{x}_i(t)$ and suffers a loss $f_{t,i}(\mathbf{x}_i(t))$. Then learner $i$ updates its decision by leveraging both the local gradient and the information $\hat{\mathbf{x}}_j(t)$ received from its neighbors:

$$\mathbf{x}_i(t+1) = P_{\mathcal{X}}\big(\mathbf{x}_i(t) - \eta_t \nabla f_{t,i}(\mathbf{x}_i(t)) + \gamma \sum_{j \in \mathcal{N}_i} P_{ij}(\hat{\mathbf{x}}_j(t) - \hat{\mathbf{x}}_i(t))\big)$$
$$\hat{\mathbf{x}}_j(t+1) = \hat{\mathbf{x}}_j(t) + \mathcal{C}(\mathbf{x}_j(t+1) - \hat{\mathbf{x}}_j(t)), \forall j \in \mathcal{N}_i, \tag{3}$$

where $\eta_t$ is the learning rate and $P_{\mathcal{X}}(\cdot)$ is the projection onto the domain $\mathcal{X}$. Different from Koloskova et al. (2019), each learner is required to project its decision onto the feasible domain in D-OCO, which inevitably introduces an extra *projection error*.

## 3.2 OUR IMPROVED ALGORITHM

To begin with, we first briefly outline the key challenges in D-OCO with compressed communication and then present the corresponding techniques we develop to address them.

**Motivation.** The regret of DC-DOGD (Tu et al., 2022) can be decomposed into the regret of the averaged decision $\overline{\mathbf{x}}(t) = \frac{1}{n} \sum_{i=1}^{n} \mathbf{x}_i(t)$ and *the approximation error*, which consists of three components: (i) consensus error, arising from the network size and topology; (ii) compression error, introduced by the compressed communication; and (iii) projection error, caused by the projection operation in (3). To achieve tighter regret bounds, we focus on controlling the approximation error.

**Online compressed gossip strategy.** Since DC-DOGD performs a single gossip step per update, the decision of each learner converges to the average decision at a slow rate. To resolve this, we use the multi-step gossip to reduce the consensus and compression errors. More precisely, we have the following lemma to establish the convergence rate.

---

**Algorithm 2** Top-DOGD

---

1: **Input:** consensus step size $\gamma$, learning rate $\eta_b$, block size $L = L_1 + L_2$
2: Initialize $\mathbf{x}_i(1) = \mathbf{0}, \hat{\mathbf{x}}_i(1) = \mathbf{0}, \forall i \in [n]$
3: **for** block $b = 1$ to $T/L$ **do**
4:   **if** $b = 1$ **then**
5:     **for** $t = 1$ to $L$ **do**
6:       Play the decision $\mathbf{x}_i(1)$ and suffer the loss $f_{t,i}(\mathbf{x}_i(1))$
7:     **end for**
8:   **else**
9:     Set $\mathbf{y}_i^{(1)}(b) = \mathbf{x}_i(b) - \eta_b \mathbf{z}_i(b-1), \hat{\mathbf{y}}_i^{(1)}(b) = \hat{\mathbf{x}}_i(b), b_1 = 1$
10:     **for** $t = (b-1)L + 1$ to $(b-1)L + L_1$ **do**
11:       Play the decision $\mathbf{x}_i(b)$ and suffer the loss $f_{t,i}(\mathbf{x}_i(b))$
12:       Transmit $\mathcal{C}(\mathbf{y}_i^{(b_1)}(b) - \hat{\mathbf{y}}_i^{(b_1)}(b))$ to neighbors $j \in \mathcal{N}_i$
13:       Compute $\hat{\mathbf{y}}_j^{(b_1+1)}(b) = \hat{\mathbf{y}}_j^{(b_1)}(b) + \mathcal{C}(\mathbf{y}_j^{(b_1)}(b) - \hat{\mathbf{y}}_j^{(b_1)}(b))$ for $j \in \mathcal{N}_i$
14:       Compute $\mathbf{y}_i^{(b_1+1)}(b)$ according to (4) and set $b_1 = b_1 + 1$
15:     **end for** $\qquad\qquad\qquad\qquad\qquad\qquad\qquad$ ▷ online compressed gossip strategy
16:     Set $\mathbf{r}_i^{(1)}(b+1) = \mathbf{0}, \mathbf{r}_i(b+1) = P_{\mathcal{X}}(\mathbf{y}_i^{(L_1+1)}(b)) - \mathbf{y}_i^{(L_1+1)}(b), b_2 = 1$
17:     **for** $t = (b-1)L + L_1 + 1$ to $bL$ **do**
18:       Play the decision $\mathbf{x}_i(b)$ and suffer the loss $f_{t,i}(\mathbf{x}_i(b))$
19:       Transmit $\Delta_i^{(b_2)}(b) = \mathcal{C}(\mathbf{r}_i(b+1) - \mathbf{r}_i^{(b_2)}(b+1))$ and send $\Delta_i^{(b_2)}(b)$ to $j \in \mathcal{N}_i$
20:       Compute $\mathbf{r}_i^{(b_2+1)}(b+1) = \mathbf{r}_i^{(b_2)}(b+1) + \Delta_i^{(b_2)}(b)$ and set $b_2 = b_2 + 1$
21:     **end for** $\qquad\qquad\qquad\qquad\qquad\qquad\qquad$ ▷ projection error compensation scheme
22:     Update $\hat{\mathbf{x}}_j(b+1) = \hat{\mathbf{y}}_j^{(L_1+1)}(b) + \mathbf{r}_j^{(L_2+1)}(b+1)$ for $j \in \mathcal{N}_i$
23:     Compute $\mathbf{z}_i(b) = \sum_{t=(b-1)L+1}^{bL} \nabla f_{t,i}(\mathbf{x}_i(b))$ and update $\mathbf{x}_i(b+1) = P_{\mathcal{X}}(\mathbf{y}_i^{(L_1+1)}(b))$
24:   **end if**
25: **end for**

---

**Lemma 2** *(Theorem 2 in Koloskova et al. (2019)) We define $e_t = \mathbb{E}_{\mathcal{C}}[\sum_{i=1}^n \|\mathbf{x}_i(t) - \overline{\mathbf{x}}(t)\|^2 + \|\mathbf{x}_i(t) - \hat{\mathbf{x}}_i(t)\|^2]$ and $\overline{\mathbf{x}}(t) = \frac{1}{n}\sum_{i=1}^n \mathbf{x}_i(t)$. The first term $\sum_{i=1}^n \|\mathbf{x}_i(t) - \overline{\mathbf{x}}(t)\|^2$ characterizes the consensus error, and the second term $\sum_{i=1}^n \|\mathbf{x}_i(t) - \hat{\mathbf{x}}_i(t)\|^2$ characterizes the compression error. Given an $\omega$-contractive compressor $\mathcal{C}(\cdot)$, for any round $t$, by setting $\gamma = \frac{\rho\omega}{16\rho + \rho^2 + 4\beta^2 + 2\rho\beta^2 - 8\rho\omega}$ and performing the update in (2) for $L_1$ rounds, we can ensure*

$$e_{t+L_1} \leq (1 - \frac{\rho^2 \omega}{82})^{L_1} e_t.$$

As can be observed, the errors decrease at an exponential rate as the number of gossip rounds increases. While repeatedly executing the gossip step can mitigate errors, this results in multiple communication rounds per update, which substantially exacerbates the communication burden we aim to alleviate. Motivated by Wan et al. (2024; 2025), we integrate the blocking update mechanism with Choco-gossip to design an online compressed gossip strategy. If we partition the total rounds into blocks and update once at the end of each block, the communications can be distributed across rounds. With an appropriate block size, this allows us to control the errors while keeping only one communication per round. Nevertheless, this mechanism alone is insufficient, as the projection step in D-OCO introduces an additional error.

**Projection error compensation scheme.** The projection operation in (3) introduces an extra error, which induces an $O(n)$ dependence in the bound of the approximation error. By carefully analyzing the error, we find that if each learner $i$ were able to add the projection error of neighbor $j \in \mathcal{N}_i$ to the auxiliary variable $\hat{\mathbf{x}}_j(t)$, the $O(n)$ dependence could be avoided. However, each learner only broadcasts the compressed data, which leads to an $O(1 - \omega)$ bias. Notably, if the projection error can be constrained in the order of $O(1/n)$, the upper bound becomes independent of $n$. Drawing inspiration from Huang et al. (2022), we employ the repeated compressor. By recursively applying a compressor over $L_2 = \lceil \ln(8n)/\omega \rceil$ rounds, we can ensure the compression error satisfies $\mathbb{E}_{\mathcal{C}}\left[\|\mathcal{C}_{L_2}(\mathbf{x}) - \mathbf{x}\|^2\right] \leq (1 - \omega)^{L_2} \|\mathbf{x}\|^2 \leq \frac{1}{8n} \|\mathbf{x}\|^2$. However, a direct application of this technique

incurs $L_2$ communication rounds per update. To overcome this dilemma, we again utilize the blocking update mechanism to distribute the communications into each round per block.

**Overall algorithm: a two-level blocking update structure.** To unify the two strategies within a single framework, we propose a *two-level blocking update framework*. Concretely, we partition the $T$ rounds into several blocks with block size $L = L_1 + L_2$. We maintain the same decision $\mathbf{x}_i(b)$ for each learner $i$ in block $b$ (we assume $T/L$ is an integer without loss of generality) and only update the decision at the end of each block. In block $b$, we first apply our online compressed gossip strategy for $L_1$ rounds and then use the projection error compensation scheme for $L_2$ rounds.

We present our Top-DOGD in Algorithm 2. For each learner $i$, we first initialize the decision $\mathbf{x}_i(1) = \mathbf{0} \in \mathbb{R}^d$ and the local replica $\hat{\mathbf{x}}_j(1) = \mathbf{0} \in \mathbb{R}^d$ to store the information from its neighbors $j \in \mathcal{N}_i$. In each block $b \geq 2$, we start with updating the surrogate decision $\mathbf{y}_i^{(1)}(b) = \mathbf{x}_i(b) - \eta_b \mathbf{z}_i(b-1)$, where $\mathbf{z}_i(b-1) = \sum_{t=(b-2)L+1}^{(b-1)L} \nabla f_{t,i}(\mathbf{x}_i(b-1))$ is the sum of gradients in block $b-1$, and set the local auxiliary variable $\hat{\mathbf{y}}_i^{(1)}(b) = \hat{\mathbf{x}}_i(b)$. Next, we perform our online compressed gossip strategy for $L_1$ rounds (Lines 5–10). For $b_1 \in [1, L_1]$, learner $i$ transmits $\mathcal{C}(\mathbf{y}_i^{(b_1)}(b) - \hat{\mathbf{y}}_i^{(b_1)}(b))$ to neighbor $j \in \mathcal{N}_i$. After receiving the information from its neighbors, learner $i$ updates $\hat{\mathbf{y}}_j^{(b_1+1)}(b)$ and then computes

$$\mathbf{y}_i^{(b_1+1)}(b) = \mathbf{y}_i^{(b_1)}(b) + \gamma \sum_{j \in \mathcal{N}_i} P_{ij}(\hat{\mathbf{y}}_j^{(b_1+1)}(b) - \hat{\mathbf{y}}_i^{(b_1)}(b)). \tag{4}$$

Within the second sub-block, we apply our projection error compensation scheme (Lines 12–15). Each learner $i$ recursively compresses the residual of the projection error $\mathbf{r}_i(b+1) = \mathbf{x}_i(b+1) - \mathbf{y}_i^{(L_1+1)}(b)$ over $L_2$ rounds and sends compressed data to its neighbors per round. At the end of the block $b$, the learner $i$ updates its decision $\mathbf{x}_i(b+1) = P_{\mathcal{X}}(\mathbf{y}_i^{(L_1+1)}(b))$. In the following, we establish the theoretical guarantees of Top-DOGD for convex and strongly convex loss functions, respectively.

**Theorem 1** *Let* $L_1 = \lceil \frac{28 \ln n}{\gamma \rho} \rceil, L_2 = \lceil \frac{\ln(8n)}{\omega} \rceil, L = L_1 + L_2 = O(\omega^{-1} \rho^{-2} \ln n), \eta_b = \eta = \frac{D}{G\sqrt{LT}},$ $\gamma = \frac{\omega \rho}{2\rho \beta^2 + 4\beta^2 + (2-\omega)(\beta^2 + 2\beta)\rho + \rho^2}$. *Under Assumptions 1, 2, 4 and 5, for any* $i \in [n]$ *and convex loss functions, Algorithm 2 ensures*

$$\mathbb{E}_{\mathcal{C}}[R(T,i)] \leq O(n\sqrt{LT}) = O(\omega^{-1/2} \rho^{-1} n \sqrt{\ln n} \sqrt{T}).$$

**Theorem 2** *Let* $L_1 = \lceil \frac{28 \ln n}{\gamma \rho} \rceil, L_2 = \lceil \frac{\ln(8n)}{\omega} \rceil, L = L_1 + L_2 = O(\omega^{-1} \rho^{-2} \ln n), \eta_b = \frac{1}{\mu(bL+8)},$ $\gamma = \frac{\omega \rho}{2\rho \beta^2 + 4\beta^2 + (2-\omega)(\beta^2 + 2\beta)\rho + \rho^2}$. *Under Assumptions 1, 3, 4 and 5, for any* $i \in [n]$ *and* $\mu$-*strongly convex loss functions, Algorithm 2 ensures*

$$\mathbb{E}_{\mathcal{C}}[R(T,i)] \leq O(nL \ln T) = O(\omega^{-1} \rho^{-2} n \ln n \ln T).$$

**Remark 2** *Compared to the previous regret bounds of* $O(\max\{\omega^{-2}\rho^{-4}n^{1/2}, \omega^{-4}\rho^{-8}\}n\sqrt{T})$ *and* $O(\max\{\omega^{-2}\rho^{-4}n^{1/2}, \omega^{-4}\rho^{-8}\}n \ln T)$ *(Tu et al., 2022), our method achieves tighter dependence on* $\omega$, $\rho$ *and* $n$. *This enhancement is particularly critical in large-scale communication environments.*

**Additional discussion.** Our refined bounds result from the coordinated use of the two strategies, as neither alone is sufficient to achieve the desired improvement. To highlight their significance, we conduct an ablation analysis by considering two scenarios: (i) performing the online compressed gossip strategy with $L_1 = 1$, and (ii) removing the projection error compensation scheme ($L_2 = 0$). First, when $L_1 = 1$, our method becomes a combination of DC-DOGD (Tu et al., 2022) with our projection error compensation scheme, which does not improve the regret bounds of Tu et al. (2022). Although we can mitigate the projection error, we cannot reduce the consensus error and compression error as we desire. If $L_2 = 0$, we suffer the projection error in each round. We can only obtain $O(\omega^{-1/2} \rho^{-1} n^{5/4} \sqrt{\ln n} \sqrt{T})$ and $O(\omega^{-1} \rho^{-2} n^{3/2} \ln n \ln T)$ regret bounds for convex and strongly convex loss functions. While the dependence on $\omega$ and $\rho$ is still tighter than the regret bounds of Tu et al. (2022), the dependence on $n$ is worse than the regret bounds of Top-DOGD.

### 3.3 LOWER BOUNDS

In this section, we present lower bounds for convex and strongly convex loss functions in D-OCO with compressed communication. In D-OCO, Wan et al. (2025) have derived the lower bounds of

$\Omega(\rho^{-1/4}n\sqrt{T})$ and $\Omega(\rho^{-1/2}n\ln T)$ for convex and strongly convex losses. Their analysis leverages the 1-connected cycle graph (Duchi et al., 2011), where the adversary can force at least one learner to suffer $\lceil n/4 \rceil$ rounds of communication delay before receiving the information of the global function $f_t(\mathbf{x})$. By leveraging this topology, they establish the aforementioned lower bounds.

The existing literature lacks lower bounds that explicitly characterize the dependence on the compression ratio $\omega$. To fill this, we model the compression effect by adopting the randomized gossip compressor $\mathcal{C}(\cdot) : \mathbb{R}^d \to \mathbb{R}^d$, which outputs $\mathcal{C}(\mathbf{x}) = \mathbf{x}$ with probability $\omega$ and $\mathcal{C}(\mathbf{x}) = \mathbf{0}$ otherwise. Under this scheme, two connected learners $i$ and $j$ can successfully exchange data only with probability $\omega$ in each round. Consequently, the expected number of rounds required for a successful transmission is $\lceil 1/\omega \rceil$. Building on this construction, we establish the following lower bounds.

**Theorem 3** *Given the feasible domain $\mathcal{X} = [\frac{-D}{2\sqrt{d}}, \frac{D}{2\sqrt{d}}]^d$ and $n = 2m + 2$ for some positive integer $m$. For any D-OCO algorithm, if $n \leq 8\omega T + 8\omega$, there exists a sequence of convex loss functions satisfying Assumption 4, a graph $\mathcal{G} = ([n], E)$, a compressor satisfying Definition 1, and a matrix $P$ satisfying Assumption 1 such that*

$$\mathbb{E}_{\mathcal{C}}\left[R(T, 1)\right] \geq \frac{nGD(\pi T)^{1/2}}{2^5 \rho^{1/4} \omega^{1/2}}.$$

**Theorem 4** *Given the feasible domain $\mathcal{X} = [0, \frac{D}{2\sqrt{d}}]^d$ and $n = 2m + 2$ for some positive integer $m$. For any D-OCO algorithm, if $16n + \omega \leq \omega T$, there exists a sequence of $\mu$-strongly convex loss functions satisfying Assumption 4 with $G = \mu D$, a graph $\mathcal{G} = ([n], E)$, a compressor satisfying Definition 1, and a matrix $P$ satisfying Assumption 1 such that*

$$\mathbb{E}_{\mathcal{C}}\left[R(T, 1)\right] \geq \frac{(\log_{16}(30\omega(T - 1)/n) - 2)(n - 2)\pi\mu D^2}{2^{22}\omega\rho^{1/2}}.$$

**Remark 3** *We have established the $\Omega(\omega^{-1/2}\rho^{-1/4}n\sqrt{T})$ and $\Omega(\omega^{-1}\rho^{-1/2}n\ln T)$ lower bounds for convex and strongly convex loss functions, which match the corresponding upper bounds up to $\rho$ and polylogarithmic factors in $n$.*

## 4 EXTENSION TO BANDIT FEEDBACK SETTING

In this section, we extend our method into the bandit feedback setting by employing classical gradient estimators, with more details provided in Appendix C. Following prior work (Flaxman et al., 2005; Agarwal et al., 2010), we present an assumption specific to the bandit feedback setting.

**Assumption 6** *(**Bounded domain**) The convex set $\mathcal{X}$ contains the ball with radius $r$, and is contained in the ball with radius $R$, i.e., it holds that $r\mathcal{B} \subseteq \mathcal{X} \subseteq R\mathcal{B}, \mathcal{B} = \{\mathbf{u} \in \mathbb{R}^d : \|\mathbf{u}\| \leq 1\}$.*

### 4.1 ONE-POINT BANDIT FEEDBACK SETTING

We first consider the one-point bandit feedback setting. The key challenge in this setting is the lack of gradients. To overcome this, we adopt the one-point gradient estimator (Flaxman et al., 2005)

$$\hat{\mathbf{g}}_{t,i} = \frac{d}{\epsilon} f_{t,i}(\mathbf{x}_i(t))\mathbf{u}_{t,i}, \tag{5}$$

where $\mathbf{x}_i(t) = \mathbf{x}_i(b) + \epsilon\mathbf{u}_{t,i}, \epsilon \in (0, 1)$ and $\mathbf{u}_{t,i}$ is *uniformly* sampled from $\mathcal{B} = \{\mathbf{u} \in \mathbb{R}^d | \|\mathbf{u}\| \leq 1\}$ for $t \in [(b-1)L + 1, bL]$. This estimator is an unbiased estimator of the gradient, i.e., $\mathbb{E}[\hat{\mathbf{g}}_{t,i}] = \nabla f_{t,i}(\mathbf{x}_i(t))$. We integrate it with Top-DOGD to develop Two-level Compressed Decentralized Online Bandit Descent with One-point Feedback (Top-DOBD-1). In each round $t \in [(b-1)L+1, bL]$, the learner plays the decision $\mathbf{x}_i(t) = \mathbf{x}_i(b) + \epsilon\mathbf{u}_{t,i}$. There are two modifications: (i) replacing the gradient $\nabla f_{t,i}(\mathbf{x}_i(t))$ with the one-point gradient estimator, and (ii) projecting onto the domain $(1 - \zeta)\mathcal{X}$, where $\zeta \in (0, 1)$ is the shrinkage size. Then, we present the regret bounds of our method.

**Theorem 5** *Let $L_1 = \lceil \frac{28 \ln n}{\gamma\rho} \rceil, L_2 = \lceil \frac{\ln(8n)}{\omega} \rceil, L = L_1 + L_2, \eta_b = \eta = \frac{R\epsilon}{d\sqrt{LT}}, \gamma = \frac{\omega\rho}{2\rho\beta^2 + 4\beta^2 + (2-\omega)(\beta^2 + 2\beta)\rho + \rho^2}, \zeta = \frac{\epsilon}{r}, \epsilon = cd^{1/2}L^{1/4}T^{-1/4}$, where $c$ is a constant such that $\epsilon \leq r$.*

*Under Assumptions 1, 2, 4 and 6, for any $i \in [n]$ and convex loss functions, Top-DOBD-1 ensures*

$$\mathbb{E}_{\mathcal{C}}\left[R(T, i)\right] \leq O(\omega^{-1/4}\rho^{-1/2}d^{1/2}n(\ln n)^{1/4}T^{3/4}).$$

**Theorem 6** *Let* $L_1 = \lceil \frac{28\ln n}{\gamma\rho} \rceil, L_2 = \lceil \frac{\ln(8n)}{\omega} \rceil, L = L_1 + L_2, \eta_b = \frac{1}{\mu(bL+8)}, \gamma = \frac{\omega\rho}{2\rho\beta^2+4\beta^2+(2-\omega)(\beta^2+2\beta)\rho+\rho^2}, \zeta = \frac{\epsilon}{r}, \epsilon = cd^{2/3}L^{1/3}(\frac{\ln(T+8)}{T})^{1/3},$ *where c is a constant such that* $\epsilon \leq r$. *Under Assumptions 1, 3, 4 and 6, for any* $i \in [n]$ *and* $\mu$-*strongly convex loss functions, Top-DOBD-1 ensures*

$$\mathbb{E}_{\mathcal{C}}\left[R(T, i)\right] \leq O(\omega^{-1/3}\rho^{-2/3}d^{2/3}n(\ln n)^{1/3}T^{2/3}(\ln T)^{1/3}).$$

**Remark 4** *In contrast to the regret bounds of* $O(\max\{\omega^{-1}\rho^{-2}n^{1/4}, \omega^{-2}\rho^{-4}\}d^{1/2}nT^{3/4})$ *and* $O(\max\{\omega^{-2/3}\rho^{-4/3}n^{1/6}, \omega^{-4/3}\rho^{-8/3}\}d^{2/3}nT^{2/3}(\ln T)^{1/3})$ *for convex and strongly convex loss functions of Tu et al. (2022), our methods achieve tighter dependence on* $\omega$, $\rho$, *and* $n$.

## 4.2 Two-point Bandit Feedback Setting

In the two-point bandit feedback setting (Agarwal et al., 2010), since the learner $i$ can have access to two loss values $f_{t,i}(\mathbf{x}_{i,1}(t))$ and $f_{t,i}(\mathbf{x}_{i,2}(t))$ in each round, the regret is redefined as

$$R_2(T, i) = \sum_{t=1}^{T}\sum_{j=1}^{n} \frac{f_{t,j}(\mathbf{x}_{i,1}(t)) + f_{t,j}(\mathbf{x}_{i,2}(t))}{2} - \min_{\mathbf{x}\in\mathcal{X}} \sum_{t=1}^{T}\sum_{j=1}^{n} f_{t,j}(\mathbf{x}).$$

For $t \in [(b-1)L+1, bL]$, each player $i$ plays the decision $\mathbf{x}_{i,1}(t) = \mathbf{x}_i(b) + \epsilon\mathbf{u}_{t,i}$ and $\mathbf{x}_{i,2}(t) = \mathbf{x}_i(b) - \epsilon\mathbf{u}_{t,i}$ and suffers two losses. We construct the gradient as

$$\hat{\mathbf{g}}_{t,i} = \frac{d}{2\epsilon}\left(f_{t,i}(\mathbf{x}_{i,1}(t)) - f_{t,i}(\mathbf{x}_{i,2}(t))\right)\mathbf{u}_{t,i}. \tag{6}$$

We replace the gradient with the two-point gradient estimator (Agarwal et al., 2010) to develop our method, named as Top-DOBD-2. In the following, we establish its regret bounds.

**Theorem 7** *Let* $\eta_b = \eta = \frac{R}{dG\sqrt{LT}}, L_1 = \lceil \frac{28\ln n}{\gamma\rho} \rceil, L_2 = \lceil \frac{\ln(8n)}{\omega} \rceil, L = L_1 + L_2, \gamma = \frac{\omega\rho}{2\rho\beta^2+4\beta^2+(2-\omega)(\beta^2+2\beta)\rho+\rho^2}, \zeta = \frac{\epsilon}{r}, \epsilon = cT^{-1/2},$ *where c is a constant such that* $\epsilon \leq r$. *Under Assumptions 1, 2, 4 and 6, for any* $i \in [n]$ *and convex loss functions, Top-DOBD-2 ensures*

$$\mathbb{E}_{\mathcal{C}}\left[R_2(T, i)\right] \leq O(\omega^{-1/2}\rho^{-1}dn\sqrt{\ln n}\sqrt{T}).$$

**Theorem 8** *Let* $\eta_b = \frac{1}{\mu(bL+8)}, L_1 = \lceil \frac{28\ln n}{\gamma\rho} \rceil, L_2 = \lceil \frac{\ln(8n)}{\omega} \rceil, L = L_1 + L_2, \gamma = \frac{\omega\rho}{2\rho\beta^2+4\beta^2+(2-\omega)(\beta^2+2\beta)\rho+\rho^2}, \zeta = \frac{\epsilon}{r}, \epsilon = \frac{c\ln T}{T},$ *where c is a constant such that* $\epsilon \leq r$. *Under Assumptions 1, 3, 4 and 6, for any* $i \in [n]$ *and* $\mu$-*strongly convex loss functions, Top-DOBD-2 ensures*

$$\mathbb{E}_{\mathcal{C}}\left[R_2(T, i)\right] \leq O(\omega^{-1}\rho^{-2}d^2n\ln n\ln T).$$

**Remark 5** *Compared to the previous regret bounds of* $O(\max\{\omega^{-2}\rho^{-4}n^{1/2}, \omega^{-4}\rho^{-8}\}dn\sqrt{T})$ *and* $O(\max\{\omega^{-2}\rho^{-4}n^{1/2}, \omega^{-4}\rho^{-8}\}d^2n\ln T)$ *(Tu et al., 2022), our method again achieves tighter dependence on* $\omega$, $\rho$ *and* $n$.

## 5 Conclusion

In this paper, we investigate decentralized online convex optimization with compressed communication. First, we introduce a novel method, named Top-DOGD, achieving better regret bounds of $\tilde{O}(\omega^{-1/2}\rho^{-1}n\sqrt{T})$ and $\tilde{O}(\omega^{-1}\rho^{-2}n\ln T)$ for convex and strongly convex loss functions. Furthermore, we demonstrate their near-optimality by establishing the $\Omega(\omega^{-1/2}\rho^{-1/4}n\sqrt{T})$ and $\Omega(\omega^{-1}\rho^{-1/2}n\ln T)$ lower bounds for convex and strongly convex loss functions, respectively. Additionally, we consider the bandit feedback setting and extend Top-DOGD by utilizing the classic gradient estimators. Our proposed algorithms improve the dependence on the compression ratio $\omega$, number of learners $n$ and the spectral gap of the communication matrix $\rho$ under both the one-point and two-point bandit feedback settings.

REPRODUCIBILITY STATEMENT

We provide clear explanations of all assumptions and include complete proofs of our theoretical claims in the appendix. For the experimental results, we specify the dataset, baseline methods, and hyperparameter choices, and we will release the code to ensure full reproducibility when this paper is published.

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

## THE USE OF LLMS

We used large language models (LLMs) solely for minor language polishing of the manuscript. The LLMs did not contribute to research ideation, algorithm design, theoretical analysis, or experimental work. Their role was strictly limited to assisting with improving readability and grammar.

## A    ADDITIONAL DISCUSSION ON RELATED WORK

### A.1    COMPRESSED COMMUNICATION

In order to reduce the volume of data exchanged between learners, several works attempt to transmit the compressed information $\mathcal{C}(\mathbf{x})$ instead of broadcasting the full vector $\mathbf{x}$, where $\mathbf{x} \in \mathbb{R}^d, \mathcal{C}(\cdot) : \mathbb{R}^d \to \mathbb{R}^d$ is an operator chosen such that $\mathcal{C}(\mathbf{x})$ can be more efficiently represented. The mainstream communication compression techniques can be summarized in two classes: unbiased compressor (Jiang & Agrawal, 2018; Tang et al., 2018; Zhang et al., 2017a) and biased (contractive) compressor (Seide et al., 2014; Wangni et al., 2018; Stich et al., 2018). More discussions of compressors can be found in Richtárik et al. (2022) and Beznosikov et al. (2023).

An unbiased compressor outputs $\mathcal{C}(\mathbf{x})$ such that $\mathbb{E}_{\mathcal{C}}[\mathcal{C}(\mathbf{x})] = \mathbf{x}$ for any input $\mathbf{x} \in \mathbb{R}^d$. Quantization, as a typical unbiased compression technique, represents 32-bit data using fewer bits. In contrast, a contractive compressor yields a biased vector with smaller variance. One popular approach is sparsification, which constructs a sparse vector by selecting a subset of the entries. Wangni et al. (2018) and Stich et al. (2018) reduce communication costs by transmitting only a few entries of $\mathbf{x}$, selected either at random or by choosing those with the largest values. To further reduce the compression error of the compressor, Huang et al. (2022) introduce the fast compressor (repeated compressor). The core idea is to compress information for $L$ rounds and communicate in each round, which reduces the compression error of compressor exponentially with the number of compression rounds $L$. While using an unbiased compressor may achieve better theoretical guarantees, contractive compressors can offer comparable and even superior empirical performance under weaker assumptions. Following Koloskova et al. (2019), we do not distinguish these two approaches, and refer to both of them as *compression operators* in this paper.

### A.2    DIFFERENCE COMPRESSION AND ERROR FEEDBACK

Since a direct combination of compressor and the standard gossip fails to converge to the correct solution. Tang et al. (2018) propose difference compression (DC) is a popular compression scheme and analyze under the unbiased compressor. DC adds replicas of neighboring states of each learner and transmits the compressed state-difference information. Later, Koloskova et al. (2019) give the analysis for the biased compressor.

Error feedback (EF) (Seide et al., 2014; Ström, 2015; Karimireddy et al., 2019) is another common compression scheme, aiming to correct errors introduced by the compressor. Specifically, DC focuses on the discrepancy between the current decision and its replica, which is widely used in distributed optimization because the exchanged state variables typically converge to a nonzero limit. In contrast, EF compresses the sum of the local gradient and an accumulated residual error, popular in federated learning problems where the exchanged gradient information is expected to vanish asymptotically.

## B    EXAMPLES OF COMPRESSOR

In this section, we present some examples of compressor.

- *Sparsification.* Randomly selecting $k$ out of $d$ coordinates (Rand-$k$), or selecting the $k$ coordinates with the largest absolute values (Top-$k$), both yield compressors with a compression ratio of $\omega = \frac{k}{d}$.

- *Randomized gossip.* Outputting $\mathcal{C}(\mathbf{x}) = \mathbf{x}$ with probability $p \in (0, 1]$ and $\mathcal{C}(\mathbf{x}) = 0$ otherwise leads to a compression ratio of $\omega = p$.

---

**Algorithm 3** Top-DOBD-1

---

1: **Input:** consensus step size $\gamma$, learning rate $\eta_b$, block size $L = L_1 + L_2$, shrinkage size $\xi$, exploration size $\epsilon$
2: Initialize $\mathbf{x}_i(1) = \mathbf{0}, \hat{\mathbf{x}}_i(1) = \mathbf{0}, \forall i \in [n]$
3: **for** block $b = 1$ to $T/L$ **do**
4:     **if** $b = 1$ **then**
5:         **for** $t = 1$ to $L$ **do**
6:             Play the decision $\mathbf{x}_i(t) = \mathbf{x}_i(1) + \epsilon\mathbf{u}_{t,i}$
7:             Suffer the loss $f_{t,i}(\mathbf{x}_i(t))$
8:             Construct the gradient $\hat{\mathbf{g}}_{t,i} = \frac{d}{\epsilon} f_{t,i}(\mathbf{x}_i(t) = \mathbf{x}_i(1) + \epsilon\mathbf{u}_{t,i})\mathbf{u}_{t,i}$
9:         **end for**
10:     **else**
11:         Set $\mathbf{y}_i^{(1)}(b) = \mathbf{x}_i(b) - \eta_b\mathbf{z}_i(b-1), \hat{\mathbf{y}}_i^{(1)}(b) = \hat{\mathbf{x}}_i(b), b_1 = 1$
12:         **for** $t = (b-1)L + 1$ to $(b-1)L + L_1$ **do**
13:             Play the decision $\mathbf{x}_i(t) = \mathbf{x}_i(b) + \epsilon\mathbf{u}_{t,i}$
14:             Suffer the loss $f_{t,i}(\mathbf{x}_i(t))$
15:             Construct the gradient $\hat{\mathbf{g}}_{t,i} = \frac{d}{\epsilon} f_{t,i}(\mathbf{x}_i(t))\mathbf{u}_{t,i}$
16:             Transmit $\mathcal{C}(\mathbf{y}_i^{(b_1)}(b) - \hat{\mathbf{y}}_i^{(b_1)}(b))$ to neighbors $j \in \mathcal{N}_i$
17:             Compute $\hat{\mathbf{y}}_j^{(b_1+1)}(b) = \hat{\mathbf{y}}_j^{(b_1)}(b) + \mathcal{C}(\mathbf{y}_j^{(b_1)}(b) - \hat{\mathbf{y}}_j^{(b_1)}(b))$ for $j \in \mathcal{N}_i$
18:             Compute $\mathbf{y}_i^{(b_1+1)}(b) = \mathbf{y}_i^{(b_1)}(b) + \gamma \sum_{j\in\mathcal{N}_i} P_{ij}(\hat{\mathbf{y}}_j^{(b_1+1)}(b) - \hat{\mathbf{y}}_i^{(b_1)}(b)), b_1 = b_1 + 1$
19:         **end for**               ▷ online compressed gossip strategy
20:         Set $\mathbf{r}_i^{(1)}(b+1) = \mathbf{0}, \mathbf{r}_i(b+1) = P_{(1-\varsigma)\mathcal{X}}(\mathbf{y}_i^{(L_1+1)}(b)) - \mathbf{y}_i^{(L_1+1)}(b), b_2 = 1$
21:         **for** $t = (b-1)L + L_1 + 1$ to $bL$ **do**
22:             Play the decision $\mathbf{x}_i(t) = \mathbf{x}_i(b) + \epsilon\mathbf{u}_{t,i}$
23:             Suffer the loss $f_{t,i}(\mathbf{x}_i(t))$
24:             Construct the gradient $\hat{\mathbf{g}}_{t,i} = \frac{d}{\epsilon} f_{t,i}(\mathbf{x}_i(b) + \epsilon\mathbf{u}_{t,i})\mathbf{u}_{t,i}$
25:             Transmit $\Delta_i^{(b_2)}(b) = \mathcal{C}(\mathbf{r}_i(b+1) - \mathbf{r}_i^{(b_2)}(b+1))$ and send $\Delta_i^{(b_2)}(b)$ to $j \in \mathcal{N}_i$
26:             Compute $\mathbf{r}_i^{(b_2+1)}(b+1) = \mathbf{r}_i^{(b_2)}(b+1) + \Delta_i^{(b_2)}(b)$ and set $b_2 = b_2 + 1$
27:         **end for**               ▷ projection error compensation scheme
28:         Update $\hat{\mathbf{x}}_j(b+1) = \hat{\mathbf{y}}_j^{(L_1+1)}(b) + \mathbf{r}_j^{(L_2+1)}(b+1)$ for $j \in \mathcal{N}_i$
29:         Compute $\mathbf{z}_i(b) = \sum_{t=(b-1)L+1}^{bL} \hat{\mathbf{g}}_{t,i}$ and update $\mathbf{x}_i(b+1) = P_{(1-\varsigma)\mathcal{X}}(\mathbf{y}_i^{(L_1+1)}(b))$
30:     **end if**
31: **end for**

---

- *Rescaled unbiased estimators.* Suppose $E_\mathcal{C}[\mathbf{x}] = \mathbf{x}, \mathbb{E}_\mathcal{C}\left[\|\mathcal{C}(\mathbf{x})\|^2\right] \leq \tau\|\mathbf{x}\|^2$, then $\mathcal{C}'(\mathbf{x}) = \frac{1}{\tau}\mathcal{C}(\mathbf{x})$ is a compressor with $\omega = \frac{1}{\tau}$.

## C   EXTENSION TO BANDIT FEEDBACK SETTING

In this section, we summarize our algorithms for bandit feedback setting. Top-DOBD-1 for the one-point bandit feedback setting is presented in Algorithm 3, and Top-DOBD-2 for the two-point bandit feedback setting is shown in Algorithm 4.

### C.1   ONE-POINT BANDIT FEEDBACK SETTING

We first consider the one-point bandit feedback setting, where each learner only has access to the loss value instead of the gradient of the loss function. We utilize the one-point gradient estimator (Flaxman et al., 2005)

$$\hat{\mathbf{g}}_{t,i} = \frac{d}{\epsilon} f_{t,i}(\mathbf{x}_i(t))\mathbf{u}_{t,i},$$

where $\mathbf{x}_i(t) = \mathbf{x}_i(b) + \epsilon\mathbf{u}_{t,i}$ $\epsilon \in (0,1)$ and $\mathbf{u}_{t,i}$ is *uniformly* sampled from $\mathcal{B} = \{\mathbf{u} \in \mathbb{R}^d | \|\mathbf{u}\| \leq 1\}$ for $t \in [(b-1)L + 1, bL]$. This estimator unbiasedly approximates the gradient, i.e., $\mathbb{E}[\hat{\mathbf{g}}_{t,i}] =$

---

**Algorithm 4** Top-DOBD-2

---

1: **Input:** consensus step size $\gamma$, learning rate $\eta_b$, block size $L = L_1 + L_2$, shrinkage size $\xi$, exploration size $\epsilon$
2: Initialize $\mathbf{x}_i(1) = \mathbf{0}, \hat{\mathbf{x}}_i(1) = \mathbf{0}, \forall i \in [n]$
3: **for** block $b = 1$ to $T/L$ **do**
4:    **if** $b = 1$ **then**
5:       **for** $t = 1$ to $L$ **do**
6:          Play the decisions $\mathbf{x}_{i,1}(t) = \mathbf{x}_i(1) + \epsilon\mathbf{u}_{t,i}$ and $\mathbf{x}_{i,2}(t) = \mathbf{x}_i(1) - \epsilon\mathbf{u}_{t,i}$
7:          Suffer the loss $f_{t,i}(\mathbf{x}_{i,1}(t))$ and $f_{t,i}(\mathbf{x}_{i,2}(t))$
8:          Construct the gradient $\hat{\mathbf{g}}_{t,i} = \frac{d}{2\epsilon}\left(f_{t,i}(\mathbf{x}_{i,1}(t)) - f_{t,i}(\mathbf{x}_{i,2}(t))\right)\mathbf{u}_{t,i}$
9:       **end for**
10:    **else**
11:       Set $\mathbf{y}_i^{(1)}(b) = \mathbf{x}_i(b) - \eta_b\mathbf{z}_i(b-1), \hat{\mathbf{y}}_i^{(1)}(b) = \hat{\mathbf{x}}_i(b), b_1 = 1$
12:       **for** $t = (b-1)L + 1$ to $(b-1)L + L_1$ **do**
13:          Play the decisions $\mathbf{x}_{i,1}(t) = \mathbf{x}_i(b) + \epsilon\mathbf{u}_{t,i}$ and $\mathbf{x}_{i,2}(t) = \mathbf{x}_i(b) - \epsilon\mathbf{u}_{t,i}$
14:          Suffer the loss $f_{t,i}(\mathbf{x}_{i,1}(t))$ and $f_{t,i}(\mathbf{x}_{i,2}(t))$
15:          Construct the gradient $\hat{\mathbf{g}}_{t,i} = \frac{d}{2\epsilon}\left(f_{t,i}(\mathbf{x}_{i,1}(t)) - f_{t,i}(\mathbf{x}_{i,2}(t))\right)\mathbf{u}_{t,i}$
16:          Transmit $\mathcal{C}(\mathbf{y}_i^{(b_1)}(b) - \hat{\mathbf{y}}_i^{(b_1)}(b))$ to neighbors $j \in \mathcal{N}_i$
17:          Compute $\hat{\mathbf{y}}_j^{(b_1+1)}(b) = \hat{\mathbf{y}}_j^{(b_1)}(b) + \mathcal{C}(\mathbf{y}_j^{(b_1)}(b) - \hat{\mathbf{y}}_j^{(b_1)}(b))$ for $j \in \mathcal{N}_i$
18:          Compute $\mathbf{y}_i^{(b_1+1)}(b) = \mathbf{y}_i^{(b_1)}(b) + \gamma\sum_{j\in\mathcal{N}_i} P_{ij}(\hat{\mathbf{y}}_j^{(b_1+1)}(b) - \hat{\mathbf{y}}_j^{(b_1)}(b)), b_1 = b_1 + 1$
19:       **end for**                     ▷ online compressed gossip strategy
20:       Set $\mathbf{r}_i^{(1)}(b+1) = \mathbf{0}, \mathbf{r}_i(b+1) = P_{(1-\zeta)\mathcal{X}}(\mathbf{y}_i^{(L_1+1)}(b)) - \mathbf{y}_i^{(L_1+1)}(b), b_2 = 1$
21:       **for** $t = (b-1)L + L_1 + 1$ to $bL$ **do**
22:          Play the decisions $\mathbf{x}_{i,1}(t) = \mathbf{x}_i(b) + \epsilon\mathbf{u}_{t,i}$ and $\mathbf{x}_{i,2}(t) = \mathbf{x}_i(b) - \epsilon\mathbf{u}_{t,i}$
23:          Suffer the loss $f_{t,i}(\mathbf{x}_{i,1}(t))$ and $f_{t,i}(\mathbf{x}_{i,2}(t))$
24:          Construct the gradient $\hat{\mathbf{g}}_{t,i} = \frac{d}{2\epsilon}\left(f_{t,i}(\mathbf{x}_{i,1}(t)) - f_{t,i}(\mathbf{x}_{i,2}(t))\right)\mathbf{u}_{t,i}$
25:          Transmit $\Delta_i^{(b_2)}(b) = \mathcal{C}(\mathbf{r}_i(b+1) - \mathbf{r}_i^{(b_2)}(b+1))$ and send $\Delta_i^{(b_2)}(b)$ to $j \in \mathcal{N}_i$
26:          Compute $\mathbf{r}_i^{(b_2+1)}(b+1) = \mathbf{r}_i^{(b_2)}(b+1) + \Delta_i^{(b_2)}(b)$ and set $b_2 = b_2 + 1$
27:       **end for**                     ▷ projection error compensation scheme
28:       Update $\hat{\mathbf{x}}_j(b+1) = \hat{\mathbf{y}}_j^{(L_1+1)}(b) + \mathbf{r}_j^{(L_2+1)}(b+1)$ for $j \in \mathcal{N}_i$
29:       Compute $\mathbf{z}_i(b) = \sum_{t=(b-1)L+1}^{bL} \hat{\mathbf{g}}_{t,i}$ and update $\mathbf{x}_i(b+1) = P_{(1-\zeta)\mathcal{X}}(\mathbf{y}_i^{(L_1+1)}(b))$
30:    **end if**
31: **end for**

---

$\nabla f_{t,i}(\mathbf{x}_i(b) + \epsilon\mathbf{u}_{t,i})$. In each round $t \in [(b-1)L + 1, bL]$, each learner $i$ plays the decision $\mathbf{x}_i(b) + \epsilon\mathbf{u}_{t,i}$ and construct the gradient estimator. Then we perform our two novel techniques. At the end of each block, we project onto the domain $(1-\zeta)\mathcal{X}$, where $\zeta \in (0, 1)$ is the shrinkage size.

## C.2 Two-point Bandit Feedback Setting

In the two-point bandit feedback setting (Agarwal et al., 2010), since the learner $i$ can have access to loss values $f_{t,i}(\mathbf{x}_{i,1}(t))$ and $f_{t,i}(\mathbf{x}_{i,2}(t))$ in each round, the regret under the two-point bandit feedback setting is redefined as

$$R_2(T, i) = \sum_{t=1}^T \sum_{j=1}^n \frac{f_{t,j}(\mathbf{x}_{i,1}(t)) + f_{t,j}(\mathbf{x}_{i,2}(t))}{2} - \min_{\mathbf{x}\in\mathcal{X}} \sum_{t=1}^T \sum_{j=1}^n f_{t,j}(\mathbf{x}).$$

We utilize the classic gradient estimator (Agarwal et al., 2010). Our method, named as Top-DOBD-2, presented in Algorithm 4. For $t \in [(b-1)L + 1, bL]$, each learner plays two decision $\mathbf{x}_{i,1}(t) = \mathbf{x}_i(b) + \epsilon\mathbf{u}_{t,i}$ and $\mathbf{x}_{i,2}(t) = \mathbf{x}_i(b) - \epsilon\mathbf{u}_{t,i}$ and suffers two loss function. We construct the gradient estimator as

$$\hat{\mathbf{g}}_{t,i} = \frac{d}{2\epsilon}\left(f_{t,i}(\mathbf{x}_{i,1}(t)) - f_{t,i}(\mathbf{x}_{i,2}(t))\right)\mathbf{u}_{t,i}.$$

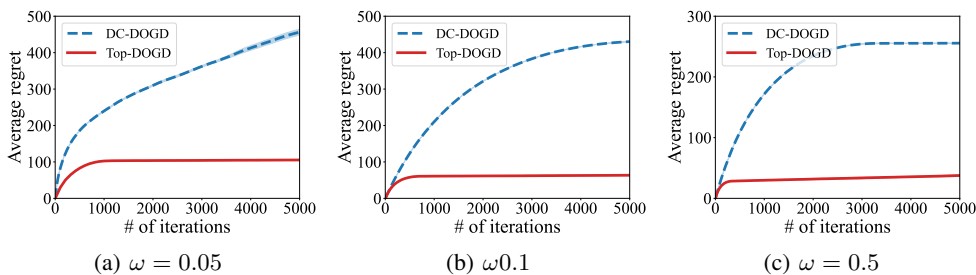

(a) $\omega = 0.05$  (b) $\omega 0.1$  (c) $\omega = 0.5$

Figure 1: Results for Convex Functions under the Full-information Setting.

In each round $t \in [(b-1)L + 1, bL]$, each learner $i$ plays two decisions $\mathbf{x}_{i,1}(t) = \mathbf{x}_i(b) + \epsilon \mathbf{u}_{t,i}$ and $\mathbf{x}_{i,2}(t) = \mathbf{x}_i(b) - \epsilon \mathbf{u}_{t,i}$ and construct the gradient estimator. Then we apply our two novel techniques. At the end of each block, we projection the decision onto the domain $(1 - \zeta)\mathcal{X}$.

# D  EXPERIMENTS

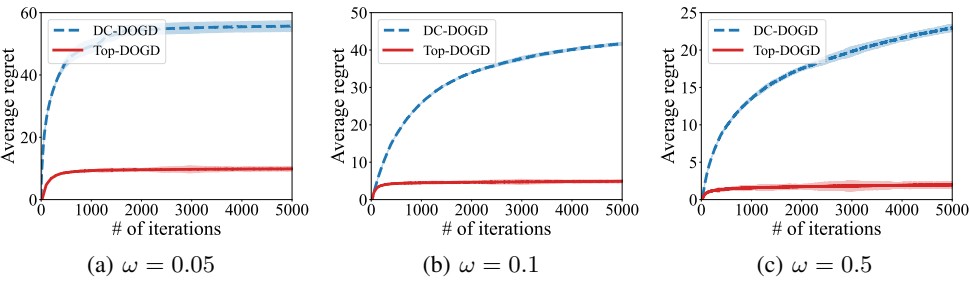

(a) $\omega = 0.05$  (b) $\omega = 0.1$  (c) $\omega = 0.5$

Figure 2: Results for Strongly Convex Functions under the Full-information Setting.

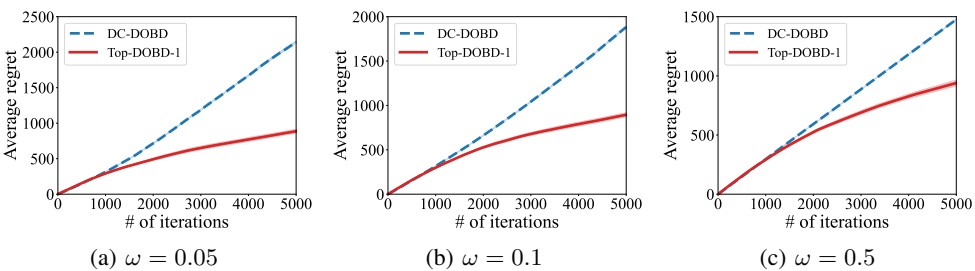

(a) $\omega = 0.05$  (b) $\omega = 0.1$  (c) $\omega = 0.5$

Figure 3: Results for Convex Functions under the One-point Bandit Setting.

In this section, we assess the performance of the proposed methods on an online classification task. We compare the proposed algorithms Top-DOGD, Top-DOBD-1, and Top-DOBD-2 with the existing methods DC-DOGD, DC-DOBD, and DC-DOBD-2 (Tu et al., 2022).

**Setup.** We perform online classification to evaluate the performance of our method with ijcnn1 dataset from LIBSVM (Chang & Lin, 2011). In each round $t \in [T]$, a batch of training examples $\{(\mathbf{w}_{t,1}, y_{t,1}), \ldots, (\mathbf{w}_{t,m}, y_{t,m})\}$ arrive, where $(\mathbf{w}_{t,k}, y_{t,k}) \in [-1, 1]^d \times \{-1, 1\}, k = 1, \ldots, m$. Each online learner $i$ aims to predict a linear model $\mathbf{x}_i(t)$ and suffers a local loss. We consider two types of convex functions. For general convex functions, the learner $i$ suffers the logistic loss $f_{t,i}(\mathbf{x}) = \frac{1}{m_i} \sum_{k=1}^{m_i} \ln(1 + \exp(-y_{t,k}\mathbf{x}^\top \mathbf{w}_{t,k}))$, and for strongly convex functions, the learner $i$ suffers the regularized hinge loss $f_{t,i}(\mathbf{x}) = \frac{1}{m_i} \sum_{k=1}^{m_i} \max\{0, 1 - y_{t,k}\mathbf{x}^\top \mathbf{w}_{t,k}\} + \frac{\mu}{2} \|\mathbf{x}\|^2$, where

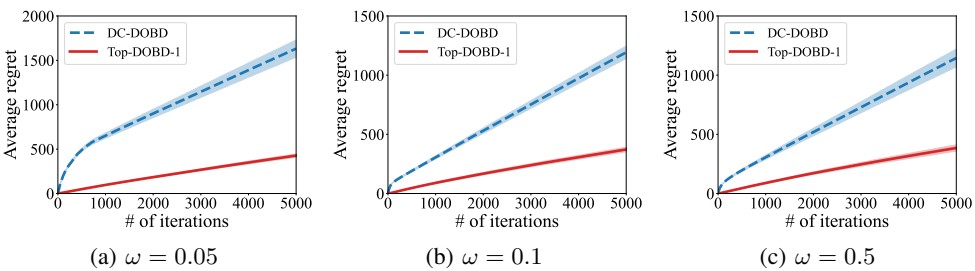

Figure 4: Results for Strongly Convex Functions under the One-point Bandit Setting.

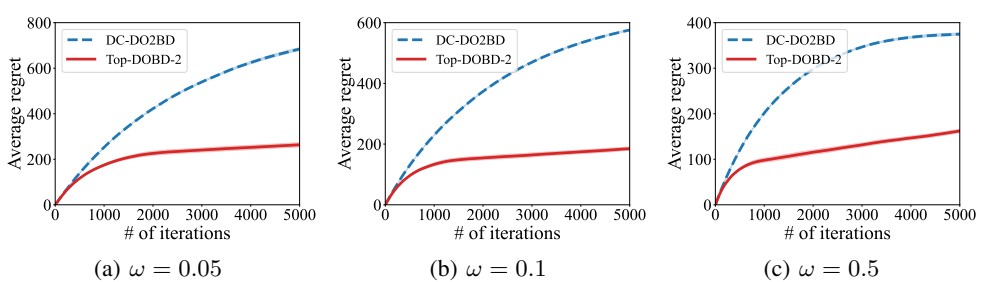

Figure 5: Results for Convex Functions under the Two-point Bandit Setting.

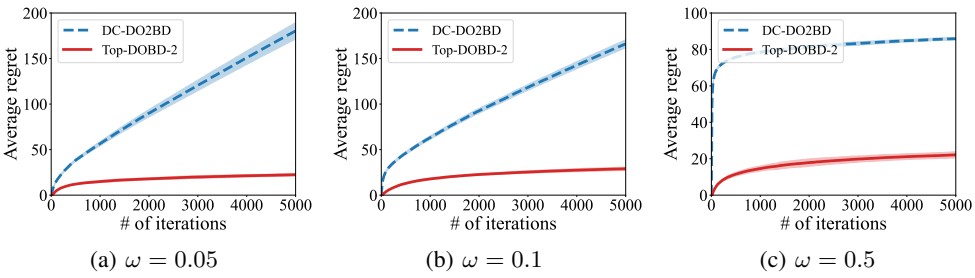

Figure 6: Results for Strongly Convex Functions under the Two-point Bandit Setting.

$\mu$ is the regularization parameter, $m_i$ is the data available to learner $i$ and $\sum_{i=1}^{n} m_i = m$. The connected communication network $\mathcal{G}([n], E)$ with $n$ nodes and $|E|$ edges is generated randomly by tool NetworkX (Hagberg et al., 2008), and then we use the Metropolis rule (Xiao & Boyd, 2004) to construct the connectivity matrix $P$. We set $T = 5000, m = 48$ and $\mu = 0.1$ for strongly convex setting. We conduct the experiments with 3 types of compressors with $\omega = 0.05, 0.1, 0.5$ under graph with $n = 16, |E| = 24$.

**Results.** We repeat each experiment 5 times and depict the mean curve with error bars. We record the average regret $\frac{1}{n} \sum_{i=1}^{n} R(T, i)$. As demonstrated in Figure 1, Figure 2, Figure 3, Figure 4, Figure 5 and Figure 6, it is evident that our methods suffer less regret, outperforming the approaches in Tu et al. (2022). Moreover, we also observe that our methods exhibit increasing improvement over the existing methods as the compression ratio $\omega$ decreases, which aligns with the theoretical guarantees. Moreover, owing to the advantage in the dependence on $\omega$, our methods are more robust compared to Tu et al. (2022).

# E   PROOF OF THEOREMS

## E.1   PROOF OF THEOREM 1

**Notation.** Let $n$ be the total number of learners, $d$ be the dimensionality, $L$ be the block size, $\omega$ be the compression ratio. In the proof, we use $\|\cdot\|$ for $\ell_2$-norm in default and $T/L$ is assumed to be an integer without loss of generality. We give some definitions

$$\tilde{\mathbf{x}}_i(b+1) = \mathbf{y}_i^{(L_1+1)}(b),$$

$$\mathbf{r}_i(b+1) = P_{\mathcal{X}}(\mathbf{y}_i^{(L_1+1)}(b)) - \mathbf{y}_i^{(L_1+1)}(b) = \mathbf{x}_i(b+1) - \tilde{\mathbf{x}}_i(b+1),$$

$$\overline{\mathbf{x}}(b) = \frac{1}{n}\sum_{i=1}^{n}\mathbf{x}_i(b),$$

$$\sum_{i=1}^{n}\sum_{j\in\mathcal{N}_i}P_{ij}(\hat{\mathbf{y}}_j^{(b_1+1)}(b) - \hat{\mathbf{y}}_i^{(b_1+1)}(b)) = 0,$$

$$\mathbf{r}_i^{\mathcal{C}}(b+1) = \mathbf{r}_i^{(L_2+1)}(b+1),$$

$$X(b) = [\mathbf{x}_1(b), ..., \mathbf{x}_n(b)] \in \mathbb{R}^{d\times n}, \tilde{X}(b) = [\tilde{\mathbf{x}}_1(b), ..., \tilde{\mathbf{x}}_n(b)] \in \mathbb{R}^{d\times n},$$

$$\overline{X}(b) = [\overline{\mathbf{x}}_1(b), ..., \overline{\mathbf{x}}_n(b)] \in \mathbb{R}^{d\times n}, R(b) = [\mathbf{r}_1(b), ..., \mathbf{r}_n(b)] \in \mathbb{R}^{d\times n},$$

$$R^{\mathcal{C}}(b) = [\mathbf{r}_1^{\mathcal{C}}(b), ..., \mathbf{r}_n^{\mathcal{C}}(b)] \in \mathbb{R}^{d\times n}, Z(b) = [\mathbf{z}_1(b), ..., \mathbf{z}_n(b)] \in \mathbb{R}^{d\times n}$$

$$Y^{(b_1)}(b) = [\mathbf{y}_1^{(b_1)}(b), ..., \mathbf{y}_n^{(b_1)}(b)] \in \mathbb{R}^{d\times n}, \hat{Y}^{(b_1)}(b) = [\hat{\mathbf{y}}_1^{(b_1)}(b), ..., \hat{\mathbf{y}}_n^{(b_1)}(b)] \in \mathbb{R}^{d\times n}.$$

The forth equality is due to the variable $\hat{\mathbf{y}}_i^{(b_1)}(b)$ is same in all $j \in \mathcal{N}_i$.

By using our definitions, we obtain the following equivalent update rules

$$\hat{X}(b+1) = \hat{Y}^{(L_1+1)}(b) + R^{\mathcal{C}}(b+1),$$

$$X(b+1) = \tilde{X}(b+1) + R(b+1).$$

We next recall two basic projection inequalities:

$$\|P_{\mathcal{X}}(\mathbf{x}) - P_{\mathcal{X}}(\mathbf{y})\| \le \|\mathbf{x} - \mathbf{y}\|, \text{ for } \forall\mathbf{x}, \mathbf{y} \in \mathbb{R}^d, \tag{7}$$

$$\langle P_{\mathcal{X}}(\mathbf{x}) - \mathbf{x}, \mathbf{x} - \mathbf{y}\rangle \le - \|P_{\mathcal{X}}(\mathbf{x}) - \mathbf{x}\|^2 \le 0, \text{ for } \forall\mathbf{x} \in \mathbb{R}^d, \forall\mathbf{y} \in \mathcal{X}. \tag{8}$$

We first present a lemma that characterizes the regret of learner $i$.

**Lemma 3** *Under Assumption 1, 2, 4, 5, the regret of learner $i$ for Algorithm 2 is*

$$\mathbb{E}_{\mathcal{C}}\left[R(T,i)\right] = \sum_{b=1}^{T/L}\sum_{t=(b-1)L+1}^{bL}\sum_{j=1}^{n}f_{t,j}(\mathbf{x}_i(b)) - \sum_{t=1}^{T}\sum_{j=1}^{n}f_{t,j}(\mathbf{x})$$

$$\le \frac{nD^2}{2\eta_{T/L}} + 3L^2G^2n\sum_{b=1}^{T/L}\eta_b + \sum_{b=1}^{T/L}\frac{3}{2\eta_b}\mathbb{E}_{\mathcal{C}}\left[\|R(b+1)\|_F^2\right] + \frac{1}{2\eta_b}\mathbb{E}_{\mathcal{C}}\left[\left\|X(b) - \tilde{X}(b+1)\right\|_F^2\right]$$

$$+ 3nGL\sum_{b=1}^{T/L}\mathbb{E}_{\mathcal{C}}\left[\left\|X(b) - \overline{X}(b)\right\|_F\right] + \sum_{b=1}^{T/L}\frac{1}{2\eta_b}\mathbb{E}_{\mathcal{C}}\left[\|R(b)\|_F^2\right].$$

According to Lemma 3, we have

$$\mathbb{E}_{\mathcal{C}}\left[R(T,i)\right] \le \frac{nD^2}{2\eta_{T/L}} + 3L^2G^2n\sum_{b=1}^{T/L}\eta_b + \sum_{b=1}^{T/L}\frac{3}{2\eta_b}\mathbb{E}_{\mathcal{C}}\left[\|R(b+1)\|_F^2\right] + \frac{1}{2\eta_b}\mathbb{E}_{\mathcal{C}}\left[\left\|X(b) - \tilde{X}(b+1)\right\|_F^2\right]$$

$$+ 3nGL\sum_{b=1}^{T/L}\mathbb{E}_{\mathcal{C}}\left[\left\|X(b) - \overline{X}(b)\right\|_F\right] + \sum_{b=1}^{T/L}\frac{1}{2\eta_b}\mathbb{E}_{\mathcal{C}}\left[\|R(b)\|_F^2\right].$$

Next, to give the bound of each term, we present the following lemma.

**Lemma 4** *Under Assumption 1, 2, 4 and 5, by selecting $L_1 = \lceil \frac{28(2\rho\beta^2+4\beta^2+(2-\omega)(\beta^2+2\beta)\rho+\rho^2)}{\omega\rho^2} \rceil, L_2 = \lceil \frac{\ln(8n)}{\omega} \rceil, \gamma = \frac{\omega\rho}{2\rho\beta^2+4\beta^2+(2-\omega)(\beta^2+2\beta)\rho+\rho^2}$ we have the following guarantees.*

$$\mathbb{E}_{\mathcal{C}}\left[\|R(b+1)\|_F^2\right] \leq \frac{2}{7n}\mathbb{E}_{\mathcal{C}}\left[\left\|X(b)-\hat{X}(b)\right\|_F^2 + \left\|X(b)-\overline{X}(b)\right\|_F^2\right] + (2n+\frac{10}{7})L^2G^2\eta_b^2,$$

$$\mathbb{E}_{\mathcal{C}}\left[\left\|X(b+1)-\overline{X}(b+1)\right\|_F^2\right] \leq \frac{1}{7n}\mathbb{E}_{\mathcal{C}}\left[\left\|X(b)-\hat{X}(b)\right\|_F^2 + \left\|X(b)-\overline{X}(b)\right\|_F^2\right] + \frac{5}{7}L^2G^2\eta_b^2$$

$$\mathbb{E}_{\mathcal{C}}\left[\left\|X(b+1)-\hat{X}(b+1)\right\|_F^2\right] \leq \frac{5}{14n}\mathbb{E}_{\mathcal{C}}\left[\left\|X(b)-\hat{X}(b)\right\|_F^2 + \left\|X(b)-\overline{X}(b)\right\|_F^2\right] + 2L^2G^2\eta_b^2$$

We define the error $e_{b+1}$ as follows

$$e_{b+1} = \mathbb{E}_{\mathcal{C}}\left[\sum_{i=1}^n \|\mathbf{x}_i(b+1)-\overline{\mathbf{x}}_i(b+1)\|^2 + \|\mathbf{x}_i(b+1)-\hat{\mathbf{x}}_i(b+1)\|^2\right]$$

$$= \mathbb{E}_{\mathcal{C}}[\|X(b+1)-\overline{X}(b+1)\|_F^2] + \mathbb{E}_{\mathcal{C}}[\|X(b+1)-\hat{X}(b+1)\|_F^2].$$

By fixing the learning rate $\eta_b = \eta$, we have the following guarantee

$$e_{b+1} \leq \frac{1}{2n}e_b + 3\eta^2L^2G^2. \tag{9}$$

By summing up, we have

$$e_{b+1} \leq \frac{1}{1-\frac{1}{2n}}3\eta^2L^2G^2 \leq 6\eta^2L^2G^2,$$

which is due to $\sum_{i=1}^b \frac{1}{(2n)^i} \leq \frac{1}{1-\frac{1}{2n}} \leq 2$ and $e_1 = 0$.

As for the term $\mathbb{E}_{\mathcal{C}}\left[\|R(b+1)\|_F^2\right]$, we have

$$\mathbb{E}_{\mathcal{C}}\left[\|R(b+1)\|_F^2\right] \leq \frac{2}{7n}e_b + (2n+\frac{10}{7})L^2G^2\eta^2$$

$$\leq \frac{1}{1-\frac{2}{7n}}(2n+\frac{10}{7})L^2G^2\eta^2$$

$$\leq (\frac{14n}{7n-2}n+\frac{10n}{7n-2})L^2G^2\eta^2$$

$$\leq (3n+2)L^2G^2\eta^2,$$

which is same to $\mathbb{E}_{\mathcal{C}}\left[\|R(b)\|_F^2\right]$. Now, we can derive the regret bound of Top-DOGD.

First, we have
$$E_{\mathcal{C}}\left[\left\|X(b)-\overline{X}(b)\right\|_F\right] \leq \sqrt{e_b} \leq \sqrt{6}\eta LG \leq 3\eta LG.$$

As for the second term, we have

$$\mathbb{E}_{\mathcal{C}}\left[\left\|\overline{X}(b)-\tilde{X}(b+1)\right\|_F^2\right]$$

$$= \mathbb{E}_{\mathcal{C}}\left[\sum_{i=1}^n \|\overline{\mathbf{x}}(b)-\tilde{\mathbf{x}}_i(b+1)\|^2\right]$$

$$= \mathbb{E}_{\mathcal{C}}\left[\sum_{i=1}^n \|\overline{\mathbf{x}}(b)-\overline{\mathbf{x}}(b+1)+\overline{\mathbf{x}}(b+1)-\tilde{\mathbf{x}}_i(b+1)\|^2\right] \tag{10}$$

$$\leq 2\mathbb{E}_{\mathcal{C}}\left[\sum_{i=1}^n \|\overline{\mathbf{x}}(b)-\overline{\mathbf{x}}(b+1)\|^2\right] + 2\mathbb{E}_{\mathcal{C}}\left[\sum_{i=1}^n \|\overline{\mathbf{x}}(b+1)-\tilde{\mathbf{x}}_i(b+1)\|^2\right]$$

$$\leq 2n\eta^2L^2G^2 + 2e_{b+1} \leq (2n+12)\eta^2L^2G^2.$$

Finally, by setting $\eta_b = \eta = \frac{D}{G\sqrt{LT}}, L = L_1 + L_2 = O(\omega^{-1}\rho^{-2}\ln n)$, we have

$$
\begin{aligned}
\mathbb{E}_{\mathcal{C}}\left[R(T,i)\right] =& \mathbb{E}_{\mathcal{C}}\left[\sum_{b=1}^{T/L}\sum_{t=(b-1)L+1}^{bL}\sum_{j=1}^{n}f_{t,j}(\mathbf{x}_i(b)) - \sum_{t=1}^{T}\sum_{j=1}^{n}f_{t,j}(\mathbf{x})\right] \\
\leq& \frac{nD^2}{2\eta_{T/L}} + 3L^2G^2n\sum_{b=1}^{T/L}\eta_b + \sum_{b=1}^{T/L}\frac{3}{2\eta_b}\mathbb{E}_{\mathcal{C}}\left[\|R(b+1)\|_F^2\right] + \frac{1}{2\eta_b}\mathbb{E}_{\mathcal{C}}\left[\left\|X(b) - \tilde{X}(b+1)\right\|_F^2\right] \\
& + 3nGL\sum_{b=1}^{T/L}\mathbb{E}_{\mathcal{C}}\left[\left\|X(b) - \overline{X}(b)\right\|_F\right] + \sum_{b=1}^{T/L}\frac{1}{2\eta_b}\mathbb{E}_{\mathcal{C}}\left[\|R(b)\|_F^2\right] \\
\leq& \frac{nD^2}{2\eta} + 3nLG^2\eta T + \sum_{b=1}^{T/L}\frac{3}{2\eta}\mathbb{E}_{\mathcal{C}}\left[\|R(b+1)\|_F^2\right] + \frac{1}{2\eta}\mathbb{E}_{\mathcal{C}}\left[\left\|\overline{X}(b) - \tilde{X}(b+1)\right\|_F^2\right] \\
& + 3nGL\sum_{b=1}^{T/L}\mathbb{E}_{\mathcal{C}}\left[\left\|X(b) - \overline{X}(b)\right\|_F\right] + \sum_{b=1}^{T/L}\frac{1}{2\eta}\mathbb{E}_{\mathcal{C}}\left[\|R(b)\|_F^2\right] \\
\leq& \frac{nD^2}{2\eta} + 3nLG^2T\eta + (5n+3)LG^2T\eta + (n+6)LG^2T\eta + 9n\eta TLG^2 + (2n+1)LG^2T\eta \\
\leq& O(n\sqrt{LT}) = O(\omega^{-1/2}\rho^{-1}n\sqrt{\ln n}\sqrt{T}).
\end{aligned}
$$

### E.2 PROOF OF THEOREM 2

The proof follows the similar structure as Theorem 1, except that we exploit strong convexity to establish improved regret bounds.

**Lemma 5** *Under Assumption 1, 3, 4, 5, the regret of learner $i$ for Algorithm 2 is*

$$
\begin{aligned}
\mathbb{E}_{\mathcal{C}}\left[R(T,i)\right] =& \sum_{b=1}^{T/L}\sum_{t=(b-1)L+1}^{bL}\sum_{j=1}^{n}f_{t,j}(\mathbf{x}_i(b)) - \sum_{t=1}^{T}\sum_{j=1}^{n}f_{t,j}(\mathbf{x}) \\
\leq& \frac{nD^2}{2}\sum_{b=1}^{T/L}(\frac{1}{\eta_b} - \frac{1}{\eta_{b-1}} - \mu L) + 3L^2G^2n\sum_{b=1}^{T/L}\eta_b + 3nGL\sum_{b=1}^{T/L}\mathbb{E}_{\mathcal{C}}\left[\left\|X(b) - \overline{X}(b)\right\|_F\right] \\
& + \sum_{b=1}^{T/L}\frac{3}{2\eta_b}\mathbb{E}_{\mathcal{C}}\left[\|R(b+1)\|_F^2\right] + \frac{1}{2\eta_b}\mathbb{E}_{\mathcal{C}}\left[\left\|X(b) - \tilde{X}(b+1)\right\|_F^2\right] + \frac{1}{2\eta_b}\mathbb{E}_{\mathcal{C}}\left[\|R(b)\|_F^2\right].
\end{aligned}
$$

According to Lemma 4, we have the following

$$
e_{b+1} \leq \frac{1}{2n}e_b + 3\eta_b^2 L^2 G^2. \tag{11}
$$

To establish the bound of $e_{b+1}$, we introduce the following lemma.

**Lemma 6** *Let $\{e_b\}_{b\geq 1}$ denote a sequence of values satisfying $e_1 = 0$ and*

$$
e_{b+1} \leq \frac{1}{2n}e_b + q\eta_b^2 L^2,
$$

*where $q > 0$, $\eta_b = \frac{1}{\mu(bL+8)}$. We have the following guarantee*

$$
e_b \leq 4qL^2\eta_b^2.
$$

Therefore, by setting $\eta_b = \frac{1}{\mu(bL+8)}$, we have

$$
e_b \leq 12L^2G^2\eta_b^2.
$$

Then we give the bound of the terms in the regret individually

$$E_{\mathcal{C}}\left[\left\|X(b) - \overline{X}(b)\right\|_F\right] \leq \sqrt{e_b} \leq 2\sqrt{3}\eta_b LG.$$

As for the second term, we have

$$
\begin{aligned}
&\mathbb{E}_{\mathcal{C}}\left[\left\|\overline{X}(b) - \tilde{X}(b+1)\right\|_F^2\right] \\
=&\mathbb{E}_{\mathcal{C}}\left[\sum_{i=1}^{n}\|\overline{\mathbf{x}}(b) - \tilde{\mathbf{x}}_i(b+1)\|^2\right] \\
=&\mathbb{E}_{\mathcal{C}}\left[\sum_{i=1}^{n}\|\overline{\mathbf{x}}(b) - \overline{\mathbf{x}}(b+1) + \overline{\mathbf{x}}(b+1) - \tilde{\mathbf{x}}_i(b+1)\|^2\right] \\
\leq&2\mathbb{E}_{\mathcal{C}}\left[\sum_{i=1}^{n}\|\overline{\mathbf{x}}(b) - \overline{\mathbf{x}}(b+1)\|^2\right] + 2\mathbb{E}_{\mathcal{C}}\left[\sum_{i=1}^{n}\|\overline{\mathbf{x}}(b+1) - \tilde{\mathbf{x}}_i(b+1)\|^2\right] \\
\leq&2n\eta_{b+1}^2 L^2 G^2 + 2e_{b+1} \leq (2n+24)\eta_{b+1}^2 L^2 G^2.
\end{aligned}
\tag{12}
$$

For $\mathbb{E}_{\mathcal{C}}\left[\frac{1}{\eta_b}\|R(b+1)\|_F^2\right]$, we have the following.

$$
\begin{aligned}
&\frac{1}{\eta_b}\mathbb{E}_{\mathcal{C}}\left[\|R(b+1)\|_F^2\right] \\
\leq&\frac{1}{\eta_b}\left(\frac{2}{7n}e_b + (2n + \frac{10}{7})L^2 G^2 \eta_b^2\right) \\
\leq&\frac{1}{\eta_b}\left((8n + \frac{40}{7})L^2 G^2 \eta_b^2\right) \\
\leq&(8n+6)L^2 G^2 \eta_b.
\end{aligned}
$$

For $\mathbb{E}_{\mathcal{C}}\left[\frac{1}{\eta_b}\|R(b)\|_F^2\right]$, we have the following.

$$
\begin{aligned}
&\frac{1}{\eta_b}\mathbb{E}_{\mathcal{C}}\left[\|R(b)\|_F^2\right] \\
\leq&\frac{1}{\eta_b}\left(\frac{2}{7n}e_{b-1} + 4nL^2 G^2 \eta_{b-1}^2\right) \\
\leq&(8n+6)L^2 G^2 \frac{\eta_{b-1}^2}{\eta_b} \\
\leq&(16n+12)L^2 G^2 \eta_{b-1},
\end{aligned}
$$

where the last inequality is due to $\frac{\eta_{b-1}}{\eta_b} \leq 2$.

Therefore, we can derive the regret bound of Algorithm 2 for strongly convex functions. By setting $\eta_b = \frac{1}{\mu(bL+8)}$, we have

$$\mathbb{E}_{\mathcal{C}}\left[R(T,i)\right] = \sum_{b=1}^{T/L}\sum_{t=(b-1)L+1}^{bL}\sum_{j=1}^{n} f_{t,j}(\mathbf{x}_i(b)) - \sum_{t=1}^{T}\sum_{j=1}^{n} f_{t,j}(\mathbf{x})$$

$$\leq \frac{nD^2}{2}\sum_{b=1}^{T/L}(\frac{1}{\eta_b} - \frac{1}{\eta_{b-1}} - \mu L) + 3L^2G^2 n\sum_{b=1}^{T/L}\eta_b + 3nGL\sum_{b=1}^{T/L}\mathbb{E}_{\mathcal{C}}\left[\left\|X(b) - \overline{X}(b)\right\|_F\right]$$

$$+ \sum_{b=1}^{T/L}\frac{3}{2\eta_b}\mathbb{E}_{\mathcal{C}}\left[\left\|R(b+1)\right\|_F^2\right] + \frac{1}{2\eta_b}\mathbb{E}_{\mathcal{C}}\left[\left\|X(b) - \tilde{X}(b+1)\right\|_F^2\right] + \frac{1}{2\eta_b}\mathbb{E}_{\mathcal{C}}\left[\left\|R(b)\right\|_F^2\right]$$

$$\leq \frac{nD^2}{2}(\frac{1}{\eta_1} - \mu L) + 3L^2G^2 n\sum_{b=1}^{T/L}\eta_b + 6\sqrt{3}nL^2G^2\sum_{b=1}^{T/L}\eta_b$$

$$+ (12n+9)L^2G^2\sum_{b=1}^{T/L}\eta_b + (n+12)L^2G^2\sum_{b=1}^{T/L}\eta_b + (8n+6)L^2G^2\sum_{b=1}^{T/L}\eta_{b-1}$$

$$\leq 4nD^2\mu + \frac{1}{\mu}\left(3LG^2 n\ln(T+8) + 6\sqrt{3}nG^2 L\ln(T+8)\right.$$

$$\left. + (20n+15)G^2 L\ln(T+8) + (n+12)G^2 L\ln(T+8)\right)$$

$$\leq O(Ln\ln(T)) = O(\omega^{-1}\rho^{-2}n\ln n\ln T).$$

where the last inequality is due to $\sum_{b=1}^{T/L}\frac{L}{\mu(bL+8)} \leq \frac{1}{\mu}\sum_{b=1}^{T/L}\sum_{t=(b-1)L+1}^{bL}\frac{1}{t+8} \leq \frac{1}{\mu}\int_0^T\frac{1}{t+8}dt \leq \frac{1}{\mu}\ln(T+8) = O(\ln T)$.

### E.3 PROOF OF THE THEOREM 3

The structure of our proof follows that of Wan et al. (2024), with the main distinction being that we incorporate a dedicated compressor to derive the lower bound. Let $A \in \mathbb{R}^{n\times n}$ denote the adjacency matrix of $\mathcal{G}$, and let $\delta_i = |N_i| - 1$ denote the degree of node $i$. As presented in Duchi et al. (2011), for any connected undirected graph, there exists a specific gossip matrix $P$ satisfying Assumption 1, i.e.,

$$P = I_n - \frac{1}{\delta_{\max} + 1}(D - A), \tag{13}$$

where $\delta_{\max} = \max\{\delta_1, ..., \delta_n\}$ and $D = \text{diag}\{\delta_1, ..., \delta_n\}$.

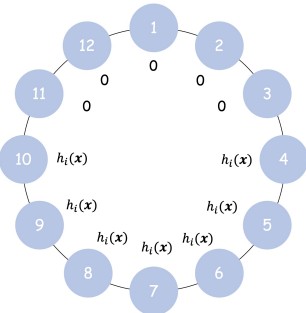

Figure 7: Example of 1-connected cycle graph

To maximize the impact of communication on the regret bound, we focus on the 1-connected cycle graph, where the graph $\mathcal{G}$ is constructed by arranging $n$ nodes on a circle and connecting each node with its immediate left and right neighbors. We adopt the randomized gossip compressor $\mathcal{C}(\cdot)$, which outputs $Q(\mathbf{x}) = \mathbf{x}$ with probability $\omega \in (0,1]$ and $Q(\mathbf{x}) = \mathbf{0}_d$ otherwise. Under this scheme, two connected learners $i$ and $j$ can successfully exchange data only with probability $\omega$ in each round. Consequently, the expected number of rounds required for a successful exchange is $1/\omega$.

To derive the lower bound, we attempt to maximize the regret of learner 1. Specifically, we set the loss functions as

$$f_{t,\,n-\lceil m/2\rceil+2}(\mathbf{x}) = \cdots = f_{t,n}(\mathbf{x}) = f_{t,1}(\mathbf{x}) = f_{t,2}(\mathbf{x}) = \cdots = f_{t,\lceil m/2\rceil}(\mathbf{x}) = 0,$$

while the other loss functions are set carefully to construct the desired lower bound. It is straightforward to see that when $\omega = 1$, learner 1 must go through $\lceil m/2 \rceil$ rounds of communication to receive information from learners $\lceil m/2 \rceil + 1, \ldots, n - \lceil m/2 \rceil + 1$. An illustrative example is provided in Figure 7. Let $K_1$ denote the communication rounds. When $\omega < 1$, the expected communication rounds becomes

$$\mathbb{E}_{\mathcal{C}}\left[K_1\right] = \left\lceil \frac{m}{2\omega} \right\rceil. \tag{14}$$

We give the proof below. For two player $i$ and $i + 1$, the expected round $a_{i,i+1}$ for a successful transmission is

$$\mathbb{E}[a_{i,i+1}] = \sum_{k=1}^{\infty} k(1-\omega)^{k-1}\omega = \omega \sum_{k=1}^{\infty} k(1-\omega)^{k-1} = \frac{1}{\omega},$$

where the second equality is due to $\sum_{k=1}^{\infty} kq^{k-1} = \frac{1}{(1-q)^2}$ for $|q| < 1$. Therefore, we have

$$\mathbb{E}[K_1] = \mathbb{E}\left[\sum_{1 \le i \le \lceil m/2\rceil} a_{i,i+1}\right] = \sum_{1 \le i \le \lceil m/2\rceil} \mathbb{E}[a_{i,i+1}] = \left\lceil \frac{m}{2\omega} \right\rceil.$$

Let $K = \lceil m/2 \rceil, Z = \lfloor (T-1)/K_1 \rfloor, c_0 = 0$ and $c_{(Z+1)K_1} = T$. The total $T$ rounds can be divided into the following $Z + 1$ intervals

$$[c_0 + 1,\, c_1],\ [c_1 + 1,\, c_2],\ \ldots,\ [c_Z + 1,\, c_{Z+1}].$$

Following Wan et al. (2025), for any $i \in \{0, 1, ..., Z\}$ and $t \in [c_i + 1,\, c_{i+1}]$, we set

$$f_{t,\,\lceil m/2\rceil+1}(\mathbf{x}) = \cdots = f_{t,n-\lceil m/2\rceil+1}(\mathbf{x}) = h_i(\mathbf{x}) = \langle \mathbf{w}_i, \mathbf{x} \rangle,$$

where the coordinates of $\mathbf{w}_i$ is $\pm G/\sqrt{d}$ with probability 0.5. Then the global loss function is

$$f_t(\mathbf{x}) = (n - 2K + 1)h_i(\mathbf{x}).$$

Moreover, it is obvious that the decision $\mathbf{x}_1(c_i + 1), \ldots, \mathbf{x}_1(c_{i+1})$ for any $i \in \{0, \ldots, Z\}$ are made before the learner 1 has the access to $h_i(\mathbf{x})$. Then we can derive the expected lower bound for $R(T, 1)$.

$$
\begin{aligned}
\mathbb{E}_{\mathbf{w}_0,\ldots,\mathbf{w}_Z}[R(T,1)] &= \mathbb{E}_{\mathbf{w}_0,\ldots,\mathbf{w}_Z}\left[\sum_{i=0}^{Z}\sum_{t=c_i+1}^{c_{i+1}}(n-2K+1)h_i(\mathbf{x}_1(t)) - \sum_{i=0}^{Z}\sum_{t=c_i+1}^{c_{i+1}}(n-2K+1)h_i(\mathbf{x})\right] \\
&= (n-2K+1)\mathbb{E}_{\mathbf{w}_0,\ldots,\mathbf{w}_Z}\left[\sum_{i=0}^{Z}\sum_{t=c_i+1}^{c_{i+1}}\langle \mathbf{w}_i, \mathbf{x}_1(t)\rangle - \min_{\mathbf{x}\in\mathcal{K}}\sum_{i=0}^{Z}(c_{i+1}-c_i)\langle \mathbf{w}_i, \mathbf{x}\rangle\right] \\
&= -(n-2K+1)\mathbb{E}_{\mathbf{w}_0,\ldots,\mathbf{w}_Z}\left[\min_{\mathbf{x}\in\mathcal{K}}\sum_{i=0}^{Z}(c_{i+1}-c_i)\langle \mathbf{w}_i, \mathbf{x}\rangle\right] \\
&= -(n-2K+1)\mathbb{E}_{\mathbf{w}_0,\ldots,\mathbf{w}_Z}\left[\min_{\mathbf{x}\in\{-D/2\sqrt{d}, D/2\sqrt{d}\}^d}\left\langle \mathbf{x}, \sum_{i=0}^{Z}(c_{i+1}-c_i)\mathbf{w}_i\right\rangle\right],
\end{aligned}
\tag{15}
$$

where the third equality is due to $\mathbb{E}_{\mathbf{w}_0,\ldots,\mathbf{w}_Z}[\langle \mathbf{w}_i, \mathbf{x}_1(t)\rangle] = 0$ for $\forall t \in [c_i + 1, c_{i+1}]$.

Then, we denote $\epsilon_{01}, ..., \epsilon_{0d}, ..., \epsilon_{Z1}, ..., \epsilon_{Zd}$ be the coordinates of $\mathbf{w}_1, ..., \mathbf{w}_Z$, which are identically distributed variables with $\mathbb{P}(\epsilon_{ij} = \pm 1) = 1/2$ for $i \in \{0, ..., Z\}$ and $j \in \{1, ..., d\}$. By using the

Khintchine inequality on (15), we have

$$
\begin{aligned}
\mathbb{E}_{\mathbf{w}_0,...,\mathbf{w}_Z}[R(T,1)] =& -(n-2K+1)\mathbb{E}_{\epsilon_{01},...,\epsilon_{Zd}}\left[\sum_{j=1}^{d}-\frac{D}{2\sqrt{d}}\left|\sum_{i=1}^{Z}(c_{i+1}-c_i)\frac{\epsilon_{ij}G}{\sqrt{d}}\right|\right] \\
=& (n-2K+1)\frac{DG}{2}\mathbb{E}_{\epsilon_{01},...,\epsilon_{Zd}}\left[\left|\sum_{i=0}^{Z}(c_{i+1}-c_i)\epsilon_{i1}\right|\right] \\
\geq& \frac{(n-2K+1)DG}{2\sqrt{2}}\sqrt{\sum_{i=0}^{Z}(c_{i+1}-c_i)^2} \\
\geq& \frac{(n-2K+1)DG}{2\sqrt{2}}\sqrt{\frac{c_{Z+1}-c_0)^2}{Z+1}} \\
=& \frac{(n-2K+1)DGT}{2\sqrt{2Z+2}},
\end{aligned}
\tag{16}
$$

where the second inequality is due to the Cauchy-Schwarz inequality.

By applying $Z = \lfloor (T-1)/\lceil \frac{m}{2\omega} \rceil \rfloor \leq \frac{2\omega(T-1)}{m}$, we have

$$
\begin{aligned}
\mathbb{E}_{\mathbf{w}_0,...,\mathbf{w}_Z}[R(T,1)] \geq& \frac{(n-2K+1)DGT}{2\sqrt{2Z+2}} \geq \frac{(n-m-1)DGT}{2\sqrt{\frac{4\omega(T-1)}{m}+2}} \\
=& \frac{(m+1)DGT}{2\sqrt{\frac{4\omega(T-1)}{m}+2}} = \frac{(m+1)\sqrt{m+1}DGT}{2\sqrt{\frac{4\omega(T-1)}{m}(m+1)+2m+2}} \\
\geq& \frac{(m+1)\sqrt{m+1}DGT}{2\sqrt{8\omega(T-1)+2m+2}} \\
\geq& \frac{n\sqrt{n}DGT}{4\sqrt{16\omega(T-1)+4m+4}} \\
\geq& \frac{n\sqrt{n}DGT}{4\sqrt{16\omega T-16\omega+2n}},
\end{aligned}
$$

where the forth inequality is due to $n = 2m+2$.

Then we introduce a lemma.

**Lemma 7** *(Lemma 6 in Wan et al. (2024)) For the 1-connected cycle graph with $n = 2(m+1)$ where $m$ denotes a positive integer, the gossip matrix defined in (13) satisfies*

$$
\frac{\pi^2}{1-\sigma_2(P)} = \frac{\pi^2}{\rho} \leq 4n^2.
$$

If $n \leq 8\omega T + 8\omega$, by utilizing Lemma 7, we have

$$
\mathbb{E}_{\mathbf{w}_0,...,\mathbf{w}_Z}[R(T,1)] \geq \frac{nDG\sqrt{nT}}{16\sqrt{2\omega}} \geq \frac{nGD\sqrt{\pi T}}{32\omega^{1/2}\rho^{1/4}}.
$$

### E.4 PROOF OF THE THEOREM 4

For the proof of the lower bound for strongly convex loss functions, we follow the analysis of Wan et al. (2025) while redefining both the loss functions and the decision domain. Specifically, we choose the domain $\mathcal{X} = [0, D/\sqrt{d}]$ and define $\mathcal{B}_p$ as the Bernoulli distribution with probability $p$ of obtaining 1. For $t \in [c_i + 1, c_{i+1}]$ and $i \in \{0, ..., Z\}$, we set

$$
f_{t,n-\lceil m/2\rceil+2}(\mathbf{x}) = \cdots = f_{t,n}(\mathbf{x}) = f_{t,1}(\mathbf{x}) = f_{t,2}(\mathbf{x}) = \cdots = f_{t,\lceil m/2\rceil}(\mathbf{x}) = \frac{\mu}{2}\|\mathbf{x}\|^2,
$$

$$f_{t,\,\lceil m/2 \rceil+1}(\mathbf{x}) = \cdots = f_{t,n-\lceil m/2 \rceil+1}(\mathbf{x}) = h_i(\mathbf{x}) = \frac{\mu}{2}\left\|\mathbf{x} - \frac{D}{\sqrt{d}}\mathbf{w}_i\right\|^2,$$

where $\mathbf{w}_i$ is sampled from the vectors $\mathbf{0}_d$ and $\mathbf{1}_d$ with $\mathbb{P}(\mathbf{w}_i = \mathbf{1}_d) = p$. Clearly, $h_i(\mathbf{x})$ satisfies Assumption 3 and Assumption 4 with $G = \mu D$. Then, for any $i \in 0, \ldots, Z$ and $t \in [c_i + 1, c_{i+1}]$, the global loss function can be expressed as

$$f_t(\mathbf{x}) = \sum_{k=1}^n f_{t,k}(\mathbf{x}) = \frac{\mu}{2}(n - 2K + 1)\left\|\mathbf{x} - \frac{D}{\sqrt{d}}\mathbf{w}_i\right\|^2 + \frac{\mu}{2}(2K-1)\|\mathbf{x}\|^2$$

$$= \frac{\mu n}{2}\|\mathbf{x}\|^2 - \frac{\mu(n-2K+1)D}{\sqrt{d}}\langle \mathbf{x}, \mathbf{w}_i\rangle + \frac{\mu(n-2K+1)^2 D^2}{2d}\|\mathbf{w}_i\|^2,$$

with expectation as

$$F(\mathbf{x}) = \mathbb{E}_{\mathbf{w}_0,\ldots,\mathbf{w}_Z}[f_t(\mathbf{x})] = \frac{\mu n}{2}\left\|\mathbf{x} - \frac{(n-2K+1)D\mathbf{p}}{n\sqrt{d}}\right\|^2 + \frac{\mu(n-2K+1)D^2}{2d}\left\langle \mathbf{1} - \frac{(n-2K+1)}{n}\mathbf{p}, \mathbf{p}\right\rangle,$$

where $\mathbf{p} = [p, ..., p] \in \mathbb{R}^d$. We denote $F(\mathbf{x}^*)$ is the minimum of $F(\mathbf{x})$. We have

$$\mathbf{x}^* = \frac{(n-2K+1)D\mathbf{p}}{n\sqrt{d}} \in \mathcal{X},$$

and we further have the following gap

$$F(\mathbf{x}) - F(\mathbf{x}^*) = \frac{\mu n}{2}\left\|\mathbf{x} - \frac{(n-2K+1)D\mathbf{p}}{n\sqrt{d}}\right\|^2. \tag{17}$$

Next, we derive the lower bound for strongly convex functions. We again choose $\mathcal{G}$ as 1-connected cycle graph, which ensures the $\mathbf{x}_1(c_i + 1), ..., \mathbf{x}_1(c_{i+1})$ are independent of $\mathbf{w}_i$.

We have

$$\mathbb{E}_{\mathbf{w}_0,\ldots,\mathbf{w}_Z}[R(T,1)] = \mathbb{E}_{\mathbf{w}_0,\ldots,\mathbf{w}_Z}\left[\sum_{i=0}^Z \sum_{t=c_i+1}^{c_{i+1}} f_t(\mathbf{x}_1(t)) - \min_{\mathbf{x}\in\mathcal{K}}\sum_{i=0}^Z \sum_{t=c_i+1}^{c_{i+1}} f_t(\mathbf{x})\right]$$

$$= \mathbb{E}_{\mathbf{w}_0,\ldots,\mathbf{w}_Z}\left[\sum_{i=0}^Z \sum_{t=c_i+1}^{c_{i+1}} F(\mathbf{x}_1(t))\right] - \mathbb{E}_{\mathbf{w}_0,\ldots,\mathbf{w}_Z}\left[\min_{\mathbf{x}\in\mathcal{K}}\sum_{i=0}^Z \sum_{t=c_i+1}^{c_{i+1}} f_t(\mathbf{x})\right]$$

$$\geq \mathbb{E}_{\mathbf{w}_0,\ldots,\mathbf{w}_Z}\left[\sum_{i=0}^Z \sum_{t=c_i+1}^{c_{i+1}} F(\mathbf{x}_1(t)) - \sum_{i=0}^Z \sum_{t=c_i+1}^{c_{i+1}} F(\mathbf{x}^*)\right]. \tag{18}$$

To give the lower bound of (18), we follow Wan et al. (2025) to show that the regret of the learner 1 on a specific $p$ is large. Following Wan et al. (2025), we introduce a perturbation of the parameter from $p$ to $p'$, and the corresponding random vectors can be rewritten as $\mathbf{w}_0', .., \mathbf{w}_Z'$ and $\mathbf{x}_1'(0), ..., \mathbf{x}_t'(T)$. We assume that the D-OCO algorithm is deterministic without loss of generality. As discussed in Wan et al. (2025), for any $t \in [c_i + 1, c_{i+1}]$, $\mathbf{x}_1(t)$ can be specified by a bit string $X \in \{0,1\}^i$ drawn from $\mathcal{B}_p^i$. For a deterministic algorithm, the local learner 1 of the D-OCO algorithm at round $t \in [c_i + 1, c_{i+1}]$ can be denoted as a mapping function $\{0,1\}^i \mapsto \mathcal{X}$ such that $\mathbf{x}_i(t) = \mathcal{A}_t(X)$.

Let $Z_1 = \lfloor \log_{16}(15Z + 16) - 1 \rfloor$ and $Z_1 \geq 1$ due to $16\omega^{-1}n + 1 \leq T$. We further divide the first $Z' = \frac{1}{15}(16^{Z_1+1} - 16)$ intervals into $Z_1$ epochs with the length $16, 16^2, ..., 16^{Z_1}$ and the $m$-th epoch $E_m$ consists of the intervals $\frac{1}{15}(16^m - 16), ..., (16^{m+1} - 16) - 1$ with length of $16^m$. We utilize a lemma.

**Lemma 8** *(Lemma 8 in Wan et al. (2025)) Fix a block $i$ and let $\epsilon \leq \frac{1}{32\sqrt{i+1}}$ be a parameter, $\xi = (n - 2K + 1)D/\sqrt{d}$ and $\mathbf{p} = [p, ..., p] \in \mathbb{R}^d$. There exists a collection of nested intervals $\left[\frac{1}{4}, \frac{3}{4}\right] \supseteq I_1 \supseteq I_2 \supseteq \cdots \supseteq I_{Z_1}$ such that interval $I_m$ corresponds to epoch $m$, with the property that $I_m$ has length $4^{-(m+3)}$, and for every $p \in I_m$, we have*

$$\mathbb{E}_X\left[\left\|\mathcal{A}_t(X) - \xi\mathbf{p}\right\|_2^2\right] \geq \frac{16^{-(m+3)}d\xi^2}{8}$$

*over at least half the rounds $t$ in intervals of epoch $m$.*

By using Lemma 8, there exists a value of $p \in \cap_{m \in [Z_1]} I_m$ such that

$$
\mathbb{E}_{\mathbf{w}_0,\ldots,\mathbf{w}_Z}[R(T,1)] \geq \mathbb{E}_{\mathbf{w}_0,\ldots,\mathbf{w}_Z}\left[\sum_{i=0}^{Z}\sum_{t=c_i+1}^{c_{i+1}} \frac{\mu n}{2}\left\|\mathbf{x}_1(t) - \frac{(n-2K+1)D\mathbf{p}}{n\sqrt{d}}\right\|^2\right]
$$

$$
\geq \mathbb{E}_{\mathbf{w}_0,\ldots,\mathbf{w}_Z}\left[\sum_{i=0}^{Z'}\sum_{t=c_i+1}^{c_{i+1}} \frac{\mu n}{2}\left\|\mathbf{x}_1(t) - \frac{(n-2K+1)D\mathbf{p}}{n\sqrt{d}}\right\|^2\right]
$$

$$
= \mathbb{E}_{\mathbf{w}_0,\ldots,\mathbf{w}_Z}\left[\sum_{m=1}^{Z_1}\sum_{i\in E_m}\sum_{t=c_i+1}^{c_{i+1}} \frac{\mu n}{2}\left\|\mathbf{x}_1(t) - \frac{(n-2K+1)D\mathbf{p}}{n\sqrt{d}}\right\|^2\right]
$$

$$
= \sum_{m=1}^{Z_1}\sum_{i\in E_m}\sum_{t=c_i+1}^{c_{i+1}} \mathbb{E}_X\left[\frac{\mu n}{2}\left\|\mathcal{A}_t(X) - \frac{(n-2K+1)D\mathbf{p}}{n\sqrt{d}}\right\|^2\right]
$$

$$
\geq \sum_{m=1}^{Z_1} \frac{(c_{\frac{1}{15}(16^{m+1}-16)} - c_{\frac{1}{15}(16^m-16)})16^{-(m+3)}\mu(n-2K+1)^2 D^2}{32n}
$$

$$
= \sum_{m=1}^{Z_1} \frac{K_1\mu(n-2K+1)^2 D^2}{16^4(2n)}
$$

$$
= \frac{K_1 Z_1 \mu(n-2K+1)^2 D^2}{16^4(2n)},
$$

where the first inequality is due to (17) and the third equality is due to $c_i = iK_1$. Moreover, we have

$$
\frac{K_1 Z_1 (n-2K+1)^2}{2n} \geq \frac{m(\log_{16}(15Z+16)-2)(n-m-1)^2}{4\omega n}
$$

$$
\geq \frac{(\log_{16}(30\omega(T-1)/n)-2)(n-2)n}{32\omega}.
$$

By using Lemma 7, we can obtain

$$
\mathbb{E}_{\mathbf{w}_0,\ldots,\mathbf{w}_Z}[R(T,1)] \geq \frac{(\log_{16}(30\omega(T-1)/n)-2)(n-2)n\mu D^2}{22^{22}\omega}
$$

$$
\geq \frac{(\log_{16}(30\omega(T-1)/n)-2)(n-2)\pi\mu D^2}{22^{22}\omega\rho^{1/2}}.
$$

### E.5 PROOF OF THEOREM 5

In the one-point bandit feedback, we perform gradient descent on the function $\hat{f}_{t,i}(\mathbf{x}) = \mathbb{E}_{\mathbf{u}_{t,i}\in\mathcal{B}}[f_{t,i}(\mathbf{x}+\epsilon\mathbf{u}_{t,i}]$ over the domain $(1-\zeta)\mathcal{X}$. Since Assumptions 4 and 6 hold, the value of the loss function is bounded. For convenience of the proof, we further assume that the absolute value of all loss functions $f_{t,i}(\cdot)$ over $\mathcal{X}$ is bounded by a constant $V$. According to the one-point gradient estimator Flaxman et al. (2005), we have the following

$$
\left\|\hat{f}_{t,i}(\mathbf{x})\right\|^2 \leq \frac{d^2}{\epsilon^2}(\hat{f}_{t,i}(\mathbf{x}))^2 \leq \frac{d^2 V^2}{\epsilon^2}.
$$

Compared to the proof under the full information setting, the additional error in the bandit setting lies in 2 aspects: (i) the error caused by the gradient estimator. (ii) the error caused by the feasible domain $(1-\zeta)\mathcal{X}$ and the domain $\mathcal{X}$.

We have the following inequality.

$$
\|\mathbf{x}-\mathbf{y}\| \leq 2R, |\hat{f}_{t,i}(\mathbf{x}) - f_{t,i}(\mathbf{x})| \leq G\epsilon.
$$

Then we introduce a lemma to give the error of Algorithm 3.

**Lemma 9** *(Observation 1 in Flaxman et al. (2005)) The optimum in $(1 - \zeta)\mathcal{X}$ is near the optimum in $\mathcal{X}$.*

$$\min_{\mathbf{x} \in (1-\zeta)\mathcal{X}} \sum_{t=1}^{T} \sum_{j=1}^{n} f_{t,j}(\mathbf{x}) \leq 2\zeta V n T + \min_{\mathbf{x} \in \mathcal{X}} \sum_{t=1}^{T} \sum_{j=1}^{n} f_{t,j}(\mathbf{x}).$$

Therefore, we can derive the regret bound in the one-point feedback bandit setting.

$$
\begin{aligned}
\mathbb{E}[R(T, i)] &= \mathbb{E}\left[\sum_{t=1}^{T} \sum_{i=1}^{n} f_{t,j}(\mathbf{x}_{i,1}(t)) - \min_{\mathbf{x} \in \mathcal{X}} \sum_{t=1}^{T} \sum_{j=1}^{n} f_{t,j}(\mathbf{x})\right] \\
&\leq \mathbb{E}\left[\sum_{b=1}^{T/L} \sum_{t=bL+1}^{(b+1)L} \sum_{j=1}^{n} f_{t,j}(\mathbf{x}_i(b) + \epsilon \mathbf{u}_{t,i}) - \min_{\mathbf{x} \in (1-\zeta)\mathcal{X}} \sum_{t=1}^{T} \sum_{j=1}^{n} f_{t,j}(\mathbf{x})\right] + 2\zeta V n T \\
&\leq \mathbb{E}\left[\sum_{b=1}^{T/L} \sum_{t=bL+1}^{(b+1)L} \sum_{j=1}^{n} f_{t,j}(\mathbf{x}_i(b)) - \min_{\mathbf{x} \in (1-\zeta)\mathcal{X}} \sum_{t=1}^{T} \sum_{j=1}^{n} f_{t,j}(\mathbf{x})\right] + 2\zeta V n T + G\epsilon n T \\
&\leq \underbrace{\mathbb{E}\left[\sum_{b=1}^{T/L} \sum_{t=bL+1}^{(b+1)L} \sum_{j=1}^{n} \hat{f}_{t,j}(\mathbf{x}_i(b)) - \min_{\mathbf{x} \in (1-\zeta)\mathcal{X}} \sum_{t=1}^{T} \sum_{j=1}^{n} \hat{f}_{t,j}(\mathbf{x})\right]}_{\alpha} + 2\zeta V n T + 3G\epsilon n T,
\end{aligned}
$$

$$(19)$$

where the first inequality is due to Lemma 9, the second inequality is due to $f_{t,i}(\mathbf{x}_i(b) + \epsilon \mathbf{u}_{t,i}) \leq f_{t,i}(\mathbf{x}_i(b)) + G\epsilon$ and the last inequality is due to $|\hat{f}_{(t,i)}(\mathbf{x}) - f_{t,i}(\mathbf{x})| \leq G\epsilon$.

The term $\alpha$ is the regret of the loss function $\hat{f}_{t,i}(\cdot)$. We can directly use the proof of Theorem 1.

$$\mathbb{E}_{\mathcal{C}}[\alpha] \leq \frac{2nR^2}{\eta} + (20n + 10)LT\eta \frac{d^2 V^2}{\epsilon^2}.$$

Therefore, by setting $\eta = \frac{R\epsilon}{d\sqrt{LT}}, \zeta = \frac{\epsilon}{r}, \epsilon = cd^{1/2}L^{1/4}T^{-1/4}$, where $c$ is a constant such that $\epsilon \leq r$, we can derive the final regret bound

$$
\begin{aligned}
\mathbb{E}_{\mathcal{C}}[R(T, i)] &\leq \frac{2nR^2}{\eta} + (20n + 10)LT\eta \frac{d^2 V^2}{\epsilon^2} + 2\zeta V n T + 3G\epsilon n T \\
&\leq O(nd^{1/2}L^{1/4}T^{3/4}) = O(\omega^{-1/4}\rho^{-1/2}d^{1/2}n(\ln n)^{1/4}T^{3/4}).
\end{aligned}
$$

### E.6 PROOF OF THEOREM 6

As for the strongly convex functions, we can directly apply Lemma 5 to the term $\alpha$ and set $\eta_b = \frac{1}{bL+8L}$, we have

$$
\begin{aligned}
\mathbb{E}_{\mathcal{C}}[\alpha] \leq &\, 2nR^2\left(\frac{1}{\eta_1} - \mu L\right) + 3L^2 n \frac{d^2 V^2}{\epsilon^2} \sum_{b=1}^{T/L} \eta_b + 6\sqrt{3}nL^2 \frac{d^2 V^2}{\epsilon^2} \sum_{b=1}^{T/L} \eta_b \\
&+ (12n + 9)L^2 \frac{d^2 V^2}{\epsilon^2} \sum_{b=1}^{T/L} \eta_b + (n + 12)L^2 \frac{d^2 V^2}{\epsilon^2} \sum_{b=1}^{T/L} \eta_b + (8n + 6)L^2 \frac{d^2 V^2}{\epsilon^2} \sum_{b=1}^{T/L} \eta_{b-1} \\
\leq &\, 16nR^2\mu + \frac{d^2 V^2}{\epsilon^2 \mu}(24n + 6\sqrt{3}n + 27)L \ln(T + 8).
\end{aligned}
$$

$$(20)$$

By combining (19) with (20) and setting $\zeta = \frac{\epsilon}{r}, \epsilon = cd^{2/3}L^{1/3}(\ln(T+8))^{1/3}T^{-1/3}$, where $c$ is a constant such that $\epsilon \le r$, we can obtain

$$
\begin{aligned}
\mathbb{E}_{\mathcal{C}}[R(T,i)] &\le \alpha + 2\zeta VnT + 3G\epsilon nT \\
&\le 16nR^2\mu + \frac{d^2V^2}{\epsilon^2\mu}(24n + 6\sqrt{3}n + 27)L\ln(T+8) + 2\zeta VnT + 3G\epsilon nT \\
&\le O(L^{1/3}d^{2/3}nT^{2/3}\ln(T+8)) \\
&= O(\omega^{-1/3}\rho^{-2/3}d^{2/3}n(\ln n)^{1/3}T^{2/3}(\ln T)^{1/3}).
\end{aligned}
$$

### E.7 PROOF OF THEOREM 7

The proof for the two-point bandit case follows a procedure analogous to that of the one-point case. The guarantees of two-point gradient estimator is

$$
\mathbb{E}\left[\hat{\mathbf{g}}_{t,i}(\mathbf{x})\right] = \nabla f_{t,i}(\mathbf{x}), \|\hat{\mathbf{g}}_{t,i}\|^2 \le d^2G^2.
$$

We have the following

$$
\begin{aligned}
\mathbb{E}_{\mathcal{C}}\left[R_2(T,i)\right] =& \mathbb{E}_{\mathcal{C}}\left[\sum_{b=1}^{T//L}\sum_{t=(b-1)L+1}^{bL}\sum_{j=1}^{n}\frac{f_{t,j}(\mathbf{x}_{i,1}(t)) + f_{t,j}(\mathbf{x}_{i,2}(t))}{2} - \min_{\mathbf{x}\in\mathcal{X}}\sum_{t=1}^{T}\sum_{j=1}^{n}f_{t,j}(\mathbf{x})\right] \\
\le& \mathbb{E}_{\mathcal{C}}\left[\sum_{b=1}^{T//L}\sum_{t=bL+1}^{bL+L}\sum_{j=1}^{n}f_{t,j}(\mathbf{x}_i(b)) - \min_{\mathbf{x}\in\mathcal{X}}\sum_{t=1}^{T}\sum_{j=1}^{n}f_{t,j}(\mathbf{x})\right] + G\epsilon nT \\
\le& \mathbb{E}_{\mathcal{C}}\left[\sum_{b=1}^{T//L}\sum_{t=bL+1}^{bL+L}\sum_{j=1}^{n}f_{t,j}(\mathbf{x}_i(b)) - \min_{\mathbf{x}\in(1-\zeta)\mathcal{X}}\sum_{t=1}^{T}\sum_{j=1}^{n}f_{t,j}(\mathbf{x})\right] + G\epsilon nT + 2\zeta VnT \\
\le& \mathbb{E}_{\mathcal{C}}\left[\alpha\right] + 2\zeta VnT + 3G\epsilon nT,
\end{aligned}
$$

where the first inequality is due to $\sum_{t=(b-1)L+1}^{bL}\frac{f_{t,j}(\mathbf{y}_{i,1}(t)) + f_{t,j}(\mathbf{y}_{i,1}(t))}{2} \le \sum_{t=(b-1)L+1}^{bL}f_{t,j}(\mathbf{x}_i(b)) + G\epsilon L$.

To bound the term $\alpha$, we directly follow the proof of Theorem 5 and replace norm of the gradient with $d^2G^2$. We have

$$
\mathbb{E}_{\mathcal{C}}\left[\alpha\right] \le \frac{2nR^2}{\eta} + (20n + 10)LT\eta d^2G^2.
$$

Therefore, by setting $\eta = \frac{2R}{dG\sqrt{LT}}, \zeta = \frac{\epsilon}{r}, \epsilon = cT^{-1/2}$, where $c$ is a constant such that $\epsilon \le r$, we can derive the final regret bound

$$
\begin{aligned}
\mathbb{E}_{C}[R(T,i)] &\le \frac{2nR^2}{\eta} + (20n + 10)LT\eta d^2G^2 + 2\zeta VnT + 3G\epsilon nT \\
&\le O(ndL^{1/2}T^{1/2}) = O(\omega^{-1/2}\rho^{-1}dn(\ln n)^{1/2}T^{1/2}).
\end{aligned}
$$

### E.8 PROOF OF THEOREM 8

The key difference of this part is to use the strong convexity to derive a tighter bound for $\alpha$. By setting $\eta_b = \frac{1}{bL+8L}$, we have

$$
\mathbb{E}_{\mathcal{C}}\left[\alpha\right] \le 16nR^2\mu + d^2G^2\frac{1}{\mu}(24n + 6\sqrt{3}n + 27)L\ln(T+8). \tag{21}
$$

By combining (19) with (21) and setting $\zeta = \frac{\epsilon}{r}, \epsilon = \frac{c \ln T}{T}$, where $c$ is a constant such that $\epsilon \leq r$, we can obtain

$$
\begin{aligned}
\mathbb{E}_{\mathcal{C}}[R(T, i)] &\leq \alpha + 2\zeta V n T + 3G \epsilon n T \\
&\leq 16 n R^2 \mu + d^2 G^2 \frac{1}{\mu}(24n + 6\sqrt{3}n + 27) L \ln(T + 8) + 2\zeta V n T + 3G \epsilon n T \\
&\leq O(d^2 L n \ln(T + 8)) \\
&= O(\omega^{-1} \rho^{-2} d^2 n \ln n \ln T).
\end{aligned}
$$

# F    PROOF FOR SUPPORTING LEMMAS

## F.1    PROOF OF LEMMA 3

Since each learner $i$ maintains the local auxiliary variable $\hat{\mathbf{x}}_j(b)$ to store the data from the neighbor $j \in \mathcal{N}_i$. The variable $\hat{\mathbf{x}}_i(b)$ is same in all learner $j \in \mathcal{N}_i$. Therefore, we have

$$
\sum_{i=1}^{n} \sum_{j \in \mathcal{N}_i} P_{ij}(\hat{\mathbf{x}}_j(b) - \hat{\mathbf{x}}_i(b)) = \mathbf{0}.
$$

Then we will demonstrate that the average decision $\overline{\mathbf{y}}^k(b)$ is same over $k \in [1, L_1 + 1]$,

$$
\frac{1}{n} \sum_{i=1}^{n} \mathbf{y}_i^{(L_1+1)}(b) = \overline{\mathbf{y}}^{k+1}(b) = \overline{\mathbf{y}}^k(b) + \gamma \frac{1}{n} \sum_{i=1}^{n} \sum_{j \in \mathcal{N}_i} P_{ij}(\hat{\mathbf{y}}_j^k(b) - \hat{\mathbf{y}}_i^b(k)) = \overline{\mathbf{y}}^k(b),
$$

which implies that

$$
\overline{\mathbf{y}}^{(L_1+1)}(b) = \frac{1}{n} \sum_{i=1}^{n} \mathbf{y}_i^{(L_1+1)}(b) = \frac{1}{n} \sum_{i=1}^{n} \mathbf{y}_i^{(1)}(b) = \frac{1}{n} \sum_{i=1}^{n} \mathbf{x}_i(b) = \overline{\mathbf{x}}(b).
$$

We can rewrite that

$$
\begin{aligned}
\overline{\mathbf{x}}(b+1) &= \frac{1}{n} \sum_{i=1}^{n} \tilde{\mathbf{x}}_i(b+1) + \mathbf{r}_i(b+1) = \frac{1}{n} \sum_{i=1}^{n} \mathbf{y}_i^{(L_1+1)}(b) + \frac{1}{n} \sum_{i=1}^{n} \mathbf{r}_i(b+1) \\
&= \frac{1}{n} \sum_{i=1}^{n} \mathbf{y}_i^{(L_1+1)}(b) + \sum_{i=1}^{n} \sum_{j \in \mathcal{N}_i} \gamma P_{ij}(\hat{\mathbf{y}}_j^{(L_1+1)}(b) - \hat{\mathbf{y}}_i^{(L_1+1)}(b)) + \frac{1}{n} \sum_{i=1}^{n} \mathbf{r}_i(b+1) \\
&= \frac{1}{n} \sum_{i=1}^{n} \mathbf{y}_i^{(L_1+1)}(b) + \frac{1}{n} \sum_{i=1}^{n} \mathbf{r}_i(b+1) \\
&= \frac{1}{n} \sum_{i=1}^{n} \mathbf{y}_i^{(1)}(b) + \frac{1}{n} \sum_{i=1}^{n} \mathbf{r}_i(b+1) \\
&= \frac{1}{n} \sum_{i=1}^{n} \mathbf{x}_i(b) - \frac{\eta_b}{n} \sum_{i=1}^{n} \mathbf{z}_i(b-1) + \frac{1}{n} \sum_{i=1}^{n} \mathbf{r}_i(b+1) \\
&= \overline{\mathbf{x}}(b) - \frac{\eta_b}{n} \sum_{i=1}^{n} \mathbf{z}_i(b-1) + \frac{1}{n} \sum_{i=1}^{n} \mathbf{r}_i(b+1).
\end{aligned}
$$

For any $\mathbf{x} \in \mathcal{X}$, we have

$$
\|\overline{\mathbf{x}}(b+1) - \mathbf{x}\|^2 = \|\overline{\mathbf{x}}(b) - \mathbf{x}\|^2 + \frac{1}{n^2} \left\| \sum_{i=1}^{n} \mathbf{r}_i(b+1) - \eta_b \sum_{j=1}^{n} \mathbf{z}_j(b-1) \right\|^2
$$

$$
+ 2\left\langle \frac{1}{n} \sum_{i=1}^{n} \mathbf{r}_i(b+1), \overline{\mathbf{x}}(b) - \mathbf{x} \right\rangle - \frac{2\eta_b}{n} \sum_{j=1}^{n} \langle \mathbf{z}_j(b-1), \overline{\mathbf{x}}(b) - \mathbf{x} \rangle
$$

$$
= \|\overline{\mathbf{x}}(b) - \mathbf{x}\|^2 + \frac{1}{n^2} \left\| \sum_{i=1}^{n} \mathbf{r}_i(b+1) - \eta_b \sum_{j=1}^{n} \mathbf{z}_j(b-1) \right\|^2
$$

$$
+ 2\left\langle \frac{1}{n} \sum_{i=1}^{n} \mathbf{r}_i(b+1), \overline{\mathbf{x}}(b) - \mathbf{x} \right\rangle - \frac{2\eta_b}{n} \sum_{j=1}^{n} \left\langle \sum_{t=(b-2)L+1}^{(b-1)L} \nabla f_{t,j}(\mathbf{x}_j(b-1)), \overline{\mathbf{x}}(b) - \mathbf{x} \right\rangle.
$$

$$(22)$$

For the second term, we have

$$
\frac{1}{n^2} \left\| \sum_{i=1}^{n} \mathbf{r}_i(b+1) - \eta_b \sum_{j=1}^{n} \mathbf{z}_j(b-1) \right\|^2 \leq \frac{2}{n} \|R(b+1)\|_F^2 + 2L^2 \eta_b^2 G^2. \tag{23}
$$

For the third term, we have

$$
2\langle \frac{1}{n} \sum_{i=1}^{n} \mathbf{r}_i(b+1), \overline{\mathbf{x}}(b) - \mathbf{x} \rangle
$$

$$
= \frac{2}{n} \sum_{i=1}^{n} \langle \mathbf{r}_i(b+1), \overline{\mathbf{x}}(b) - \tilde{\mathbf{x}}_i(b+1) + \tilde{\mathbf{x}}_i(b+1) - \mathbf{x} \rangle
$$

$$
= \frac{2}{n} \sum_{i=1}^{n} \langle \mathbf{r}_i(b+1), \overline{\mathbf{x}}(b) - \tilde{\mathbf{x}}_i(b+1) \rangle + \frac{2}{n} \sum_{i=1}^{n} \langle \mathbf{r}_i(b+1), \tilde{\mathbf{x}}_i(b+1) - \mathbf{x} \rangle \tag{24}
$$

$$
= \frac{2}{n} \sum_{i=1}^{n} \langle \mathbf{r}_i(b+1), \overline{\mathbf{x}}(b) - \tilde{\mathbf{x}}_i(b+1) \rangle + \frac{2}{n} \sum_{i=1}^{n} \langle P_{\mathcal{X}}(\tilde{\mathbf{x}}_i(b+1)) - \tilde{\mathbf{x}}_i(b+1), \tilde{\mathbf{x}}_i(b+1) - \mathbf{x} \rangle
$$

$$
\leq \frac{1}{n} \sum_{i=1}^{n} \left( \|\mathbf{r}_i(b+1)\|^2 + \|\overline{\mathbf{x}}(b) - \tilde{\mathbf{x}}_i(b+1)\|^2 \right)
$$

$$
\leq \frac{1}{n} (\|R(b+1)\|_F^2 + \left\| \overline{X}(b) - \tilde{X}(b+1) \right\|_F^2),
$$

where the first inequality is due to $2\langle a, b \rangle \leq \|a\|^2 + \|b\|^2$ and inequality (8).

By using the convexity, we have

$$
f_{t,j}(\mathbf{x}_j(b)) \geq f_{t,j}(\mathbf{x}_i(b)) - G \|\mathbf{x}_i(b) - \mathbf{x}_j(b)\|,
$$

and

$$
-\frac{\eta_b}{n} \sum_{j=1}^{n} \sum_{t=(b-2)L+1}^{(b-1)L} \langle \nabla f_{t,j}(\mathbf{x}_j(b-1)), \overline{\mathbf{x}}(b) - \mathbf{x} \rangle
$$

$$
= -\frac{\eta_b}{n} \sum_{j=1}^{n} \sum_{t=(b-2)L+1}^{(b-1)L} \langle \nabla f_{t,j}(\mathbf{x}_j(b-1)), \overline{\mathbf{x}}(b) - \overline{\mathbf{x}}(b-1) + \overline{\mathbf{x}}(b-1) - \mathbf{x} \rangle
$$

$$
= -\frac{\eta_b}{n} \sum_{j=1}^{n} \sum_{t=(b-2)L+1}^{(b-1)L} \langle \nabla f_{t,j}(\mathbf{x}_j(b-1)), \overline{\mathbf{x}}(b) - \overline{\mathbf{x}}(b-1) \rangle - \frac{\eta_b}{n} \sum_{j=1}^{n} \sum_{t=(b-2)L+1}^{(b-1)L} \langle \nabla f_{t,j}(\mathbf{x}_j(b-1)), \overline{\mathbf{x}}(b-1) - \mathbf{x} \rangle.
$$

Next, we give the bound of these two terms. For the first term, we have

$$
-\frac{\eta_b}{n} \sum_{j=1}^{n} \sum_{t=(b-2)L+1}^{(b-1)L} \langle \nabla f_{t,j}(\mathbf{x}_j(b-1)), \overline{\mathbf{x}}(b) - \overline{\mathbf{x}}(b-1) \rangle
$$

$$
= \langle \frac{\eta_b}{n} \sum_{j=1}^{n} \sum_{t=(b-2)L+1}^{(b-1)L} \nabla f_{t,j}(\mathbf{x}_j(b-1)), \frac{\eta_{b-1}}{n} \sum_{j=1}^{n} \sum_{t=(b-2)L+1}^{(b-1)L} \nabla f_{t,j}(\mathbf{x}_j(b-2)) \rangle
$$

$$
- \langle \frac{\eta_b}{n} \sum_{j=1}^{n} \sum_{t=(b-2)L+1}^{(b-1)L} \nabla f_{t,j}(\mathbf{x}_j(b-1)), \frac{1}{n} \sum_{i=1}^{n} \mathbf{r}_i(b) \rangle
\tag{25}
$$

$$
\leq \left\| \frac{\eta_b}{n} \sum_{j=1}^{n} \sum_{t=(b-2)L+1}^{(b-1)L} \nabla f_{t,j}(\mathbf{x}_j(b-1)) \right\| \left\| \frac{\eta_{b-1}}{n} \sum_{j=1}^{n} \sum_{t=(b-2)L+1}^{(b-1)L} \nabla f_{t,j}(\mathbf{x}_j(b-2)) \right\|
$$

$$
+ \frac{1}{2} \left\| \frac{\eta_b}{n} \sum_{j=1}^{n} \sum_{t=(b-2)L+1}^{(b-1)L} \nabla f_{t,j}(\mathbf{x}_j(b-1)) \right\|^2 + \frac{1}{2} \left\| \frac{1}{n} \sum_{i=1}^{n} \mathbf{r}_i(b) \right\|^2
$$

$$
\leq \eta_b \eta_{b-1} G^2 L^2 + \frac{1}{2} \eta_b^2 G^2 L^2 + \frac{1}{2n} \| R(b) \|_F^2 ,
$$

where the first equality is due to $\overline{\mathbf{x}}(b) = \overline{\mathbf{x}}(b-1) - \frac{\eta_{b-1}}{n} \sum_{j=1}^{n} \sum_{t=(b-2)L+1}^{(b-1)L} \nabla f_{t,j}(\mathbf{x}_j(b-2)) + \frac{1}{n} \sum_{i=1}^{n} \mathbf{r}_i(b)$ and the first inequality is due to $\langle a, b \rangle \leq \|a\| \|b\|$ and $-\langle a, b \rangle \leq \frac{\|a\|^2 + \|b\|^2}{2}$.

For the second term, we have

$$
-\frac{\eta_b}{n} \sum_{j=1}^{n} \sum_{t=(b-2)L+1}^{(b-1)L} \langle \nabla f_{t,j}(\mathbf{x}_j(b-1)), \overline{\mathbf{x}}(b-1) - \mathbf{x} \rangle
$$

$$
= \frac{\eta_b}{n} \sum_{j=1}^{n} \sum_{t=(b-2)L+1}^{(b-1)L} \langle \nabla f_{t,j}(\mathbf{x}_j(b-1)), \mathbf{x} - \overline{\mathbf{x}}(b-1) \rangle
$$

$$
= \frac{\eta_b}{n} \sum_{j=1}^{n} \sum_{t=(b-2)L+1}^{(b-1)L} \langle \nabla f_{t,j}(\mathbf{x}_j(b-1)), \mathbf{x} - \mathbf{x}_j(b-1) \rangle
$$

$$
+ \frac{\eta_b}{n} \sum_{j=1}^{n} \sum_{t=(b-2)L+1}^{(b-1)L} \langle \nabla f_{t,j}(\mathbf{x}_j(b-1)), \mathbf{x}_j(b-1) - \overline{\mathbf{x}}(b-1) \rangle
$$

$$
\leq \frac{\eta_b}{n} \sum_{j=1}^{n} \sum_{t=(b-2)L+1}^{(b-1)L} f_{t,j}(\mathbf{x}) - f_{t,j}(\mathbf{x}_j(b-1)) + \frac{\eta_b}{n} \sum_{j=1}^{n} GL \| \mathbf{x}_j(b-1) - \overline{\mathbf{x}}(b-1) \|
$$

$$
= \frac{\eta_b}{n} \sum_{j=1}^{n} \sum_{t=(b-2)L+1}^{(b-1)L} f_{t,j}(\mathbf{x}) - f_{t,j}(\mathbf{x}_i(b-1)) + f_{t,j}(\mathbf{x}_i(b-1)) - f_{t,j}(\mathbf{x}_j(b-1))
$$

$$
+ \frac{\eta_b}{n} \sum_{j=1}^{n} GL \| \mathbf{x}_j(b-1) - \overline{\mathbf{x}}(b-1) \|
$$

$$
\leq \frac{\eta_b}{n} \sum_{j=1}^{n} \sum_{t=(b-2)L+1}^{(b-1)L} f_{t,j}(\mathbf{x}) - f_{t,j}(\mathbf{x}_i(b-1)) + \frac{\eta_b}{n} GL \sum_{j=1}^{n} \| \mathbf{x}_i(b-1) - \mathbf{x}_j(b-1) \|
$$

$$
+ \frac{\eta_b}{n} GL \sum_{j=1}^{n} \| \mathbf{x}_j(b-1) - \overline{\mathbf{x}}(b-1) \| ,
$$

where the first and the second inequalities are due to the convexity.

By using the fact that

$$\sum_{j=1}^{n} \|\mathbf{x}_i(b-1) - \overline{\mathbf{x}}(b-1)\| \leq \sqrt{n} \left\| X(b-1) - \overline{X}(b-1) \right\|_F,$$

$$\sum_{j=1}^{n} \|\mathbf{x}_i(b-1) - \mathbf{x}_j(b-1)\|$$

$$= \sum_{j=1}^{n} \|\overline{\mathbf{x}}(b-1) - \mathbf{x}_j(b-1)\| + n \|\mathbf{x}_i(b-1) - \overline{\mathbf{x}}(b-1)\|$$

$$\leq \sqrt{n} \left\| X(b-1) - \overline{X}(b-1) \right\|_F + n \left\| X(b-1) - \overline{X}(b-1) \right\|_F,$$

and thus we have

$$-\frac{\eta_b}{n} \sum_{j=1}^{n} \sum_{t=(b-2)L+1}^{(b-1)L} \langle \nabla f_{t,j}(\mathbf{x}_i(b-1)), \overline{\mathbf{x}}(b) - \mathbf{x} \rangle$$

$$\leq \frac{\eta_b}{n} \sum_{j=1}^{n} \sum_{t=(b-2)L+1}^{(b-1)L} f_{t,j}(\mathbf{x}) - f_{t,j}(\mathbf{x}_i(b-1)) + \frac{\eta_b}{n} 3nGL \left\| X(b-1) - \overline{X}(b-1) \right\|_F. \tag{26}$$

By combining (23), (24), (25) and (26), we can derive

$$\|\overline{\mathbf{x}}(b+1) - \mathbf{x}\|^2 = \|\overline{\mathbf{x}}(b) - \mathbf{x}\|^2 + 3L^2\eta_b^2 G^2 + \frac{3}{n} \|R(b+1)\|_F^2 + \frac{1}{n} \left\| X(b) - \tilde{X}(b+1) \right\|_F + 2\eta_b\eta_{b-1}G^2L^2$$

$$+ \frac{1}{n} \|R(b)\|_F^2 + \frac{2\eta_b}{n} \sum_{j=1}^{n} \sum_{t=(b-2)L+1}^{(b-1)L} f_{t,j}(\mathbf{x}) - f_{t,j}(\mathbf{x}_i(b-1)) + \frac{6\eta_b}{n} nGL \left\| X(b-1) - \overline{X}(b-1) \right\|_F,$$

which implies

$$\sum_{t=(b-2)L+1}^{(b-1)L} \sum_{j=1}^{n} f_{t,j}(\mathbf{x}_i(b-1)) - f_{t,j}(\mathbf{x})$$

$$\leq \frac{n}{2\eta_b} (\|\overline{\mathbf{x}}(b) - \mathbf{x}\|^2 - \|\overline{\mathbf{x}}(b+1) - \mathbf{x}\|^2) + \frac{3}{2\eta_b} \|R(b+1)\|_F^2 + \frac{1}{2\eta_b} \|R(b)\|_F^2 + \frac{3}{2} L^2 n\eta_b G^2 + L^2 n\eta_{b-1}G^2$$

$$+ \frac{1}{2\eta_b} \left\| X(b) - \tilde{X}(b+1) \right\|_F^2 + 3nGL \left\| X(b-1) - \overline{X}(b-1) \right\|_F.$$

By summing up over all blocks, we can derive

$$\mathbb{E}_{\mathcal{C}}[R(T,i)] = \sum_{b=1}^{T/L} \sum_{t=(b-1)L+1}^{bL} \sum_{j=1}^{n} f_{t,j}(\mathbf{x}_i(b)) - \sum_{t=1}^{T} \sum_{j=1}^{n} f_{t,j}(\mathbf{x})$$

$$\leq \frac{nD^2}{2\eta_{T/L}} + 3L^2G^2n \sum_{b=1}^{T/L} \eta_b + \sum_{b=1}^{T/L} \frac{3}{2\eta_b} \mathbb{E}_{\mathcal{C}} \left[ \|R(b+1)\|_F^2 \right] + \frac{1}{2\eta_b} \mathbb{E}_{\mathcal{C}} \left[ \left\| X(b) - \tilde{X}(b+1) \right\|_F^2 \right]$$

$$+ 3nGL \sum_{b=1}^{T/L} \mathbb{E}_{\mathcal{C}} \left[ \left\| X(b) - \overline{X}(b) \right\|_F \right] + \sum_{b=1}^{T/L} \frac{1}{2\eta_b} \mathbb{E}_{\mathcal{C}} \left[ \|R(b)\|_F^2 \right]. \tag{27}$$

## F.2 PROOF OF LEMMA 4

Before we give the proof of Lemma 4, we first introduce a lemma to give the guarantee of our online gossip technique.

**Lemma 10** *Given a $\omega$-contractive compressor $\mathcal{C}(\cdot)$ and setting the communication rounds $L_1 = \lceil \frac{28 \ln n}{\gamma \rho} \rceil$ and step size $\gamma = \frac{\omega \rho}{2\rho\beta^2 + 4\beta^2 + (2-\omega)(\beta^2 + 2\beta)\rho + \rho^2}$, we have*

$$e_{L_1+1} \leq \frac{1}{14n} e_1.$$

For projection error $\mathbf{r}_i(b+1)$, since $\mathcal{X}$ is convex, $\overline{\mathbf{x}}(b) = \frac{1}{n}\sum_{i=1}^{n} \mathbf{x}_i(b) \in \mathcal{X}$ and $(1-\gamma)\mathbf{x}_i(b) + \gamma \sum_{j \in \mathcal{N}_i} P_{ij}\mathbf{x}_j(b) \in \mathcal{X}$, for $\gamma \in (0,1]$, we have

$$\begin{aligned}
\|\mathbf{r}_i(b+1)\|^2 &= \|P_{\mathcal{X}}(\tilde{\mathbf{x}}_i(b+1)) - \tilde{\mathbf{x}}_i(b+1)\|^2 \\
&\leq \left\| \overline{\mathbf{x}}(b) - \mathbf{y}_i^{(L_1+1)}(b) \right\|^2 \\
&= \left\| \overline{\mathbf{x}}(b) - \overline{\mathbf{y}}^{(L_1+1)}(b) + \overline{\mathbf{y}}^{(L_1+1)}(b) - \mathbf{y}_i^{(L_1+1)}(b) \right\|^2 \\
&= \left\| \overline{\mathbf{x}}(b) - \overline{\mathbf{y}}^{(1)}(b) + \overline{\mathbf{y}}^{(L_1+1)}(b) - \mathbf{y}_i^{(L_1+1)}(b) \right\|^2 \\
&= \left\| \overline{\mathbf{x}}(b) - (\overline{\mathbf{x}}(b) - \frac{\eta_b}{n}\sum_{i=1}^{n} \mathbf{z}_i(b-1)) + \overline{\mathbf{y}}^{(L_1+1)}(b) - \mathbf{y}_i^{(L_1+1)}(b) \right\|^2 \\
&\leq 2 \left\| \frac{\eta_b}{n}\sum_{i=1}^{n} \mathbf{z}_i(b-1) \right\|^2 + 2 \left\| \overline{\mathbf{y}}^{(L_1+1)}(b) - \mathbf{y}_i^{(L_1+1)}(b) \right\|^2 \\
&\leq 2\eta_b^2 L^2 G^2 + 2 \left\| \overline{\mathbf{y}}^{(L_1+1)}(b) - \mathbf{y}_i^{(L_1+1)}(b) \right\|^2
\end{aligned}$$

where the third equality is due to $\overline{\mathbf{y}}^{(L_1+1)}(b) = \overline{\mathbf{y}}^{(1)}(b)$.

By using Lemma 10, we have

$$\begin{aligned}
\mathbb{E}_{\mathcal{C}}\left[\|R(b+1)\|_F^2\right] &= \mathbb{E}_{\mathcal{C}}\left[\sum_{i=1}^{n} \|\mathbf{r}_i(b)\|^2\right] \\
&\leq 2\sum_{i=1}^{n} \left\| \overline{\mathbf{y}}^{(L_1+1)}(b) - \mathbf{y}_i^{(L_1+1)}(b) \right\|^2 + 2\eta_b^2 n L^2 G^2 \\
&\leq \frac{1}{7n}\mathbb{E}_{\mathcal{C}}\left[\sum_{i=1}^{n} \left\| \overline{\mathbf{y}}^{(1)}(b) - \mathbf{y}_i^{(1)}(b) \right\|^2 + \left\| \hat{\mathbf{y}}^{(1)}(b) - \mathbf{y}_i^{(1)}(b) \right\|^2\right] + 2n n\eta_b^2 L^2 G^2 \\
&= \frac{1}{7n}\mathbb{E}_{\mathcal{C}}\left[\sum_{i=1}^{n} \|\overline{\mathbf{x}}(b) - \eta_b\overline{\mathbf{z}}(b-1) - \mathbf{x}_i(b) + \eta_b\mathbf{z}_i(b-1)\|^2 + \|\hat{\mathbf{x}}(b) - \mathbf{x}_i(b) + \eta_b\mathbf{z}_i(b-1)\|^2\right] \\
&\quad + 2n\eta_b^2 L^2 G^2 \\
&\leq \frac{2}{7n}\mathbb{E}_{\mathcal{C}}\left[\sum_{i=1}^{n} \|\overline{\mathbf{x}}(b) - \mathbf{x}_i(b)\|^2 + \|\hat{\mathbf{x}}(b) - \mathbf{x}_i(b)\|^2\right] + \frac{10}{7}\eta_b^2 L^2 G^2 + 2n\eta_b^2 L^2 G^2.
\end{aligned}$$

For the second term, we have

$$\mathbb{E}_{\mathcal{C}}\left[\|X(b+1) - \overline{X}(b+1)\|_F^2\right] = \mathbb{E}_{\mathcal{C}}\left[\sum_{i=1}^{n} \left\| \mathbf{x}_i(b+1) - \frac{1}{n}\sum_{j=1}^{n} \mathbf{x}_j(b+1) \right\|^2\right].$$

Different from the pervious work that introduces the additional projection error term, we will prove the equality

$$\sum_{i=1}^{n} \left\| \mathbf{x}_i(b+1) - \frac{1}{n}\sum_{j=1}^{n} \mathbf{x}_j(b+1) \right\|^2 = \frac{1}{2n}\sum_{i=1}^{n}\sum_{j=1}^{n} \|\mathbf{x}_i(b+1) - \mathbf{x}_j(b+1)\|^2, \qquad (28)$$

which avoids the incurrence of an additional projection error term.

As for the left term, we have

$$\sum_{i=1}^{n} \left\| \mathbf{x}_i(b+1) - \frac{1}{n} \sum_{j=1}^{n} \mathbf{x}_j(b+1) \right\|^2$$

$$= \sum_{i=1}^{n} \|\mathbf{x}_i(b+1)\|^2 + \left\| \frac{1}{n} \sum_{j=1}^{n} \mathbf{x}_j(b+1) \right\|^2 - 2\langle \mathbf{x}_i(b+1), \frac{1}{n} \sum_{j=1}^{n} \mathbf{x}_j(b+1) \rangle$$

$$= \sum_{i=1}^{n} \|\mathbf{x}_i(b+1)\|^2 + \frac{1}{n} \left\| \sum_{j=1}^{n} \mathbf{x}_j(b+1) \right\|^2 - \frac{2}{n} \langle \sum_{j=1}^{n} \mathbf{x}_j(b+1), \sum_{j=1}^{n} \mathbf{x}_j(b+1) \rangle$$

$$= \sum_{i=1}^{n} \|\mathbf{x}_i(b+1)\|^2 - \frac{1}{n} \left\| \sum_{j=1}^{n} \mathbf{x}_j(b+1) \right\|^2 .$$

For the right term, we have

$$\frac{1}{2n} \sum_{i=1}^{n} \sum_{j=1}^{n} \|\mathbf{x}_i(b+1) - \mathbf{x}_j(b+1)\|^2 = \frac{1}{2n} (2n \sum_{i=1}^{n} \|\mathbf{x}_i(b+1)\|^2 - 2 \sum_{i=1}^{n} \sum_{j=1}^{n} \langle \mathbf{x}_i(b+1), \mathbf{x}_j(b+1) \rangle)$$

$$= \sum_{i=1}^{n} \|\mathbf{x}_i(b+1)\|^2 - \frac{1}{n} \langle \sum_{i=1}^{n} \mathbf{x}_i(b+1), \sum_{j=1}^{n} \mathbf{x}_j(b+1) \rangle)$$

$$= \sum_{i=1}^{n} \|\mathbf{x}_i(b+1)\|^2 - \frac{1}{n} \left\| \sum_{i=1}^{n} \mathbf{x}_i(b+1) \right\|^2 .$$

Therefore we can derive equality (28). By using equality (28), we have

$$\mathbb{E}_{\mathcal{C}} \left[ \left\| X(b+1) - \overline{X}(b+1) \right\|_F^2 \right]$$

$$= \mathbb{E}_{\mathcal{C}} \left[ \sum_{i=1}^{n} \left\| \mathbf{x}_i(b+1) - \frac{1}{n} \sum_{j=1}^{n} \mathbf{x}_j(b+1) \right\|^2 \right]$$

$$= \frac{1}{2n} \sum_{i=1}^{n} \sum_{j=1}^{n} \mathbb{E}_{\mathcal{C}} \left[ \|\mathbf{x}_i(b+1) - \mathbf{x}_j(b+1)\|^2 \right]$$

$$\leq \frac{1}{2n} \sum_{i=1}^{n} \sum_{j=1}^{n} \mathbb{E}_{\mathcal{C}} \left[ \|\tilde{\mathbf{x}}_i(b+1) - \tilde{\mathbf{x}}_j(b+1)\|^2 \right]$$

$$= \sum_{i=1}^{n} \mathbb{E}_{\mathcal{C}} \left[ \left\| \tilde{\mathbf{x}}_i(b+1) - \frac{1}{n} \sum_{j=1}^{n} \tilde{\mathbf{x}}_j(b+1) \right\|^2 \right] .$$

Then we can further derive the upper bound

$$\sum_{i=1}^{n} \mathbb{E}_{\mathcal{C}} \left[ \left\| \tilde{\mathbf{x}}_i(b+1) - \frac{1}{n} \sum_{j=1}^{n} \tilde{\mathbf{x}}_j(b+1) \right\|^2 \right]$$

$$= \sum_{i=1}^{n} \mathbb{E}_{\mathcal{C}} \left[ \left\| \mathbf{y}_i^{(L_1+1)}(b) - \overline{\mathbf{y}}_i^{(L_1+1)}(b) \right\|^2 \right]$$

$$\leq \frac{1}{14n} \mathbb{E}_{\mathcal{C}} \left[ \sum_{i=1}^{n} \left\| \mathbf{y}_i^{(1)}(b) - \overline{\mathbf{y}}_i^{(1)}(b) \right\|^2 \right] + \frac{1}{14n} \mathbb{E}_{\mathcal{C}} \left[ \sum_{i=1}^{n} \left\| \mathbf{y}_i^{(1)}(b) - \hat{\mathbf{y}}_i^{(1)}(b) \right\|^2 \right]$$

$$= \frac{1}{14n} \mathbb{E}_{\mathcal{C}} \left[ \sum_{i=1}^{n} \left\| \mathbf{x}_i(b) - \eta_b \mathbf{z}_i(b-1) - \overline{\mathbf{x}}_i(b) + \eta_b \overline{\mathbf{z}}(b-1) \right\|^2 \right] + \frac{1}{14n} \mathbb{E}_{\mathcal{C}} \left[ \sum_{i=1}^{n} \left\| \mathbf{x}_i(b) - \eta_b \mathbf{z}_i(b-1) - \hat{\mathbf{x}}_i(b) \right\|^2 \right]$$

$$\leq \frac{1}{7n} \mathbb{E}_{\mathcal{C}} \left[ \sum_{i=1}^{n} \left\| \mathbf{x}_i(b) - \overline{\mathbf{x}}_i(b) \right\|^2 + \sum_{i=1}^{n} \left\| \mathbf{x}_i(b) - \hat{\mathbf{x}}_i(b) \right\|^2 \right] + \frac{5}{7} L^2 G^2 \eta_b^2$$

$$\leq \frac{1}{7n} \left( \left\| X(b) - \overline{X}(b) \right\|_F^2 + \left\| X(b) - \hat{X}(b) \right\|_F^2 \right) + \frac{5}{7} L^2 G^2 \eta_b^2,$$

where the second inequality is due to $\|a + b\|^2 \leq 2 \|a\|^2 + 2 \|b\|^2$.

Next, we bound the term $\mathbb{E}_{\mathcal{C}} \left[ \left\| X(b+1) - \hat{X}(b+1) \right\|_F^2 \right]$. As for the repeated compressor, we have $\mathbb{E}_{\mathcal{C}} \left[ \|\mathcal{C}_{L_2}(\mathbf{x}) - \mathbf{x}\|^2 \right] \leq (1-\omega)^{L_2} \|\mathbf{x}\|^2$. By setting $L_2 = \lceil \frac{\ln(8n)}{\omega} \rceil$, we have $(1-\omega)^{L_2} \leq \frac{1}{8n}$, which means $\mathbb{E}_{\mathcal{C}} \left[ \|\mathcal{C}_{L_2}(\mathbf{x}) - \mathbf{x}\|^2 \right] \leq \frac{1}{8n} \|\mathbf{x}\|^2$.

$$\mathbb{E}_{\mathcal{C}} \left[ \left\| X(b+1) - \hat{X}(b+1) \right\|_F^2 \right]$$

$$= \sum_{i=1}^{n} \mathbb{E}_{\mathcal{C}} \left[ \left\| \mathbf{x}_i(b+1) - \hat{\mathbf{x}}_i(b+1) \right\|^2 \right]$$

$$= \sum_{i=1}^{n} \mathbb{E}_{\mathcal{C}} \left[ \left\| \mathbf{y}_i^{(L_1+1)}(b) + \mathbf{r}_i(b+1) - \hat{\mathbf{y}}_i^{(L_1+1)}(b) - \mathbf{r}_i^{\mathcal{C}}(b+1) \right\|^2 \right]$$

$$= 2 \sum_{i=1}^{n} \mathbb{E}_{\mathcal{C}} \left[ \left\| \mathbf{y}_i^{(L_1+1)}(b) - \hat{\mathbf{y}}_i^{(L_1+1)}(b) \right\|^2 \right] + 2 \mathbb{E}_{\mathcal{C}} \left[ \sum_{i=1}^{n} \left\| \mathbf{r}_i(b+1) - \mathbf{r}_i^{\mathcal{C}}(b+1) \right\|^2 \right]$$

$$\leq \frac{1}{7n} \mathbb{E}_{\mathcal{C}} \left[ \sum_{i=1}^{n} \left\| \mathbf{y}_i^{(1)}(b) - \hat{\mathbf{y}}_i^{(1)}(b) \right\|^2 + \left\| \mathbf{y}_i^{(1)}(b) - \overline{\mathbf{y}}_i^{(1)}(b) \right\|^2 \right] + \frac{1}{4n} \mathbb{E}_{\mathcal{C}} \left[ \sum_{i=1}^{n} \left\| \mathbf{r}_i(b+1) \right\|^2 \right]$$

$$= \frac{1}{7n} \mathbb{E}_{\mathcal{C}} \left[ \sum_{i=1}^{n} \left\| \mathbf{x}_i(b) - \eta_b \mathbf{z}_i(b-1) - \hat{\mathbf{x}}_i(b) \right\|^2 + \left\| \mathbf{x}_i(b) - \eta_b \mathbf{z}_i(b-1) - \overline{\mathbf{x}}(b) + \eta_b \overline{\mathbf{z}}(b-1) \right\|^2 \right]$$

$$\quad + \frac{1}{4n} \mathbb{E}_{\mathcal{C}} \left[ \sum_{i=1}^{n} \left\| \mathbf{r}_i(b+1) \right\|^2 \right]$$

$$\leq \frac{2}{7n} \mathbb{E}_{\mathcal{C}} \left[ \sum_{i=1}^{n} \left\| \mathbf{x}_i(b) - \hat{\mathbf{x}}_i(b) \right\|^2 + \left\| \mathbf{x}_i(b) - \overline{\mathbf{x}}(b) \right\|^2 \right] + \frac{1}{4n} \mathbb{E}_{\mathcal{C}} \left[ \sum_{i=1}^{n} \left\| R(b+1) \right\|_F^2 \right] + \frac{5}{7} L^2 G^2 \eta_b^2$$

$$\leq \frac{5}{14n} \mathbb{E}_{\mathcal{C}} \left[ \sum_{i=1}^{n} \left\| \mathbf{x}_i(b) - \hat{\mathbf{x}}_i(b) \right\|^2 + \left\| \mathbf{x}_i(b) - \overline{\mathbf{x}}(b) \right\|^2 \right] + 2 L^2 G^2 \eta_b^2.$$

### F.3 PROOF OF LEMMA 5

The proof is similar to Lemma 3, the key difference is that we need to utilize the strong convexity. According to the proof of Lemma 3, we first have the following

$$\|\overline{\mathbf{x}}(b+1) - \mathbf{x}\|^2 = \left\| \frac{1}{n} \sum_{i=1}^{n} \overline{\mathbf{x}}(b) - \mathbf{x} \right\|^2 + \frac{1}{n^2} \left\| \sum_{i=1}^{n} \mathbf{r}_i(b+1) - \eta_b \sum_{j=1}^{n} \mathbf{z}_j(b-1) \right\|^2$$

$$+ 2 \left\langle \frac{1}{n} \sum_{i=1}^{n} \mathbf{r}_i(b+1), \overline{\mathbf{x}}(b) - \mathbf{x} \right\rangle - \frac{2\eta_b}{n} \sum_{j=1}^{n} \left\langle \sum_{t=(b-2)L+1}^{(b-1)L} \nabla f_{t,j}(\mathbf{x}_j(b-1)), \overline{\mathbf{x}}(b) - \mathbf{x} \right\rangle.$$

For the last term, we have

$$-\frac{\eta_b}{n} \sum_{j=1}^{n} \sum_{t=(b-2)L+1}^{(b-1)L} \langle \nabla f_{t,j}(\mathbf{x}_j(b-1)), \overline{\mathbf{x}}(b) - \mathbf{x} \rangle$$

$$= -\frac{\eta_b}{n} \sum_{j=1}^{n} \sum_{t=(b-2)L+1}^{(b-1)L} \langle \nabla f_{t,j}(\mathbf{x}_j(b-1)), \overline{\mathbf{x}}(b) - \overline{\mathbf{x}}(b-1) + \overline{\mathbf{x}}(b-1) - \mathbf{x} \rangle$$

$$= -\frac{\eta_b}{n} \sum_{j=1}^{n} \sum_{t=(b-2)L+1}^{(b-1)L} \langle \nabla f_{t,j}(\mathbf{x}_j(b-1)), \overline{\mathbf{x}}(b) - \overline{\mathbf{x}}(b-1) \rangle - \frac{\eta_b}{n} \sum_{j=1}^{n} \sum_{t=(b-2)L+1}^{(b-1)L} \langle \nabla f_{t,j}(\mathbf{x}_j(b-1)), \overline{\mathbf{x}}(b-1) - \mathbf{x} \rangle.$$

For the first term, we can directly use (25). For the second term, we have

$$-\frac{\eta_b}{n} \sum_{j=1}^{n} \sum_{t=(b-2)L+1}^{(b-1)L} \langle \nabla f_{t,j}(\mathbf{x}_j(b-1)), \overline{\mathbf{x}}(b-1) - \mathbf{x} \rangle$$

$$= \frac{\eta_b}{n} \sum_{j=1}^{n} \sum_{t=(b-2)L+1}^{(b-1)L} \langle \nabla f_{t,j}(\mathbf{x}_j(b-1)), \mathbf{x} - \overline{\mathbf{x}}(b-1) \rangle$$

$$= \frac{\eta_b}{n} \sum_{j=1}^{n} \sum_{t=(b-2)L+1}^{(b-1)L} \langle \nabla f_{t,j}(\mathbf{x}_j(b-1)), \mathbf{x} - \mathbf{x}_j(b-1) \rangle + \frac{\eta_b}{n} \sum_{j=1}^{n} \sum_{t=(b-2)L+1}^{(b-1)L} \langle \nabla f_{t,j}(\mathbf{x}_j(b-1)), \mathbf{x}_j(b-1) - \overline{\mathbf{x}}(b-1) \rangle$$

$$\leq \frac{\eta_b}{n} \sum_{j=1}^{n} \sum_{t=(b-2)L+1}^{(b-1)L} f_{t,j}(\mathbf{x}) - f_{t,j}(\mathbf{x}_j(b-1)) - \frac{\mu}{2} \|\mathbf{x} - \mathbf{x}_j(b-1)\|^2 + \frac{\eta_b}{n} \sum_{j=1}^{n} GL \|\mathbf{x}_j(b-1) - \overline{\mathbf{x}}(b-1)\|$$

$$= \frac{\eta_b}{n} \sum_{j=1}^{n} \sum_{t=(b-2)L+1}^{(b-1)L} f_{t,j}(\mathbf{x}) - f_{t,j}(\mathbf{x}_i(b-1)) + f_{t,j}(\mathbf{x}_i(b-1)) - f_{t,j}(\mathbf{x}_j(b-1))$$

$$+ \frac{\eta_b}{n} \sum_{j=1}^{n} GL \|\mathbf{x}_j(b-1) - \overline{\mathbf{x}}(b-1)\| - \frac{\mu L}{2} \|\mathbf{x} - \mathbf{x}_j(b-1)\|^2$$

$$\leq \frac{\eta_b}{n} \sum_{j=1}^{n} \sum_{t=(b-2)L+1}^{(b-1)L} f_{t,j}(\mathbf{x}) - f_{t,j}(\mathbf{x}_i(b-1)) + \frac{\eta_b}{n} GL \sum_{j=1}^{n} \|\mathbf{x}_i(b-1) - \mathbf{x}_j(b-1)\|$$

$$\frac{\eta_b}{n} \left( GL \sum_{j=1}^{n} \|\mathbf{x}_j(b-1) - \overline{\mathbf{x}}(b-1)\| - \frac{\mu L}{2} \|\mathbf{x} - \mathbf{x}_j(b-1)\|^2 \right),$$

where the first inequality is due to the strong convexity. By using the fact that

$$\sum_{j=1}^{n} \|\mathbf{x} - \mathbf{x}_j(b-1)\|^2 \geq \frac{1}{n} \left\| \sum_{j=1}^{n} \mathbf{x} - \mathbf{x}_j(b-1) \right\|^2 \geq \frac{1}{n} \|n\mathbf{x} - n\overline{\mathbf{x}}(b-1)\|^2 \geq n \|\mathbf{x} - \overline{\mathbf{x}}(b-1)\|^2,$$

and we have

$$- \frac{\eta_b}{n} \sum_{j=1}^{n} \sum_{t=(b-2)L+1}^{(b-1)L} \langle \nabla f_{t,j}(\mathbf{x}_j(b-1)), \overline{\mathbf{x}}(b-1) - \mathbf{x} \rangle$$

$$\leq \frac{\eta_b}{n} \sum_{j=1}^{n} \sum_{t=(b-2)L+1}^{(b-1)L} f_{t,j}(\mathbf{x}) - f_{t,j}(\mathbf{x}_i(b-1)) + \frac{\eta_b}{n} GL \sum_{j=1}^{n} \|\mathbf{x}_i(b-1) - \mathbf{x}_j(b-1)\|$$

$$\frac{\eta_b}{n} GL \sum_{j=1}^{n} \|\mathbf{x}_j(b-1) - \overline{\mathbf{x}}(b-1)\| - \frac{\mu L}{2} \|\mathbf{x} - \mathbf{x}_j(b-1)\|^2$$

$$\leq \frac{\eta_b}{n} \sum_{j=1}^{n} \sum_{t=(b-2)L+1}^{(b-1)L} f_{t,j}(\mathbf{x}) - f_{t,j}(\mathbf{x}_i(b-1)) + \frac{\eta_b}{n} 3nGL \left\| X(b-1) - \overline{X}(b-1) \right\|_F - \frac{\eta_b \mu L}{2} \|\mathbf{x} - \overline{\mathbf{x}}(b-1)\|^2 .$$

(29)

By combining (23), (24), (25) and (29), we can derive

$$\|\overline{\mathbf{x}}(b+1) - \mathbf{x}\|^2 = \|\overline{\mathbf{x}}(b) - \mathbf{x}\|^2 - \eta_b \mu L \|\mathbf{x} - \overline{\mathbf{x}}(b-1)\|^2 + 3L^2 \eta_b^2 G^2 + 2\eta_b \eta_{b-1} L^2 G^2$$

$$+ \frac{3}{n} \|R(b+1)\|_F^2 + \frac{1}{n} \left\| X(b) - \tilde{X}(b+1) \right\|_F + \frac{1}{n} \|R(b)\|_F^2$$

$$+ \frac{2\eta_b}{n} \sum_{j=1}^{n} \sum_{t=(b-2)L+1}^{(b-1)L} f_{t,j}(\mathbf{x}) - f_{t,j}(\mathbf{x}_i(b-1)) + \frac{2\eta_b}{n} 3nGL \left\| X(b-1) - \overline{X}(b-1) \right\|_F ,$$

which implies

$$\sum_{t=(b-2)L+1}^{(b-1)L} \sum_{j=1}^{n} f_{t,j}(\mathbf{x}_i(b-1)) - f_{t,j}(\mathbf{x})$$

$$\leq \frac{n}{2\eta_b} (\|\overline{\mathbf{x}}(b) - \mathbf{x}\|^2 - \|\overline{\mathbf{x}}(b+1) - \mathbf{x}\|^2) - \frac{n\mu L}{2\eta_b} \|\mathbf{x} - \overline{\mathbf{x}}(b-1)\|^2 + \frac{3}{2\eta_b} \|R(b+1)\|_F^2 + \frac{3}{2} L^2 n\eta_b G^2 + L^2 n\eta_{b-1} G^2$$

$$+ \frac{1}{2\eta_b} \left\| X(b) - \tilde{X}(b+1) \right\|_F^2 + 3nGL \left\| X(b-1) - \overline{X}(b-1) \right\|_F + \frac{1}{2\eta_b} \|R(b)\|_F^2 .$$

By summing up over all blocks, we can derive

$$\mathbb{E}_{\mathcal{C}}[R(T,i)] = \sum_{b=1}^{T/L} \sum_{t=(b-1)L+1}^{bL} \sum_{j=1}^{n} f_{t,j}(\mathbf{x}_i(b)) - \sum_{t=1}^{T} \sum_{j=1}^{n} f_{t,j}(\mathbf{x})$$

$$\leq \frac{nD^2}{2} \sum_{b=1}^{T/L} (\frac{1}{\eta_b} - \frac{1}{\eta_{b-1}} - \mu L) + 3L^2 G^2 n \sum_{b=1}^{T/L} \eta_b + 3nGL \sum_{b=1}^{T/L} \mathbb{E}_{\mathcal{C}} \left[ \left\| X(b) - \overline{X}(b) \right\|_F \right]$$

$$+ \sum_{b=1}^{T/L} \frac{3}{2\eta_b} \mathbb{E}_{\mathcal{C}} \left[ \|R(b+1)\|_F^2 \right] + \frac{1}{2\eta_b} \mathbb{E}_{\mathcal{C}} \left[ \left\| X(b) - \tilde{X}(b+1) \right\|_F^2 \right] + \frac{1}{2\eta_b} \mathbb{E}_{\mathcal{C}} \left[ \|R(b)\|_F^2 \right] .$$

(30)

## F.4    PROOF OF LEMMA 10

The efficient implementation of Choco-gossip is summarized in Algorithm 5, where each learner $i$ only needs to maintain three additional variables.

In the following, we give the proof for Lemma 10. First, we provide its matrix version of Choco-gossip in Algorithm 6 to simplify our proof. The proof of this lemma is based on the analysis of Koloskova et al. (2019). The key difference is that we choose a different $\gamma$ to obtain a tighter guarantee. We introduce the following lemma

---

**Algorithm 5** Efficient Choco-gossip

---
1: **Input:** communication round $L, \mathbf{x}_i(1) \in \mathbb{R}^d$ for $i \in [n], \hat{\mathbf{x}}_i(1) = \mathbf{0}$ for $i \in [n]$
2: **for** learner $i \in [n]$ **do**
3:     **for** $t = 1$ to $L$ **do**
4:         $\mathbf{x}_i(t+1) = \mathbf{x}_i(k) + \gamma\left(\mathbf{s}_i(t) - \hat{\mathbf{x}}_i(t)\right)$
5:         $\mathbf{q}_i(t) = \mathcal{C}(\mathbf{x}_i(t+1) - \hat{\mathbf{x}}_i(t))$
6:         Send $\mathbf{q}_i(t)$ and receive $\mathbf{q}_j(t)$
7:         $\hat{\mathbf{x}}_i(t+1) = \hat{\mathbf{x}}_i(t) + \mathbf{q}_i(t)$
8:         $\mathbf{s}_i(t+1) = \mathbf{s}_i(t) + \sum_{j \in \mathcal{N}_i} P_{ij}\mathbf{q}_j(t)$
9:     **end for**
10: **end for**

---

**Algorithm 6** Choco-gossip

---
1: **Input:** Communication round $L, \mathbf{x}_i(1) \in \mathbb{R}^d$ for $i \in [n], \hat{\mathbf{x}}_i(1) = \mathbf{0}$ for $i \in [n]$
2: **for** learner $i \in [n]$ **do**
3:     **for** $t = 1$ to $L$ **do**
4:         $X(t+1) = X(t) + \gamma \hat{X}(t)(P - I)$
5:         $Q(t) = \mathcal{C}(X(t+1) - \hat{X}(t))$
6:         $\hat{X}_i(t+1) = \hat{X}(t) + Q(t)$
7:     **end for**
8: **end for**

---

**Lemma 11** *(Lemma 16 in Koloskova et al. (2019)) For $P$ satisfying Assumption 1 and $t \in \mathbb{N}_+$, we have*

$$\left\|P^t - \frac{1}{n}\mathbf{1}\mathbf{1}^\top\right\|_2 \leq (1 - \rho)^t.$$

Since the variable $\hat{\mathbf{x}}_i(t)$ is same in all neighbors $j \in \mathcal{N}_i$, we have $\sum_{i=1}^n \sum_{j \in \mathcal{N}_i} P_{ij}(\hat{\mathbf{x}}_j(t) - \hat{\mathbf{x}}_i(t)) = \mathbf{0}$. During iterates of the Algorithm 6, we can derive

$$\overline{\mathbf{x}}(t+1) = \overline{\mathbf{x}}(t) + \gamma\frac{1}{n}\sum_{i=1}^n \sum_{j \in \mathcal{N}_i} P_{ij}(\hat{\mathbf{x}}_j(t) - \hat{\mathbf{x}}_i(t)) = \overline{\mathbf{x}}(t),$$

which means the average decision is same over all rounds. We denote $\overline{X} = \overline{X}(1) = \cdots = \overline{X}(L_1)$ and can derive the following

$$
\begin{aligned}
\left\|X(t+1) - \overline{X}\right\|_F^2 &= \left\|X(t) - \overline{X} + \gamma \hat{X}(t)(P - I)\right\|_F^2 \\
&= \left\|X(t) - \overline{X} + \gamma\left(X(t) - \overline{X}\right)(P - I) + \gamma\left(\hat{X}(t) - X(t)\right)(P - I)\right\|_F^2 \\
&= \left\|\left(X(t) - \overline{X}\right)\left((1 - \gamma)I + \gamma P\right) + \gamma\left(\hat{X}(t) - X(t)\right)(P - I)\right\|_F^2 \\
&\leq (1 + \frac{\gamma\rho}{2})\left\|\left(X(t) - \overline{X}\right)\left((1 - \gamma)I + \gamma P\right)\right\|_F^2 + (1 + \frac{2}{\gamma\rho})\left\|\gamma\left(\hat{X}(t) - X(t)\right)(P - I)\right\|_F^2 \\
&\leq (1 + \frac{\gamma\rho}{2})\left\|\left(X(t) - \overline{X}\right)\left((1 - \gamma)I + \gamma P\right)\right\|_F^2 + (1 + \frac{2}{\gamma\rho})\gamma^2 \|P - I\|_2^2 \left\|\hat{X}(t) - X(t)\right\|_F^2 \\
&\leq (1 + \frac{\gamma\rho}{2})\left\|\left(X(t) - \overline{X}\right)\left((1 - \gamma)I + \gamma P\right)\right\|_F^2 + (1 + \frac{2}{\gamma\rho})\gamma^2 \|P - I\|_2^2 \left\|\hat{X}(t) - X(t)\right\|_F^2
\end{aligned}
$$

where the second equality is due to $\overline{X}(P - I) = 0$.

---

**Algorithm 7** Choco-gossip for matrix

---

1: **Input:** gossip round $L$, $\mathbf{x}_i(1) \in \mathbb{R}^d$ for $i \in [n]$, $\hat{\mathbf{x}}_i(1) = \mathbf{0}$ for $i \in [n]$
2: **for** learner $i \in [n]$ **do**
3:    **for** $t = 1$ to $L$ **do**
4:       Compute $\mathbf{x}_i(t+1) = \mathbf{x}_i(k) + \gamma \sum_{j \in \mathcal{N}_i} P_{ij}(\hat{\mathbf{x}}_j(t) - \hat{\mathbf{x}}_i(t))$
5:       Compute $\mathbf{q}_i(t) = \mathcal{C}(\mathbf{x}_i(t+1) - \hat{\mathbf{x}}_i(t))$
6:       **for** neighbors $j \in \mathcal{N}_i$ **do**
7:         Send $\mathbf{q}_i(t)$ and receive $\mathbf{q}_j(t)$
8:         Compute $\hat{\mathbf{x}}_j(t+1) = \hat{\mathbf{x}}_j(t) + \mathbf{q}_j(t)$
9:       **end for**
10:   **end for**
11: **end for**

---

As for the first term, we have

$$
\left\| (X(t) - \overline{X}) ((1-\gamma)I + \gamma P) \right\|_F \le 1(1-\gamma) \left\| X(t) - \overline{X} \right\|_F + \gamma \left\| (X(t) - \overline{X}) P \right\|_F
$$
$$
= (1-\gamma) \left\| X(t) - \overline{X} \right\|_F + \gamma \left\| (X(t) - \overline{X}) (P - \mathbf{1}\mathbf{1}^\top/n) \right\|_F
$$
$$
= (1-\gamma) \left\| X(t) - \overline{X} \right\|_F + \gamma \left\| P (X(t) - \overline{X}) \right\|_F
$$
$$
\le (1 - \gamma\rho) \left\| X(t) - \overline{X} \right\|_F ,
$$

where the second equality is due to $(X(t) - \overline{X}) \mathbf{1}\mathbf{1}^\top/n = 0$.

Therefore, we have

$$
\left\| X(t+1) - \overline{X} \right\|_F^2 \le (1 + \frac{\gamma\rho}{2})(1-\gamma\rho)^2 \left\| X(t) - \overline{X} \right\|_F + (1 + \frac{2}{\gamma\rho})\gamma^2\beta^2 \left\| \hat{X}(t+1) - X(t) \right\|_F^2 .
$$

$$
\mathbb{E}_{\mathcal{C}} \left[ \left\| X(t+1) - \hat{X}(t+1) \right\|_F^2 \right]
$$
$$
= \mathbb{E}_{\mathcal{C}} \left[ \left\| X(t+1) - \hat{X}(t) - \mathcal{C}(X(t+1) - \hat{X}(t)) \right\|_F^2 \right]
$$
$$
\le (1 - \omega) \mathbb{E}_{\mathcal{C}} \left[ \left\| X(t+1) - \hat{X}(t) \right\|_F^2 \right] .
$$

Then we give the bound of the other term.

$$
\mathbb{E}_{\mathcal{C}} \left[ \left\| X(t+1) - \hat{X}(t+1) \right\|_F^2 \right]
$$
$$
\le (1-\omega) \mathbb{E}_{\mathcal{C}} \left[ \left\| X(t+1) - \hat{X}(t) \right\|_F^2 \right]
$$
$$
= (1-\omega) \mathbb{E}_{\mathcal{C}} \left[ \left\| X(t) + \gamma\hat{X}(t)(P - I) - \hat{X}(t) \right\|_F^2 \right]
$$
$$
= (1-\omega) \mathbb{E}_{\mathcal{C}} \left[ \left\| (X(t) - \hat{X}(t)) ((1+\gamma)I - \gamma P) + \gamma(X(t) - \overline{X})(P - I) \right\|_F^2 \right]
$$
$$
\le (1 + \frac{\omega}{2})(1-\omega) \mathbb{E}_{\mathcal{C}} \left[ \left\| (X(t) - \hat{X}(t)) ((1+\gamma)I - \gamma P) \right\|_F^2 \right]
$$
$$
+ (1 + \frac{2}{\omega})(1-\omega) \mathbb{E}_{\mathcal{C}} \left[ \left\| \gamma(X(t) - \overline{X})(P - I) \right\|_F^2 \right]
$$
$$
\le (1 + \frac{\omega}{2})(1-\omega)(1+\gamma\beta)^2 \mathbb{E}_{\mathcal{C}} \left[ \left\| X(t) - \hat{X}(t) \right\|_F^2 \right] + (1 + \frac{2}{\omega})(1-\omega)\gamma^2\beta^2 \mathbb{E}_{\mathcal{C}} \left[ \left\| X(t) - \overline{X} \right\|_F^2 \right] .
$$

We define

$$e_{t+1} = \mathbb{E}_{\mathcal{C}}\left[\left\|X(t+1) - \hat{X}(t+1)\right\|_F^2 + \left\|X(t+1) - \overline{X}\right\|_F^2\right]$$

$$= \mathbb{E}_{\mathcal{C}}\left[\sum_{i=1}^n \|\mathbf{x}_i(t+1) - \hat{\mathbf{x}}_i(t+1)\|^2 + \|\mathbf{x}_i(t+1) - \overline{\mathbf{x}}\|^2\right].$$

We further have

$$e_{t+1} \leq \max\{(1 + \frac{\gamma\rho}{2})(1 - \gamma\rho)^2 + (1 - \omega)(1 + \frac{2}{\omega})\gamma^2\beta^2, (1 + \frac{2}{\gamma\rho})\gamma^2\beta^2 + (1 - \omega)(1 + \frac{\omega}{2})(1 + \gamma\beta)^2\}e_t.$$

We want to select a appropriate $\gamma$, which satisfies

$$e_{t+1} \leq (1 - \frac{\rho\gamma}{2})e_t.$$

We have to ensure

$$(1 + \frac{\gamma\rho}{2})(1 - \gamma\rho)^2 + (1 - \omega)(1 + \frac{2}{\omega})\gamma^2\beta^2 \leq 1 - \frac{\rho}{2}\gamma, \tag{31}$$

$$(1 + \frac{2}{\gamma\rho})\gamma^2\beta^2 + (1 - \omega)(1 + \frac{\omega}{2})(1 + \gamma\beta)^2 \leq 1 - \frac{\rho}{2}\gamma. \tag{32}$$

According to inequality (31), we have

$$\gamma \leq \frac{2\omega\rho}{8\beta^2 + \omega\rho^2}.$$

According to inequality (32), we have

$$\gamma \leq \frac{\omega\rho}{2\rho\beta^2 + 4\beta^2 + (2 - \omega)(\beta^2 + 2\beta)\rho + \rho^2}.$$

Therefore, we choose $\gamma = \frac{\omega\rho}{2\rho\beta^2 + 4\beta^2 + (2-\omega)(\beta^2 + 2\beta)\rho + \rho^2}$, we have

$$e_{t+1} \leq (1 - \frac{\gamma\rho}{2})e_t \leq (1 - \frac{\gamma\rho}{2})^t e_1.$$

By setting block size $L = \lceil \frac{28 \ln n}{\gamma\rho} \rceil$, we have $e_{L+1} \leq (1 - \frac{\gamma\rho}{2})^{\lceil \frac{28 \ln n}{\gamma\rho} \rceil} \leq \frac{1}{14n}e_1$.

### F.5 PROOF OF LEMMA 6

We prove this lemma by Induction.

(i) When $b = 1$, this inequality holds. Suppose that the statement holds for $k$. Then for $k + 1$,

$$e_{k+2} \leq \frac{1}{2n}e_{k+1} + q\eta_{k+1}^2 L^2$$

$$\leq \frac{2}{n}qL^2\eta_k^2 L^2 + q\eta_{k+1}^2 L^2.$$

Then we need to prove that

$$\frac{2}{n}q\eta_k^2 L^2 + q\eta_{k+1}^2 L^2 \leq 3qL^2\eta_{k+1}^2,$$

which is equal to prove

$$\frac{\eta_{k+1}^2}{\eta_k^2} \leq 1.$$

As $\eta_{k+1} \leq \eta_k$, this ineqlity holds. We finish the proof.

## G   THE CHOICE OF PARAMETER IN PREVIOUS WORK

In the proof of Tu et al. (2022), they need to minimized the term

$$e_t = \mathbb{E}_{\mathcal{C}}[\|X(t+1) - \overline{X}(t+1)\|_F^2] + \mathbb{E}_{\mathcal{C}}[\|X(t+1) - \hat{X}(t+1)\|_F^2].$$

According to the proof of Tu et al. (2022), they have

$$e_{t+1} \leq \|U(\gamma)\| \, e_t + C_3 \gamma^{-1} \rho^{-1} n \eta_t^2$$

$$\leq \lambda_{\max}(U(\gamma)) e_t + C_3 \gamma^{-1} \rho^{-1} n \eta_t^2,$$

where $C_3$ is a constant and

$$U(\gamma) = \begin{bmatrix} 1 - \rho\gamma & u_1\gamma \\ u_2\gamma^2 & 1 - \frac{\omega}{2} - \frac{\omega^2}{2} + u_3\gamma \end{bmatrix},$$

and $u_1 = 9\left(1 + \frac{2}{\rho}\right)(1 - \omega)\beta^2, u_2 = 3\left(1 + \frac{2}{\omega}\right)\beta^2, u_3 = \left(1 + \frac{\omega}{2}\right)(1 - \omega)(\beta^2 + 2\beta) + 6\left(1 + \frac{2}{\omega}\right)(1 - \omega)\beta^2.$

However, their use of the inequality is incorrect due to $\lambda_{\max}(U(\gamma)) \leq \|U(\gamma)\|$.

We give a correct proof here.

We have $e_t = (1 - \rho\gamma + u_2\gamma^2)\mathbb{E}_{\mathcal{C}}[\|X(t+1) - \overline{X}(t+1)\|_F^2] + (u_1\gamma + u_3\gamma + 1 - \frac{\omega}{2} - \frac{\omega^2}{2})\mathbb{E}_{\mathcal{C}}[\|X(t+1) - \hat{X}(t+1)\|_F^2] + C_3\gamma^{-1}\rho^{-1}n\eta_t^2.$

We need to choose $\gamma$ that ensures $\max\{(1 - \rho\gamma + u_2\gamma), (u_1\gamma + u_3\gamma + 1 - \frac{\omega}{2} - \frac{\omega^2}{2})\} \leq 1 - \frac{3}{4}\gamma\rho$, which means

$$1 - \rho\gamma + u_2\gamma^2 \leq 1 - \frac{3}{4}\gamma\rho,$$

$$u_1\gamma + u_3\gamma + 1 - \frac{\omega}{2} - \frac{\omega^2}{2} \leq 1 - \frac{3}{4}\gamma\rho.$$

We have

$$\gamma \leq \frac{\rho}{4u_2},$$

$$\gamma \leq \frac{\omega + \omega^2}{2(\rho + u_1 + u_3)}.$$

Therefore, we can have $\gamma \leq \frac{\rho(\omega^2 + \omega)}{2(\rho + u_1 + 4u_2 + u_3)}$, because $\omega^2 + \omega \leq 2$.

We choose $\gamma = \frac{3\rho^3\omega^2(\omega+1)}{2\rho^2\omega + 9\beta^2(\rho+2)(\omega-\omega^2) + 24\beta^2(\omega+2) + \omega(\omega+2)(1-\omega)(\beta^2+2\beta) + 12\beta^2(\omega+2)(1-\omega)} < 1.$

And we have $\gamma^{-1} \leq O(\omega^{-2}\rho^{-3})$, which is on the same order with the result in Tu et al. (2022).

## H   PROOF FOR ADDITIONAL DISCUSSIONS

(i) $L_1 = 1$. It is not hard to verify that, when $L_1 = 1$, the sum of the consensus error and the compression error is on the same order as Tu et al. (2022). Although we can reduce the projection error to $O(1)$, the consensus error is still the same as Tang et al. (2018), which is the leading term in the final regret. Thus, it does not helps to improve the existing regret bounds.

(ii) $L_2 = 0$. When $L_2 = 0$, the upper bound of term $\|\mathbf{x}_i(b) - \hat{\mathbf{x}}_i(b)\|^2$ contains an additional projection error $\|\mathbf{r}_i(b)\|^2$ of the order $O(1)$, which consequently induces an $O(n)$ dependence on $e_{b+1}$, that is

$$e_{b+1} \leq \frac{1}{2n}e_b + O(n\eta^2 L^2 G^2) \leq O(n\eta^2 L^2 G^2).$$

It is not hard to verify that we can only obtain $O(\omega^{-1/2}\rho^{-1}n^{5/4}\sqrt{\ln n}\sqrt{T})$ and $O(\omega^{-1}\rho^{-2}n^{3/2}\ln n \ln T)$ regret bounds for convex and strongly convex loss functions.

