# OpenReview forum: "Improved Regret for Decentralized Online Convex Optimization with Compressed Communication"
_ICLR.cc/2026/Conference — Submitted to ICLR 2026_

### Official Review · Reviewer_is8i · 2025-11-02

**Soundness:** 3
**Presentation:** 3
**Contribution:** 2
**Rating:** 4
**Confidence:** 4

**Summary:**

The paper investigates the problem of decentralized online convex optimization (D-OCO) where communication between $n$ learners is compressed. The paper proposes a new algorithm, based on a two-level blocking framework designed to mitigate the three primary sources of error: consensus error, compression error, and projection error. The authors claim their algorithm achieves improved regret bounds of $\tilde{O}(\omega^{-1/2}\rho^{-1}n\sqrt{T})$ for convex functions and $\tilde{O}(\omega^{-1}\rho^{-2}n\ln T)$ for strongly convex functions. Furthermore, the paper establishes lower bounds for this problem, which the authors claim justify the optimality of their results. The framework is also extended to the bandit feedback setting, where similar improvements are claimed.

**Strengths:**

* The paper identifies a weakness in the prior art, specifically the D-OCO algorithm from Tu et al. (2022), which suffers from a high-order polynomial dependence on the inverse compression ratio ($\omega^{-1}$) and the inverse spectral gap ($\rho^{-1}$). The paper proposes a new algorithm, Top-DOGD, which achieves an exponential improvement in these dependencies, reducing the regret terms from $O(\omega^{-4}\rho^{-8})$ to $\tilde{O}(\omega^{-1/2}\rho^{-1})$.

**Weaknesses:**

*  While the improved bounds stated by the paper are correct, the algorithm used to achieve it is a synthetic combination of components drawn directly from the existing offline distributed optimization literature (Koloskova et al. (2019) and Huang et al. (2022)), an approach which lacks algorithmic innovation. The concept of using a "blocking" update to amortize multiple communication rounds within a single gradient step is also a standard, non-novel paradigm.

* The paper's lower bound technique is also a direct and incremental adaptation of existing methods. The authors state their proof "follows that of Wan et al. (2024)". Their sole modification is to "incorporate a dedicated compressor", which they model as a "randomized gossip" protocol where communication succeeds with probability $\omega$. This, in turn, modifies the expected communication delay by a factor of $1/\omega$. This exact idea---constructing a lower bound by modeling the compressor as a probabilistic communication failure over a hard-to-communicate graph structure---was already introduced in Huang et al. (2022), which the authors fail to properly cite. Huang et al. constructed a zero-chain function split into two components and assigned them to different workers, forcing communication for the algorithm to make progress. They then introduced a "rand-s" compressor and explicitly modeled its effect as a probabilistic failure to transmit the necessary information, directly linking the convergence rate to the compression parameter.

* Notwithstanding the above weaknesses, I believe the paper is not of interest and relevance to the ICLR community and is more appropriate for a more theoretical venue like ICML, NeurIPS, JMLR, or control journals like Automatica and IEEE TAC.

**Questions:**

* Please compare your method with that of Cao & Basar (2023) which seem to be the closest to this work in terms of achieving the same regret.

---

> ### Author Response · Authors · 2025-11-20
>
> Many thanks for the constructive reviews!
>
> ---
>
> __Q1__: ... an approach which lacks algorithmic innovation.
>
> __A1__: We acknowledge our method builds on the existing techniques [Koloskova et al., 2019; Huang et al., 2022].
>
>
>
> However, directly performing multi-step Choco gossip [Koloskova et al., 2019] results in multiple communication rounds per update. To address it, we design an online compressed gossip strategy by using the blocking update technique to reduce the communication times per update.
>
> Moreover, in D-OCO, each learner must project its decision onto the feasible domain at every round. This introduces an additional projection error. Only using the online compressed gossip strategy cannot handle this issue and only yields unsatisfactory regret bounds, which we also mention in our paper (Line 364-373). To deal with this, we propose a novel projection error compensation scheme by using the multi-step compression [Huang et al., 2022] to mitigate the projection error.
>
>
>
> Furthermore, we design a _two-level blocking update framework_ to reduce the communication rounds and incorporate the two strategies, which enable us to effectively control the consensus error, compression error, and projection error at the same time. These techniques are specifically tailored for the D-OCO setting rather than being a straightforward extension of existing offline decentralized optimization methods with compression.
>
> ---
>
> __Q2__: The paper's lower bound technique is also a direct and incremental adaptation of existing methods.
>
> __A2__: While both our proof and Huang et al. [2022] employ the concept of transmission failures, their proof cannot be directly applied to derive lower bounds for D-OCO. Huang et al. [2022] study the offline distributed optimization where the loss function remains fixed across all rounds and the central server can aggregate information from all workers simultaneously. In contrast, our work addresses online decentralized optimization where loss functions change adversarially at each round and information propagates learner-by-learner through the network graph without any central coordinator. Moreover, the construction of the loss functions in our paper is different from that of Huang et al. [2022]. To clearly distinguish these contributions, we will cite Huang et al. [2022] in the revised version and make these distinctions explicit.
>
> ---
>
> __Q3__: Compare method with that of Cao & Basar [2023].
>
> __A3__:  Cao & Basar [2023] is a contemporaneous work with Tu et al. [2022], and both works adopt essentially the same algorithmic design. Both works perform a single round of Choco gossip [Koloskova et al., 2019] per update. As a result, the algorithms of Cao & Basar [2023] and Tu et al. [2022] inherit the same structure and enjoy the same order of regret guarantees.
>
> Our algorithm adopts a two-level blocking update framework that incorporates the online compressed gossip and projection error compensation strategies, and effectively controls the consensus error, compression error, and projection error at the same time.

---

### Official Review · Reviewer_RFz8 · 2025-11-03

**Soundness:** 2
**Presentation:** 2
**Contribution:** 2
**Rating:** 2
**Confidence:** 4

**Summary:**

This work studies decentralized online convex optimization with compressed communication. A two-level blocking method—combining online gossip and error compensation—yields improved regret for convex and strongly convex losses with optimal dependence on the compression quality and spectral gap (backed by new lower bounds), and extends to bandit feedback via standard gradient estimators.

**Strengths:**

1. Provide improved lower bounds for convex online programming. Present a lot of results, e.g., improved upper bounds, new lower bounds, and extension to bandit feedback scenario.

2. Established lower bounds for convex online programming. Proved that the upper bound matches with lower bound in terms of $T$ and $\omega$.

3. Extend the algorithm to the bandit feedback scenario.

**Weaknesses:**

1. **Novelty**. I am concerned about the work’s novelty. The method largely combines two well-established components—multi-step gossip and multi-step compression—both already used to attain optimal rates in decentralized optimization and communication compression. It seems that the paper does not offer sufficiently new insights or algorithmic innovations.

2. **Dependence on $\omega$ and $\rho$**. I am confused that the proposed bounds exhibit different dependences on $\omega$ and $\rho$ between the convex and strongly convex settings. Prior results [A1] and [A2] indicate a $\rho^{-1/2}$ lower-bound dependence regardless of convexity, and [A3] suggests a uniform $\omega^{-1}$ dependence across strongly convex, convex, and nonconvex regimes. Please clarify why the proposed bounds exhibit different dependences on $\omega$ and $\rho$.

3. **Incomplete lower-bound characterization**. The lower-bound proofs appear to rely on restrictive assumptions—bounded gradients ($G$) and bounded domain diameter ($D$)—which are not used in decentralized stochastic optimization [A2] or compressed stochastic optimization. Please justify the necessity of these assumptions. In addition, the bounds do not make explicit the dependence on $L$ and $\mu$; for strongly convex problems one typically expects a factor of $\sqrt{L/\mu}$, which is not reflected here.

4. **Lower bounds for general vs. specific $\rho$**. Theorems 3 and 4 claim lower bounds for any $\rho\in(0,1)$. However, to the best of my knowledge, the result in [A4] (the journal version of [A2]) for stochastic optimization—which is closely related to online optimization—holds only for the specific choice $\rho=\cos(\pi/n)$. Please clarify how your argument extends to arbitrary $\rho$.

5. **Limited experiments**. The experiments are mainly focused on logistic regression. It is suggested to conduct modern experiments such as ResNet image classification and LLM fine-tuning.

[A1] Optimal algorithms for smooth and strongly convex distributed optimization in networks

[A2] Optimal Complexity in Decentralized Training

[A3] Lower Bounds and Accelerated Algorithms in Distributed Stochastic Optimization with Communication Compression

[A4] Decentralized Learning: Theoretical Optimality and Practical Improvements (The journal version of A2)

**Questions:**

1. Highlight the novelty and insights.

2. Clarify why the proposed bounds exhibit different dependences on $\omega$ and $\rho$.

3. Justify the necessity of bounded gradient and bounded domain in lower bounds.

4. Clarify the dependence on $L$ and $\mu$ in the lower bounds.

5. Conduct modern experiments rather than the simple toy example on logistic regression.

---

> ### Author Response · Authors · 2025-11-20
>
> Many thanks for the constructive reviews!
>
> ---
>
> __Q1__: Highlight the novelty and insights.
>
> __A1__: We acknowledge that our method incorporates multi-step gossip and multi-step compression, but we would like to emphasize that the novelty of our work also includes a _projection error compensation scheme_ and a _two-level blocking update framework_.
>
> In D-OCO, each learner must project its decision onto the feasible domain at every round. This introduces an additional projection error. Directly applying multi-step gossip and multi-step compression cannot control this error and yields unsatisfactory regret bounds, which we also mention in our paper (Line 364-373). To deal with this, we propose a novel projection error compensation scheme to mitigate the projection error. Furthermore, we design a _two-level blocking update framework_ to reduce the communication rounds and incorporate the two strategies, which enables us to effectively control the consensus error, compression error, and projection error at the same time. These two techniques are specifically tailored for the D-OCO setting rather than being a straightforward extension of existing offline decentralized optimization methods with compression.
>
> Moreover, we would like to draw your attention to the contribution that our paper establishes the _first_ lower bounds for D-OCO with compressed communication that explicitly characterize the effect of the compression operator.
>
> ---
>
> __Q2__. Why the proposed bounds exhibit different dependences on $\omega$ and $\rho$.
>
> __A2__. The different dependencies on $\omega$ and $\rho$ stem from the distinct analysis of the convex and strongly convex settings.
>
> In our algorithm design, we set the block size $L = O(\omega^{-1}\rho^{-2}\ln n)$.
>
> In the convex setting, the regret is $\mathbb{E}[R(T,i)]\leq O(n\sum_{b=1}^{T/L}\left(\frac{1}{\eta_b}-\frac{1}{\eta_{b-1}}\right) + nL^2\sum_{b=1}^{T/L}\eta_b)$, where $\eta_b$ is the learning rate at block $b$ and we assume $1/\eta_0 = 0$. To minimize the regret, we set $\eta_b = O(1/\sqrt{LT})$, which further leads to an $O(n\sqrt{LT})=O(\omega^{-1/2}\rho^{-1} n\sqrt{\ln n}\sqrt{T})$ regret bound.
>
> In the $\mu$-strongly convex setting, the regret is $\mathbb{E}[R(T,i)]\leq O(n\sum_{b=1}^{T/L}(\frac{1}{\eta_b}-\frac{1}{\eta_{b-1}}-\mu L)+ nL^2\sum_{b=1}^{T/L}\eta_b)$, where $\eta_b$ is the learning rate at block $b$ and we assume $1/\eta_0 = 0$. To exploit the curvature and improve the dependence on $T$, we set $\eta_b=1/(\mu bL)$ at block $b$. This choice leads to the regret $R(T,i)\leq O(\mu^{-1} nL\ln T)=O(\mu^{-1}\omega^{-1}\rho^{-2}n\ln n\ln T)$. While the dependence on $\omega$ and $\rho$ becomes worse in the strongly convex case, the dependence on $T$ is significantly better.
>
> ---
>
> __Q3__: Justify the necessity of bounded gradient and bounded domain in lower bounds.
>
> __A3__: First, we would like to clarify that the assumptions of bounded gradients ($G$) and a bounded domain ($D$) are standard in online convex optimization and are widely adopted in prior work [1,2].
>
>
>
> Second, the work [3] studies the *offline stochastic non-convex* setting and assumes _smoothness_ of the loss functions. Our paper investigates decentralized online convex optimization for non-smooth loss functions. Because the problem is fundamentally different, the assumptions are not comparable.
>
> ---
>
> __Q4__: Clarify the dependence on $L$ and $\mu$ in the lower bounds.
>
> __A4__: The lower bound for $\mu$-strongly convex loss functions is $\Omega\left(\frac{(\log_{16}(30\omega(T-1)/n)-2)(n-2)\pi\mu D^2}{2^{22}\omega\rho^{1/2}}\right)$, which depends linearly on $\mu$ (Theorem 4). Regarding $L$, in our algorithm $L$ denotes the block size used in the two-level blocking framework, and it does not appear in the lower bound proof.
>
> ---
>
> __Q5__: Lower bounds for general vs. specific $\rho$.
>
> __A5__: We take the convex setting as an example. In our proof, we first construct a $1$-connected graph and use the Randomized Gossip compressor. Based on this, we derive an $\Omega(\frac{nDG\sqrt{nT}}{16\sqrt{2\omega}})$ lower bound. Next, we introduce a lemma from [2]: for any $1$-connected cycle graph, there exists a gossip matrix $P$ satisfying Assumption 1, which satisfies $\frac{\pi^2}{\rho}\leq 4n^2.$ By using this lemma, we can derive the $\Omega(\frac{nGD\sqrt{\pi T}}{32\omega^{1/2}\rho^{1/4}})$ lower bound, which holds for general $\rho$.
>
> ---
>
> __Reference__
>
> [1] Introduction to Online Convex Optimization. Foundations and Trends in Optimization, 2016.
>
> [2] Nearly Optimal Regret for Decentralized Online Convex Optimization. COLT, 2024.
>
> [3] Optimal Complexity in Decentralized Training. ICML, 2021.

---

### Official Review · Reviewer_18Zx · 2025-11-03

**Soundness:** 3
**Presentation:** 3
**Contribution:** 2
**Rating:** 4
**Confidence:** 2

**Summary:**

This paper investigates decentralized online convex optimization (D-OCO) with compressed communication, where $n$ learners collaboratively minimize a sequence of global loss functions using only local information and compressed data from neighbors. The authors propose Top-DOGD, a novel algorithm achieving improved regret bounds of $\tilde{O}(\omega^{-1/2}\rho^{-1}n\sqrt{T})$ and $\tilde{O}(\omega^{-1}\rho^{-2}n \ln T)$ for convex and strongly convex functions respectively, significantly improving upon prior work's quadratic/quartic dependence on $\omega^{-1}$. The paper also establishes matching lower bounds and extends results to bandit feedback settings.

**Strengths:**

1. The paper provides substantial theoretical improvements in regret bounds for D-OCO with compressed communication, specifically reducing the dependence on the compression parameter $\omega$ (from quadratic/quartic to linear/sublinear) and the spectral gap $\rho$.

2. The two-level blocking update framework—blending an online compressed gossip strategy with a projection error compensation scheme—demonstrates a creative approach to balancing communication constraints with optimization speed.

**Weaknesses:**

1. The paper is hard to follow, and the writing should be improved
2. In a few places, explanation could be made crisper. For example, in Algorithm 2 (Page 6), some variable reuse (e.g., $\hat{\mathbf{y}}{j}^{(b_1)}(b)$, $\hat{\mathbf{y}}{i}^{(b_1)}(b)$) is a source of confusion, especially since projections, auxiliary variables, and block indices are deeply intertwined. Further, the definition of certain parameters (e.g., constants hidden in $O(\cdot)$ in Theorems 1–2) is not always explicit, which could impede reproducibility or direct application.
3. require knowing $\rho, \omega, \beta$ (matrix norms), which may be unknown or drift in practice.

**Questions:**

How to set $L_1, L_2, \gamma$ without $\rho, \omega$ ? Can you provide adaptive or data-driven rules that don't need prior spectral/compression knowledge-e.g., doubling tricks or online estimation of $\rho$ / compressor quality?

---

> ### Author Response · Authors · 2025-11-20
>
> Many thanks for the constructive reviews!
>
> ---
>
> __Q1__: In a few places, explanation could be made crisper.
>
> __A1__: Thanks for your suggestion. We will refine the exposition to make it crisper.
>
> ---
>
> __Q2__: The definition of certain parameters is not always explicit.
>
> __A2__: We apologize for this. In our paper, we follow the common presentation style in the OCO literature, where the gradient norm bound $G$ and the domain diameter $D$ are typically omitted in the big-$O$ notation [1, 2]. Similarly, in line with prior work on decentralized optimization [3], we omit the matrix norm $\beta$, which does not affect the order of the regret.
>
>
>
> For clarity, we provide the regret bounds including these factors in the following:
>
> |               |            $\qquad $$\qquad $Convex$\qquad $             |           $\qquad $Strongly convex$\qquad $            |
> | :-----------: | :------------------------------------------------------: | :----------------------------------------------------: |
> | Regret bounds | $O(\omega^{-1/2}\rho^{-1}DG\beta n\sqrt{\ln n}\sqrt{T})$ | $O(\mu^{-1}\omega^{-1}\rho^{-2}G^2\beta^2n\ln n\ln T)$ |
>
> ---
>
> __Q3__: How to set parameters without knowing $\rho,\omega,\beta$?
>
> __A3__: In decentralized optimization problem, the communication matrix $P$ is assumed to be known in advance, which is standard in the literature [2, 3]. Once $P$ is given, the parameters $\rho$ and $\beta$ are also known. Similarly, the compressor is given to the algorithm in advance, and its compression ratio $\omega$ is therefore available. This is also a widely used assumption in prior work on compressed decentralized optimization [3, 4].
>
> ---
>
> __References:__
>
> [1] Introduction to Online Convex Optimization. Foundations and Trends in Optimization, 2016.
>
> [2] Nearly Optimal Regret for Decentralized Online Convex Optimization. COLT, 2024.
>
> [3] Decentralized Stochastic Optimization and Gossip Algorithms with Compressed Communication. ICML, 2019.
>
> [4] Lower Bounds and Nearly Optimal Algorithms in Distributed Learning with Communication Compression. NeurIPS, 2022.
>
> ---
> We hope that our responses can address your concerns, and we would greatly appreciate it if you could re-evaluate the contributions of our work.

---

### Official Review · Reviewer_kFty · 2025-11-04

**Soundness:** 3
**Presentation:** 2
**Contribution:** 3
**Rating:** 2
**Confidence:** 4

**Summary:**

This paper studies decentralized online convex optimization (D-OCO) under compressed communication, where multiple agents cooperate over a network but exchange only quantized or compressed messages. Prior work (e.g., Tu et al., 2022) achieved regret bounds that scale poorly with the compression factor $\omega$, the network spectral gap $\rho$, and the number of agents $n$.

The authors propose a new algorithm, called **Top-DOGD**, which employs a two-level blocking strategy that combines compressed gossip updates with a projection-based correction step. The method achieves improved regret bounds:
$$
\tilde{O}(\omega^{-1/2}\rho^{-1}n\sqrt{T}) \text{ for convex losses, and }
\tilde{O}(\omega^{-1}\rho^{-2}n\ln T) \text{ for strongly convex losses,}
$$
along with lower bounds
$$
\Omega(\omega^{-1/2}\rho^{-1/4}n\sqrt{T}) \quad \text{and} \quad
\Omega(\omega^{-1}\rho^{-1/2}n\ln T),
$$
which nearly match the upper rates up to logarithmic factors. The paper also extends the framework to bandit feedback and provides regret guarantees using both one-point and two-point gradient estimators.

**Strengths:**

The paper's writing is organized, and the results are internally consistent. It makes a solid theoretical contribution by tightening the dependence of regret on the compression factor and network parameters. The improvement in the convex setting from $\omega^{-2}$ to $\omega^{-1/2}$ (and similar for $\rho$) is meaningful and clearly improves the state of the art, likewise for the strongly convex setting. The analysis is technically careful, with an intuitive exposition that demarcates consensus error, compression error, and projection error. The overall approach remains quite simple—blocking and error compensation are standard ideas, but combining them in this way for D-OCO under compression appears to be novel and is well-executed. The lower bounds are also valuable, as they lend credibility to the claim of near-optimality.

**Weaknesses:**

**Missing References and the Issue with First-order Feedback**

I believe that some essential references are missing, including [Patel et al.](https://proceedings.mlr.press/v202/patel23a.html) and the papers cited within. Discussing these references from the centralized setting, along with their results, will provide additional context for the paper's findings.

Specifically, Patel et al. point out that in the centralized setting, there is no benefit of collaboration when first-order information is available on each client. I believe this remains true in the context of this paper, and with non-collaborative OGD on each client, there is no need to do any compression or communication whatsoever. Could the authors clarify this issue? From what I can see, there appears to be no benefit to increasing $n$ in the regret guarantees, which makes me wonder if the first-order setting is essentially pointless to study. Unless I am missing something, this is my primary concern.

Another issue I have is about the regret minimization problem (1) itself. In principle, solving an online problem collaboratively only seems sensible if there is indeed some shared information between the clients. In the extreme case of distributed stochastic optimization, this is achieved by assuming some data similarity between the clients. However, even in the online case [Patel et al.](https://proceedings.mlr.press/v202/patel23a.html) examined the effect of having bounded gradient dissimilarity across clients, and their two-point bandit feedback algorithm does improve with lower heterogeneity. Can this benefit be seen in this paper's analysis as well? Perhaps for smooth online convex optimization?

Finally, I am curious if Algorithm 2 of [Patel et al.](https://proceedings.mlr.press/v202/patel23a.html) can also be implemented using the ideas introduced in this paper. And if so, what guarantees of regret can be obtained? The projection error might need to be handled differently for this algorithm.

**Lower Bounds**

While the lower bounds seem correct, they are based on the worst possible compressors satisfying the paper's assumptions, which allows for the use of a very unstable compressor that, with some probability, outputs zero. I think this is a very pathological compressor, and I wonder if the authors could consider a more reasonable class of compressors and provide a lower bound for those. I understand that this might require changing the assumption on the compressor, and the upper bounds might improve as well (for instance, if the authors explicitly assume something like a randK compressor).

While I currently give a score of 2, I am open to increasing it if my queries are answered, because I do believe there is technical novelty in the paper.

**Questions:**

See weaknesses above.

---

> ### Author Response · Authors · 2025-11-20
>
> Many thanks for your constructive reviews and bringing this work [1] to our attention! We want to clarify that the __problem setting__ and __regret definition__ of [1] are fundamentally different from our work. We hope the reviewer could reevaluate our paper, and are very happy to respond more questions during the reviewer-author discussion period.
>
> ---
>
> __Q1__: There is no need to do any compression or communication whatsoever. Could the authors clarify this issue?
>
> __A1__: We believe this is a misunderstanding because our setting and regret definition differ fundamentally from those in [1].
>
> - [1] measures the performance of learners by using the regret defined as $R^\prime(T)=\sum_{i=1}^n\left(\sum_{t=1}^Tf_{t,i}(\mathbf{x}_ i(t))-f_ {t,i}(\mathbf{x}^\star)\right)$, where $\mathbf{x}^*=\arg\min_{\mathbf{x}\in \mathcal{X}}\sum_{t=1}^T\sum_{i=1}^nf_{t,i}(\mathbf{x})$. This regret measures the performance of learner $i$ only using the local loss functions $\sum_{t=1}^Tf_{t,i}(\mathbf{x}_i(t))$.
>
>   Remarkably, $R^\prime(T)$ can be bounded by the sum of the _local regret_ of each learner, that is,
>
>    $R^\prime(T)\leq\sum_{i=1}^n\underbrace{\left(\sum_{t=1}^Tf_{t,i}(\mathbf{x}_ i(t))-f_ {t,i}(\mathbf{x}_ i^\star)\right)}_{\text{local regret}},$
>
>   where $\mathbf{x}_ i^*=\arg\min_ {\mathbf{x}\in \mathcal{X}}\sum_ {t=1}^Tf_ {t,i}(\mathbf{x})$. In the full information setting, by running OGD solely with local gradients, each learner is able to attain the optimal local regret. Thus, there is no benefit of collaboration.  In contrast, we aim to minimize the _global regret_ defined as $R(T,i) = \sum_{t=1}^T\sum_{j=1}^n f_{t,j}(\mathbf{x}_ i(t)) - \sum_ {t=1}^T\sum_{j=1}^n f_{t,j}(\mathbf{x}^\star).$  For each learner $i$, our regret measures its performance by using the global functions $\sum_{j=1}^n\sum_{t=1}^Tf_{t,j}(\mathbf{x}_ i(t))$. Intuitively, this requires each learner to choose a decision that minimizes the sum of *all learners’* loss functions. Performing local OGD is insufficient to minimize the global regret because learner $i$ cannot access information about other learners' loss functions $f_{t,j}(\cdot), j\neq i$, which are necessary to compute the global gradient. Therefore, communication with neighbors in decentralized online convex optimization is essential.
>
> -  [1] investigates the _distributed_ (federated) setting where $n$ workers are connected to a central server to exchange information. In contrast, we focus on the _decentralized_ setting where each learner communicates with its neighbors based on a weight matrix $P$ (no central coordinator).
>
> ---
>
> __Q2__: ...effect of having bounded gradient dissimilarity across clients... Can this benefit be seen in this paper's analysis as well?
>
> __A2__: Yes, we can achieve this. In our proof, we only assume the bounded gradient of each loss function, that is, $\\|\nabla f_{t,i}(\mathbf{x})\\|^2\leq G^2$, which means $\\|\nabla f_{t,i}(\mathbf{x})-\frac{1}{n}\sum_{j=1}^n\nabla f_{t,j}(\mathbf{x})\\|^2\leq 4G^2$. If we assume similarity between the learners, i.e., $\\|\nabla f_{t,i}(\mathbf{x})-\frac{1}{n}\sum_{j=1}^n\nabla f_{t,j}(\mathbf{x})\\|^2\leq \alpha\leq4G^2$, we can obtain tighter regret bounds.
>
> ---
>
> __Q3__: If Algorithm 2 of [1] can also be implemented using the ideas introduced in this paper.
>
> __A3__: Regarding the problem setting in [1], it seems highly non-trivial to implement their Algorithm 2 using the online gossip strategy and error compensation scheme introduced in our paper.
>
> The work [1] focuses on the *distributed* setting with a central server that aggregates local information from learners. Since the central server can directly aggregate learners' local information, we cannot adopt the online gossip strategy to achieve consensus among learners. Our error compensation scheme is specifically tailored to the online compressed gossip strategy and is therefore not applicable to the setting of [1]. Addressing projection errors in  the distributed setting requires developing new techniques.
>
> ---
>
> __Q4__: I wonder if the authors could consider a more reasonable class of compressors and provide a lower bound for those.
>
> __A4__: Thanks for your insightful comment. We want to clarify that the Randomized Gossip compressor is also used in previous work [3]. The purpose of employing this compressor in our proof is to construct the worst-case scenario for the general compressor class (Definition 1).  While other compressors such as $\text{Rand}_k$ may achieve smaller regret, it does not characterize the worst-case behavior of all compressors considered in Definition 1.
>
>
>
> ---
>
> __References__
>
> [1] Federated Online and Bandit Convex Optimization. ICML, 2023.
>
> [2] EF21-P and Friends: Improved Theoretical Communication Complexity for Distributed Optimization with Bidirectional Compression. ICML, 2023.
>
> [3] Decentralized Stochastic Optimization and Gossip Algorithms with Compressed Communication. ICML, 2019.

---

### Meta-Review · Area_Chair_wZGg · 2025-12-26

**Summary:**

The reviews broadly agree that the paper provides a careful theoretical treatment of decentralized online convex optimization with compressed communication, and that it improves the regret dependence on compression quality and network connectivity relative to prior work, notably (Tu et al., 2022). Reviewers also viewed the inclusion of lower bounds that explicitly reflect the compression parameter as a meaningful supporting contribution. The primary reason for a negative recommendation is that the proposed algorithm is perceived as a combination of established ingredients (multi-step gossip, blocking, and repeated compression), and multiple reviewers were not convinced that the resulting design provides a sufficiently new conceptual insight beyond known techniques. In addition, a limitation of the claimed upper bound contribution is that the analysis relies on relatively strong regularity assumptions (in particular, bounded gradients, together with a bounded feasible set), which restricts the scope of the theoretical improvement and leaves open whether similar compression and network dependences can be achieved under weaker conditions.

**Reviewer Concerns:**

A main concern raised by reviewer kFty is the definition of regret and how it contrasts with closely related settings, such as those presented in Patel et al. (2023). The rebuttal argues that the regret notions and problem settings differ, but the response is mostly verbal and would have been stronger with a short, explicit mathematical comparison that makes the distinction unambiguous. As written, it partially clarifies the authors’ intent, but it does not fully resolve the reviewer’s concern. A second main concern, shared by multiple reviewers, is novelty: several reviews view the algorithm as largely combining known techniques. The rebuttal emphasizes that handling projection error via a correction step is a key novel and necessary ingredient. However, skepticism persists about whether this constitutes a sufficiently novel conceptual contribution. Beyond these two points, several smaller issues were at least partially addressed in the rebuttal, including clarification of why the convex and strongly convex bounds exhibit different parameter dependencies, clarification of which constants are hidden in big O notation, and additional discussion of baselines. Requests for broader experiments were not addressed, but this is a secondary issue given the paper’s primarily theoretical scope.

**Reviewer Scores:**

Reviewer kFty explicitly indicates openness to a higher score if their questions are resolved. The rebuttal’s explanation makes an increase plausible, potentially up to 4, but I suspect further back-and-forth would still be needed because the response does not fully pin down the mathematical distinction to Patel et al. (2023) and its related technical implications.

For reviewer RFz8, the rebuttal addresses several concrete questions (for example, the differing parameter dependencies across convex versus strongly convex regimes and the stated sources of novelty). This could plausibly have moved the score upward, likely to 4, but not higher, since the reviewer’s central novelty concerns remain largely open.

---

### Decision · Program_Chairs · 2026-01-26

Reject